# Perimenopausal state oestradiol to progesterone imbalance drives Alzheimer's risk via ERRα dysregulation and energy dyshomeostasis

Jacquelyne Ka-Li Sun[1], Amy Zexuan Peng[1], Ronald P. Hart [2], Karl Herrup [3], Deng WU[1], Genper Chi-Ngai Wong[1] & Kim Hei-Man Chow [1,4] ✉

Sex-biased differences in Alzheimer's disease (AD) are well documented, but the mechanisms underlying increased vulnerability in postmenopausal women remain unclear. This study aimed to model the effects of perimenopausal hormonal fluctuations on AD pathophysiology. Using a VCD-induced accelerated ovarian failure model in young female C57BL/6 J and 3xTg mice, we simulated a perimenopausal state with hormonal changes characterised by elevated oestradiol levels and reduced progesterone levels. Supporting human brain transcriptomic and metabolomic data from the ROSMAP study revealed that impaired oestrogen-related receptor alpha (ERRα) function was a key driver of female sex-biased vulnerability. In female mice, progesterone-guided oestrogen receptor signalling maintained ERRα activity by regulating neuronal cholesterol homoeostasis and the TCA cycle. Hormonal imbalances disrupted this mechanism, triggering an aspartate-driven "minicycle," which increased glutamate release, neuronal excitability, ATP depletion, and energy crisis susceptibility. This study demonstrates how perimenopausal hormonal imbalances exacerbate AD risk via ERRα dysfunction, linking neuronal cholesterol and energy homeostasis to disease vulnerability.

Comprehending the critical influence of biological sex on health and disease is fundamental to advancing personalised medicine[1,2]. Females have been hypothesised to play a central role in the late-onset Alzheimer's disease (LOAD) epidemic. Although studies from North and South American populations have not consistently identified biological sex-biased differences in LOAD incidence, research conducted in Europe and Asia suggests a higher incidence of the disease among women, particularly those older than 80 years[3,4]. Human autopsy studies and investigations using murine models of AD have consistently demonstrated that females exhibit elevated levels of amyloid-beta (Aβ) plaques and neurofibrillary tangles, indicating potential biological sex-

specific differences in the underlying neuropathology[5-7]. Furthermore, with respect to the degree of neuropathological burden observed at autopsy, compared with men, women consistently experience more pronounced cognitive decline and accelerated hippocampal atrophy[8,9]. Collectively, these findings highlight a significant public health challenge that warrants focused attention.

Age is widely acknowledged as the primary risk factor for LOAD[10], and the higher incidence of this disease in females may, in part, be attributed to their greater life expectancy[11]. However, longevity alone does not sufficiently explain the lower prevalence of LOAD in younger women than in older females people[12]. Oestrogen and progesterone,

[1]School of Life Sciences, Faculty of Science, The Chinese University of Hong Kong, Hong Kong, Hong Kong. [2]Department of Cell Biology and Neuroscience, Rutgers University, Piscataway, NJ, USA. [3]Department of Neurobiology, School of Medicine, University of Pittsburgh, Pittsburgh, PA, USA. [4]Gerald Choa Neuroscience Institute, The Chinese University of Hong Kong, Hong Kong, Hong Kong. ✉e-mail: heimanchow@cuhk.edu.hk

the predominant ovarian hormones, exert opposing effects on the brain depending on their relative dominance during different phases of the menstrual cycle. At the molecular level, the activated progesterone receptor (PR) functions as a molecular rheostat for the oestrogen receptor (ER), modulating its chromatin binding and transcriptional activity[13]. In vitro studies have shown that oestradiol promotes the formation of dendritic spines in hippocampal pyramidal neurons by suppressing the inhibitory signalling of GABAergic interneurons[14]. However, the addition of progesterone counteracts the effects of oestradiol through its downstream metabolite, tetrahydroprogesterone, which enhances GABAergic inhibition[14]. This dynamic interplay between oestrogen and progesterone, which governs the balance of excitation and inhibition, is further reflected in vivo. Phases of the menstrual cycle characterised by a high oestrogen-to-progesterone ratio are associated with increased excitatory input[15-17] and enhanced learning and memory functions[18,19]. Nevertheless, the persistence of an imbalance favouring high oestradiol and low progesterone levels, as frequently observed during perimenopause (lasting approximately 6–9 years)[20-24], has been linked to heightened susceptibility to seizures[25], stroke[26] and binge drinking[27]− conditions recognised as risk factors for Alzheimer's disease and Alzheimer's disease-related dementias (AD/ADRD)[28-30]. During perimenopause, this hormonal imbalance is reported in up to one-third of all cycles and occurs predominantly in women aged 40–58 years[20-24]. Furthermore, randomised clinical trials and observational studies have indicated that the perimenopausal phase is particularly associated with memory decline and an increased risk of mild cognitive impairment and dementia[31-36]. Despite these findings, the relationship between hormone replacement therapy (HRT) and dementia risk has remained a topic of intense debate. The Women's Health Initiative Memory Study (WHIMS) is the only randomised controlled trial to investigate the effects of HRT on dementia incidence. The study concluded that oestrogen-progestin therapy doubled the risk of all-cause dementia[37,38]. However, a significant limitation of the WHIMS trial was its use of progestin, a synthetic analogue of progesterone. Progestin can be metabolised into oestrogenic[39] or androgenic compounds, which activate ERs or glucocorticoid receptors[40], respectively, potentially amplifying the effects of supplemented oestrogen. This phenomenon, therefore, may inadvertently reproduce the condition of high oestradiol-low progesterone imbalance. Moreover, the WHIMS findings were derived exclusively from postmenopausal women aged 65 years or older, a population unlikely to reverse brain damage initiated by hormonal changes occurring during perimenopause a decade earlier[37,38]. These observations underscore the critical importance of maintaining a dynamic and balanced ratio of oestrogen to progesterone for women's health, particularly for preserving brain health as they age[41].

The 17β-oestradiol (E2), the predominant form of ovarian oestrogen produced during the early or mid-follicular phase of the human menstrual cycle[42] (analogous to the morning of the proestrus phase in mice), and progesterone (P4), predominantly produced during the luteal phase (analogous to the evening of the proestrus phase in mice)[21,43,44], are highly lipophilic hormones capable of crossing the blood–brain barrier (BBB)[45]. While these hormones are traditionally recognised for their regulatory roles in fertility and reproduction, signalling through oestrogen receptor alpha (ERα) and PR has also been implicated in the modulation of peripheral fuel metabolic homoeostasis[46,47]. This raises the possibility that, upon crossing the BBB, these female sex hormones may directly influence brain metabolic networks, potentially contributing to the female-biased vulnerability observed in AD.

In this study, we report that female subjects affected by Alzheimer's disease in the Religious Orders Study and Memory and Ageing Project (ROSMAP) cohort exhibited more pronounced transcriptomic and metabolomic changes in the brain than their male counterparts did. To model the effects of peripheral hormonal changes on the brain, we employed a VCD-induced accelerated ovarian failure (AOF) mouse model. Our findings revealed unexpected regulatory connections between hormonal signalling, neuronal cholesterol metabolism, and bioenergetic balance. This disruption in homeostasis rewires the N-acetyl-aspartyl-glutamate (NAAG) metabolic axis, leading to increased spontaneous postsynaptic activity that depletes cellular ATP levels. These changes undermine neuronal resilience to additional excitatory stimuli or insults, culminating in an accelerated decline in electrophysiological and behavioural functions in a mouse model of AD.

## Results

### Female patients with LOAD exhibit a more pronounced decline in the ERRα-regulated bioenergetic network within neurons

Previous neuroimaging and blood biomarker studies have implicated brain hypometabolism as a contributor to the biologically sex-dimorphic effects observed in LOAD[48,49]. To explore this further, bulk transcriptomic analysis was performed on brain tissue from 613 participants in the ROSMAP cohort[50]. This cohort included 133 males and 219 females who were dementia free, as well as 88 males and 173 females who were diagnosed with LOAD (Fig. 1a–d and Supplementary Data 1). Phenotypic classification (LOAD vs nondementia [ND]) was determined on the basis of clinicopathological features, including neuritic plaque load (CERAD score), neurofibrillary tangle pathology (Braak stage), and cognitive status (Cogdx and DCFDX) (Fig. 1a). Differentially expressed gene (DEG) analysis comparing LOAD and ND samples within each biological sex revealed a substantially greater number of DEGs in females (6615 DEGs) than in males (439 DEGs) (Fig. 1b). Notably, the small number of DEGs identified in males did not cluster into any meaningful pathways. In contrast, the DEGs that were upregulated in the female LOAD group were significantly enriched in neuroinflammatory pathways, including TNFα signalling via NF-κB, IL-2/STAT5 signalling, TGFβ signalling, and IL-6/JAK/STAT3 signalling. In addition, the downregulated DEGs in females were enriched primarily in pathways related to mitochondrial oxidative phosphorylation (OXPHOS) (Fig. 1c). These observations were validated as disease-specific changes in female LOAD subjects rather than inherent biological sex differences between male and female ND participants (Supplementary Fig. 1).

Among the enriched pathways, OXPHOS was the most significant, with the smallest adjusted $p$-value. Analysis of potential transcriptional regulators of the enriched genes revealed that oestrogen-related receptor alpha (ERRα; also known as NR3B1) functioned as a key regulator (Fig. 1d). ERRα is a nuclear receptor that shares significant DNA sequence homology with ERα, although E2 is not considered its endogenous ligand[51]. Single-nucleus transcriptomic analysis[52] revealed that the ESRRA transcript, which encodes ERRα, was expressed at the highest level in neurons (Fig. 1e). This neuronal specificity was further corroborated by the enrichment of PPARGC1A, encoding the ERRα coactivator PGC1α, in the same cell type (Fig. 1e). Despite these findings, the log2-fold change (log2FC) in ESRRA transcript expression between the LOAD and ND groups was minimal ($\leq |0.07|$) but statistically significant ($p < 0.05$) (Supplementary Fig. 2b). These results were confirmed via quantitative PCR analysis (Supplementary Fig. 2a).

To investigate whether the subcellular localisation of ERRα was affected by disease state, immunohistochemistry was performed. This analysis demonstrated reduced nuclear localisation of ERRα in neurons, regardless of its excitatory or inhibitory nature. This reduction was more pronounced in female subjects, particularly at advanced ages (Fig. 1f and Supplementary Fig. 2c, d), and was further exacerbated in those with LOAD (Fig. 1f). These findings suggested that the loss of ERRα nuclear activity might serve as a mechanistic link between the hormonal changes associated with the menopausal state and the female-specific pathological changes observed in LOAD patients.

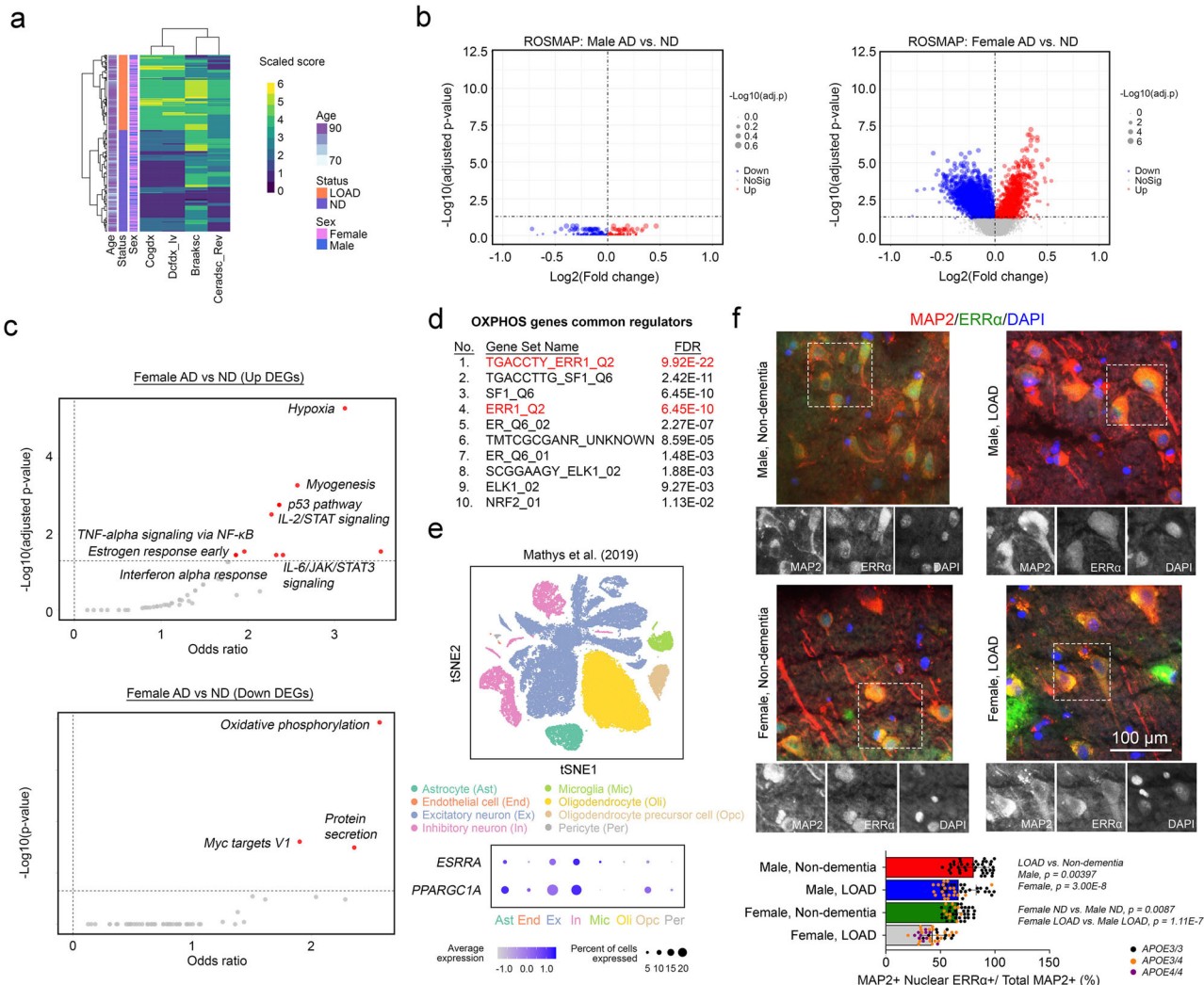

**Fig. 1 | Female patients with LOAD exhibit a more pronounced decline in the ERRα-regulated bioenergetic network within neurons. a** The disease status of the ROSMAP study samples was defined on the basis of multiple clinicopathological parameters (x-axis). **b** DEG profiles curated from the comparison between LOAD and ND samples in a sex-specific manner [males (left panel) and females (right panel)] (two-sided Limma with Benjamini–Hochberg correction). **c** Functional enrichment analysis of the DEGs identified from LOAD vs ND samples in female subjects [upregulated (top panel) and downregulated (bottom panel)] on Enrichr[164] (two-sided Fisher's exact test with correction). **d** Common transcription regulator analysis of DEGs enriched in the OXPHOS pathway on the GSEA platform[165] with reference to the TFT_LEGACY subset of TFT (Mann–Whitney rank sum test with

corrections)[166]. **e** t-SNE plot of a total of 70,634 nuclei derived from a total of 48 LOAD (n = 24) and ND (n = 24), age (mean ± SD = 85.646 ± 4.215), and biological sex-matched (12 males and 12 females per group) prefrontal cortex samples from Brodmann area 10[52]. A dot plot illustrates the relative expression levels of *ESRRA* and *PPARGC1A* among all the brain cell types. **f** Representative immuno-fluorescence staining images of human prefrontal cortex tissues reveal changes in nuclear signals of ERRα in MAP2-positive neurons in LOAD vs ND samples of different sexes. Quantification is shown (n = 4 biological replicates/group; 10 experimental replicates/sample; Kruskal–Wallis test with Dunn's multiple comparisons correction). N represents biological replicates. The values represent the mean ± s.d. Source data are provided as a source data file.

## Correlation between E2:P4 ratios and reduced cognitive and memory capacity at menopausal state onset in a VCD-induced AOF model

The perimenopausal transition in women is marked by elevated, erratic, and unpredictable E2 levels and a sharp reduction in P4 levels[20,21,53]. To investigate how these endocrine changes impact brain function, a perimenopausal state was artificially induced in laboratory mice using 4-vinylcyclohexene diepoxide (VCD), a chemical that selectively destroys small preantral ovarian follicles by accelerating follicular atresia while preserving the overall ovarian anatomy[54,55]. Young adult mice (postnatal day 70–75, P70–75) were utilized to minimize the confounding effects of chronological ageing (Fig. 2a). Following VCD treatment, elongation of the oestrous cycle duration became apparent by the 6th–7th cycle after completion of the treatment (Fig. 2a). This phenomenon was accompanied by

persistent elevations in circulating levels of follicle-stimulating hormone (FSH) (Fig. 2b). While consistent reductions in circulating E2 and P4 levels were observed by Cycle 14 in all VCD-treated animals, a subset of animals at Cycle 7 exhibited marked increases in E2 (Fig. 2c) or decreases in P4 levels (Fig. 2d), resulting in aberrantly high E2:P4 ratios (≥2.5 pg/ng) (Fig. 2e). This condition, termed "higher oestradiol–lower progesterone imbalance," characterized the onset of the perimenopausal state transition, which generally began at Cycle 6–7, while the full menopausal state (characterized by general reductions in both E2 and P4 levels) was successfully induced by Cycle 14.

To assess the functional consequences of these endocrine changes, a series of behavioural tests was performed. VCD-treated animals showed no significant changes in motor function, as assessed by the rotarod test and open-field test (Fig. 2h and Supplementary Fig. 3e–g).

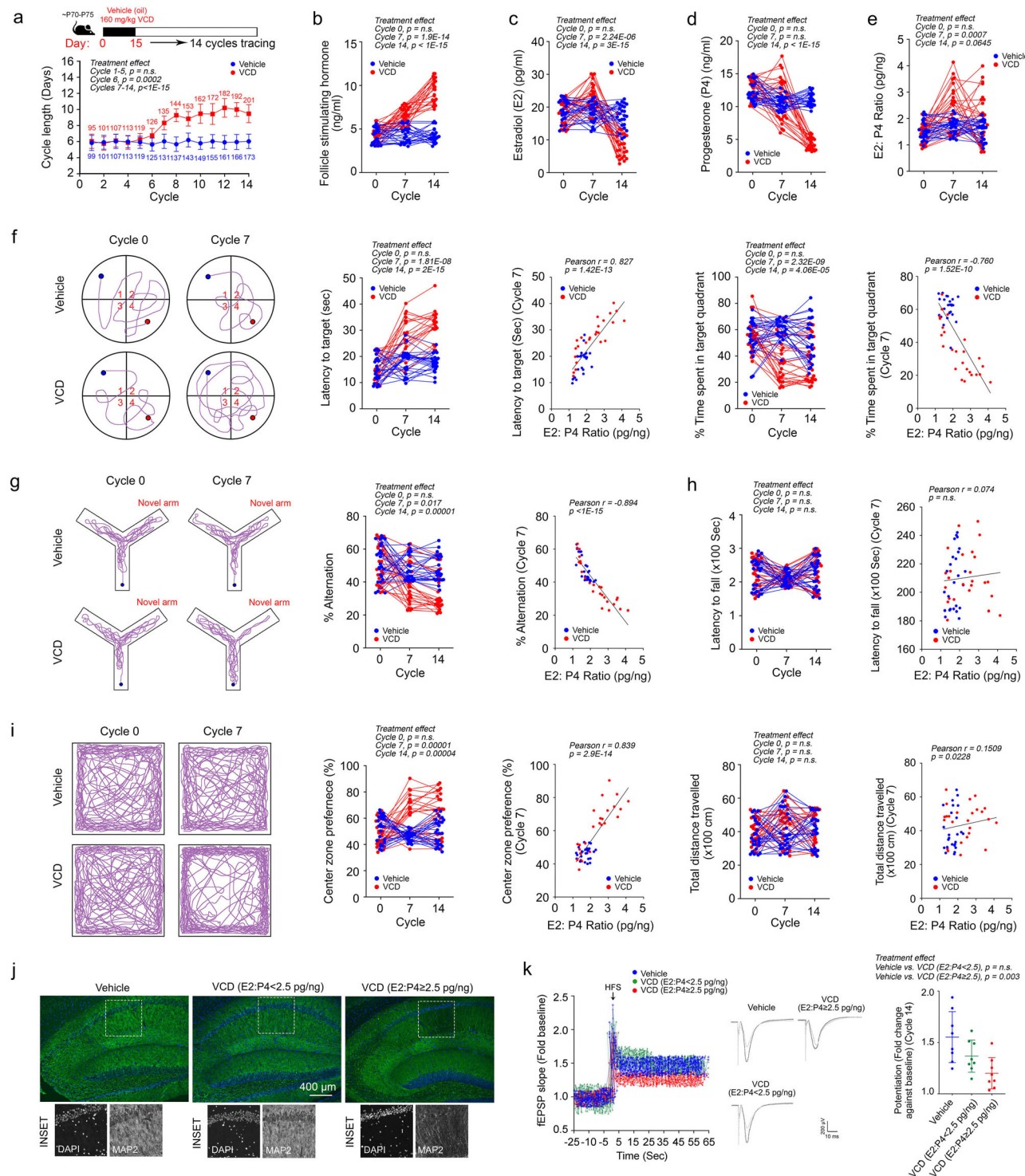

However, significant declines in spatial learning and memory, as evaluated by the Morris water maze (MWM) (Fig. 2f and Supplementary Fig. 3a–c), and in short-term working memory, as assessed by the Y-maze, were observed beginning at Cycle 7 (Fig. 2g and Supplementary Fig. 3d). Furthermore, the animals displayed mild anxiety-like behaviour, as indicated by the significantly reduced time spent in the centre zone during the open-field test (Fig. 2i and Supplementary Fig. 3g). Importantly, the decline in behavioural performance at the onset of the perimenopausal state transition (Cycle 7) was most strongly correlated with the E2:P4 ratio (Figs. 2f–g, i) rather than with the circulating levels of E2 or P4 alone (Supplementary Fig. 3b–d, g). Immunohistochemical analysis further supported the negative impact

of elevated E2:P4 ratios (≥2.5 pg/ng) on brain cellular integrity, revealing significant neurite loss in the Cornu Ammonis area 1 (CA1) of the hippocampus (Fig. 2j and Supplementary Fig. 3h–j). These anatomical changes were accompanied by impairments in neurophysiological function. Field excitatory postsynaptic potentials (fEPSPs) recorded from the Schaffer collateral pathway showed significant deficits in VCD-treated animals (Fig. 2k, left panel), along with diminished long-term potentiation (LTP) (Fig. 2k, right panel). Together, these findings demonstrated that aberrantly elevated E2:P4 ratios during the perimenopausal state transition were associated with cognitive and memory impairments, mild anxiety-like behaviour, neurite loss in the hippocampus, and reduced synaptic plasticity,

**Fig. 2 | Diminished cognitive and memory capacities correlate with the resulting E2:P4 ratios following the induction of AOF by VCD. a** Schematic of VCD administration and the associated timeline. The average length of the oestrous cycle (days) was defined as the number of days between 2 successive demonstrations of proestrus as evaluated by vaginal cytology ($n = 25$; two-way ANOVA with Šídák's multiple comparisons test). **b–e** Changes in circulating levels of **b** FSH, **c** oestradiol (E2), **d** progesterone (P4) and **e** E2:P4 ratio at the proestrus phase of cycles 0, 7, and 14 ($n = 25$, two-way ANOVA with Šídák's multiple comparisons test). **f** Representative swimming patterns of mice in the MWM paradigm. Latency to target and time spent in target quadrant plots of the probe trial test, as well as their correlations with the resulting E2:P4 ratios at cycle 7, are shown ($n = 25$; two-way ANOVA with Šídák's multiple comparisons test; Pearson correlation test). **g** Representative walking patterns of mice in a Y-maze paradigm. The percentage of alternations and the correlations to the resulting E2:P4 ratios after treatment are shown ($n = 25$, two-way ANOVA with Šídák's multiple comparisons test, Pearson

correlation test). **h** Latency to fall statistics in a rotarod paradigm and correlations with the resulting E2:P4 ratios at cycle 7 are shown ($n = 25$; two-way ANOVA with Šídák's multiple comparisons test and Pearson correlation test). **i** Representative walking patterns of mice in an open-field test paradigm. The percentage of time spent in the centre zone and the correlations to the resulting E2:P4 ratios at cycle 7 ($n = 25$, two-way ANOVA with Šídák's multiple comparisons test, Pearson correlation test). **j** Representative immunofluorescence staining images of mouse hippocampal tissue harvested after cycle 14 ($n = 5$ biological samples/batch, 5 batches). **k** Field excitatory postsynaptic potentials (fEPSPs) evoked by Schaffer collateral pathway stimulation during the LTP experiment in acute hippocampal slices ($n = 8$). Representative traces displaying electrophysiological recordings from acute hippocampal slices. Quantification of average fEPSPs during the last 10 min ($n = 8$, one-way ANOVA with Tukey's multiple comparisons test). HFS high-frequency stimulation. N represents biological replicates. Values represent the mean ± s.d. Source data are provided as a source data file.

underscoring the detrimental effects of hormonal imbalances on brain function.

## High oestradiol but low progesterone imbalance induces brain transcriptomic changes reflecting impairment of the ERRα signalling network

The behavioural and electrophysiological findings revealed a functional connection between a higher oestradiol-lower progesterone imbalance and brain dysfunction. To further investigate the underlying molecular mechanism, we conducted a bulk transcriptomic analysis of the cerebral cortex from VCD-treated animals, which exhibited a greater oestradiol–lower progesterone imbalance during the perimenopausal state transition (E2:P4 ≥ 2.5 pg/ng in oestrous Cycle 7; Fig. 2e). Principal component analysis (PCA) effectively distinguished the transcriptomic profiles of VCD-treated animals from those of vehicle-treated controls (Fig. 3a). This analysis revealed 346 upregulated and 544 downregulated transcripts in the VCD-treated group (Fig. 3b). While the upregulated DEGs did not cluster into any meaningful pathways, the downregulated DEGs were strongly enriched in pathways associated with mitochondrial energetics, including OXPHOS and the citrate cycle (TCA cycle). Furthermore, these genes were also involved in pathways related to neurodegenerative disorders such as Parkinson's disease, prion disease, Huntington's disease, AD, and amyotrophic lateral sclerosis (Fig. 3c, d). These findings highlighted the central role of defective OXPHOS in the pathogenesis of age-related neurodegenerative diseases. Subsequent transcription factor analysis of the downregulated DEGs revealed ERR1/ERRα as the primary upstream regulator, with at least one highly conserved ERRα binding motif (TGACCTY) predicted within the 4 kb region centred on the transcription start sites of these genes [−2 kb, +2 kb] (Fig. 3e). Consistent with findings from human brain tissues (Fig. 1f), immunocytochemistry demonstrated that ERRα, along with its coactivator PGC1α, was predominantly localized in neuronal nuclei under control conditions. However, with the VCD-induced increase in the oestradiol-lower progesterone imbalance, the nuclear localisation of ERRα and PGC1α was significantly reduced (Fig. 3f and Supplementary Fig. 4). Further immunoblotting analyses revealed a disruption in protein–protein interactions between ERRα and PGC1α, accompanied by a significant reduction in the protein level of PGC1α (Fig. 3g), an intrinsically disordered protein prone to cytoplasmic degradation[56].

Previous studies have shown that the ligand binding domain (LBD) and activation function-2 (AF2) domain of ERRα mediate its interaction with the third LXXLL motif of PGC1α[57]. Furthermore, the binding of cholesterol, a natural ERRα ligand, to the LBD enhances the ERRα-PGC1α interaction[58]. To explore this mechanism, we performed in silico simulations to provide structural insights. Using previously reported but incomplete, open structures of the ERRα LBD (PDB: 7E2E and 2PJL)[59,60], we generated a complete open conformation of the ERRα LBD by reconstructing the missing alpha-helix segment with

AlphaFold2[61] (Supplementary Fig. 5a, b), which allowed us to model the effect of cholesterol ligand binding on the interaction with another AlphaFold2-simulated 3rd LXXLL motif (amino acids 208–216) of PGC1α[61] using the HADDOCK 2.4 algorithm[62]. Docking simulations revealed that cholesterol binding to the open LBD structure induced conformational remodelling near the cholesterol binding pocket of ERRα, which facilitated enhanced hydrophobic interactions and hydrogen bond formation with the third LXXLL motif of PGC1α (Fig. 3h). In contrast, docking simulations with the closed ERRα LBD structure (PDB: 1XB7)[60], which is incompatible with cholesterol binding, failed to reveal similar interactions (Supplementary Fig. 5c). In cerebral cortex tissues from VCD-treated animals, we observed a reduction in ERRα-bound cholesterol levels (Fig. 3i). This reduction likely disrupted ERRα-PGC1α interactions and diminished the downstream transcriptional activity of ERRα, which was consistent with previous findings linking cholesterol binding to enhanced ERRα-PGC1α interactions[58]. Taken together, these results demonstrated that a greater imbalance between oestradiol levels and progesterone levels disrupted the ERRα-PGC1α signalling network by reducing the level of ERRα-bound cholesterol and impairing protein–protein interactions.

## Progesterone-induced ERα signalling regulates the downstream ERRα-PGC1α axis by maintaining cholesterol homoeostasis

The reduction in ERRα-bound cholesterol in the brains of VCD-induced AOF mice with a higher oestradiol-lower progesterone imbalance suggested a broad metabolic reprogramming effect, including local disruptions in brain progesterone and ER signalling. Unbiased metabolomics (Fig. 4a, b) and targeted KEGG metabolic gene expression analysis (Fig. 4c) revealed significant impairments in cholesterol biosynthesis, accompanied by disruptions in central carbon metabolism through glycolysis (Fig. 4d). These metabolic changes were neuron-specific, as demonstrated by cell-specific metabolomic analyses (Supplementary Fig. 6).

In neurons, glucose serves as a primary energy source and a precursor for cholesterol biosynthesis. Glucose enters glycolysis, where it is converted to acetyl-CoA in mitochondria[63]. Acetyl-CoA can then combine with oxaloacetate to form citrate, which is exported to the cytoplasm and converted back to acetyl-CoA by citrate lyase, feeding the mevalonate pathway for cholesterol biosynthesis (Fig. 4d). The two main mechanisms underlying the supply of cholesterol to neurons are de novo cholesterol biosynthesis and lipoprotein-mediated transfer from glial cells[64]. To determine whether specific ER signalling contributed to the observed metabolic changes, we reevaluated dysregulated metabolites and genes in a primary neuronal culture model. Neurons were exposed to oestradiol cypionate (E2, 100 nM) alone or in combination with an ERα antagonist (MPP dihydrochloride, 100 nM), an ERβ agonist (PTHPP, 100 nM), or a GPER agonist (G-15, 100 nM) for 120 h (Fig. 4e). The results demonstrated that ERα primarily mediated the effects of E2 on neurons (Fig. 4e).

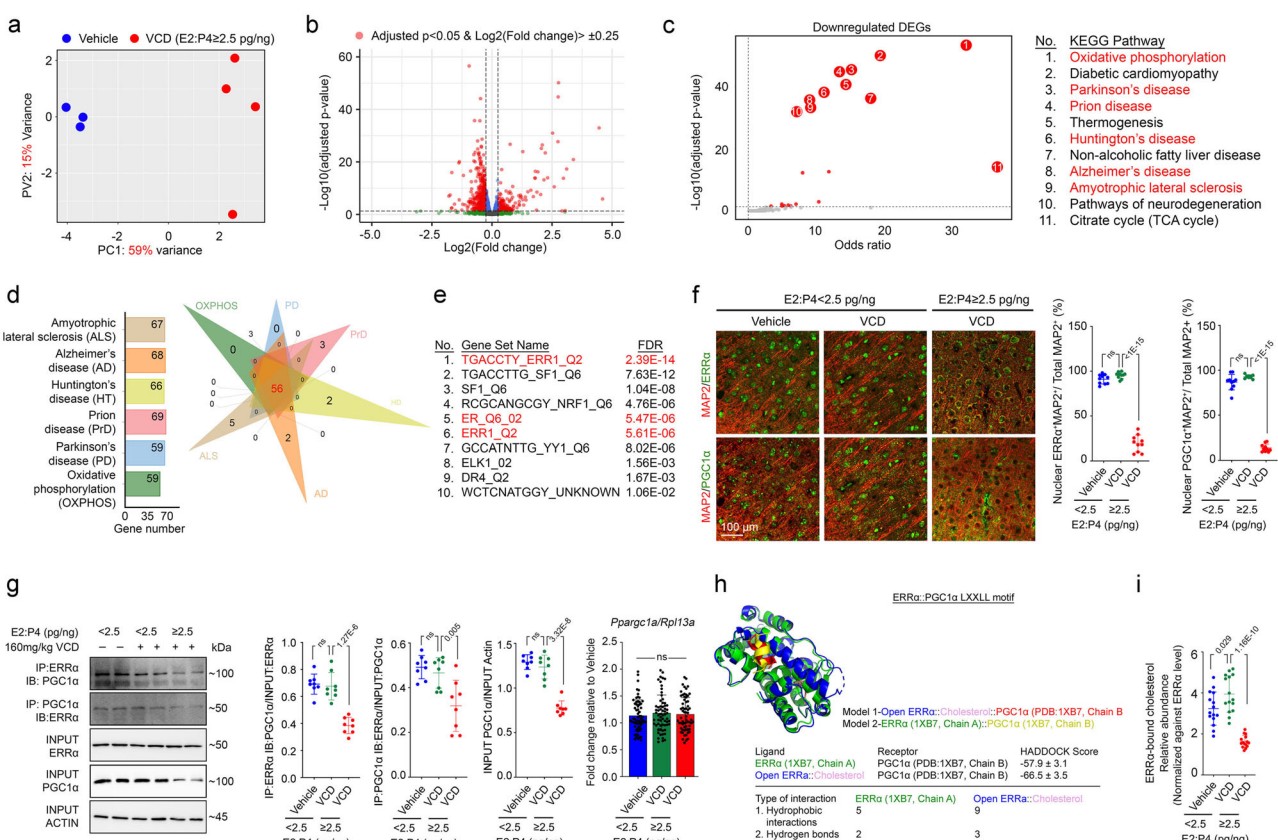

**Fig. 3 | Brain transcriptomic changes induced by the perimenopausal state are related to impairment of the ERRα signalling network. a** PCA indicating that samples from the vehicle ($n = 3$) and VCD (E2:P4 ≥ 2.5 pg/ng) ($n = 4$) groups were distinctly clustered. **b** Volcano plot indicating that a total of 346 up- and 544 downregulated DEGs were found (adjusted $p$-value < 0.05; log2(fold change) > | 0.25|) (two-sided Wald test with Benjamini–Hochberg correction). **c** Functional enrichment analysis of all downregulated DEGs found in the VCD (E2:P4 ≥ 2.5 pg/ng) group on Enrichr[164] (two-sided Fisher's exact test with correction). **d** Venn diagram revealing the number of genes enriched in OXPHOS and neurodegenerative disease pathways that overlapped. **e** Common transcription regulator analysis of all downregulated DEGs found in (**b**) on the GSEA platform[165] with reference to the TFT_LEGACY subset (Mann–Whitney rank sum test with corrections)[166]. **f** Representative immunofluorescence staining images of mouse brain prefrontal cortex tissue reveal changes in ERRα and PGC1α nuclear signals ($n = 10$; one-way ANOVA with Šídák's multiple comparisons test). **g** Representative immunoblots

reveal how treatment affects the protein–protein interaction between ERRα and PGC1α and their total levels in prefrontal cortex tissues ($n = 8$; one-way ANOVA with Šídák's multiple comparisons test). On the far right are the results of quantitative PCR analysis of *Ppargc1a/Rpl13a* in the same set of samples ($n = 8$, 8 technical repeats; one-way ANOVA with Tukey's multiple comparisons test). **h** Predicted molecular models of a published version of a closed ligand binding domain (LBD) of ERRα (i.e. green–PDB: 1XB7, chain A) or an AlphaFold2-simulated version of the same ERRα LBD, except that it was in an open configuration (i.e. blue–open ERRα) and bound to its natural ligand cholesterol (i.e. pink). These two versions of the ERRα LBD were docked against the LXLLL motif of PGC1α (PDB: 1XB7) using the HADDOCK 2.4 algorithm[62]. The corresponding HADDOCK scores and numbers and types of interactions are shown. **i** Quantitative measurements of bound cholesterol in ERRα coimmunoprecipitated from the brain samples ($n = 15$; one-way ANOVA with Šídák's multiple comparisons test). N represents biological replicates. The values represent the mean ± s.d. Source data are provided as a source data file.

Furthermore, on the basis of initial predictions made with the ENCODE, CHEA and ChIP-Atlas databases (Supplementary Fig. 7a), chromatin immunoprecipitation (ChIP) analysis confirmed that five dysregulated metabolic genes identified in the VCD-induced AOF model were bona fide ERRα targets, including *Pdha1* (encoding a component of the pyruvate dehydrogenase complex), *Dhcr24*, and *Cyp51* (both involved in cholesterol biosynthesis) (Fig. 4e). In addition to de novo cholesterol biosynthesis, neurons can acquire cholesterol from apolipoprotein E (ApoE) released by astrocytes[65]. Genes encoding ApoE receptors, such as *Lrp1* and *Vldlr*, were also ERRα sensitive (Fig. 4e). ChIP–PCR revealed robust E2-induced ERRα binding to the promoter regions of these receptor genes, which was blocked by the ERRα antagonist MPP dihydrochloride (Fig. 4f and Supplementary Fig. 7a–c)[66]. In support of neuronal specificity, mouse SMART-seq data from the Allen Brain Atlas confirmed the enrichment of these genes in neurons (Supplementary Fig. 8). In the ROSMAP brain transcriptomics dataset, the expression of these genes (except for *LRP1*) was selectively reduced in female patients and correlated with cognitive impairment (Fig. 4g and Supplementary Fig. 9).

These findings suggest that ERα regulates a cholesterol biosynthetic axis in mature neurons. However, the global loss of cholesterol homeostasis under increased oestradiol-lower progesterone imbalance, as revealed by transcriptomic and metabolomic analysis, implies that ERα signalling is influenced by P4 and its receptor PR. Reanalysis of ChIP-sequencing data from the hormone-sensitive breast cancer cell line MCF7 (GSE68359)[13] revealed that cotreatment with E2 and natural P4, but not E2 alone or with synthetic P4 (R5020), induced the co-occupancy of ERα, PR, and p300 histone acetyltransferase at the promoters of cholesterol biosynthesis genes (Supplementary Fig. 7d). In a routine neuronal culture system, natural P4 is present in the B27 supplement[67], which may have enabled E2 alone to induce cholesterol biosynthesis (Fig. 4e). To address the role of P4, neurons were cotreated with mifepristone (MIF), a PR antagonist. MIF abolished the stimulatory effects of E2 on cholesterol homeostasis (Fig. 4h–i). Conversely, supplementation with additional P4 further enhanced the E2-induced upregulation of cholesterol biosynthesis-related gene expression (Fig. 4h–i and Supplementary Fig. 7e). In addition, stable isotope tracing analysis

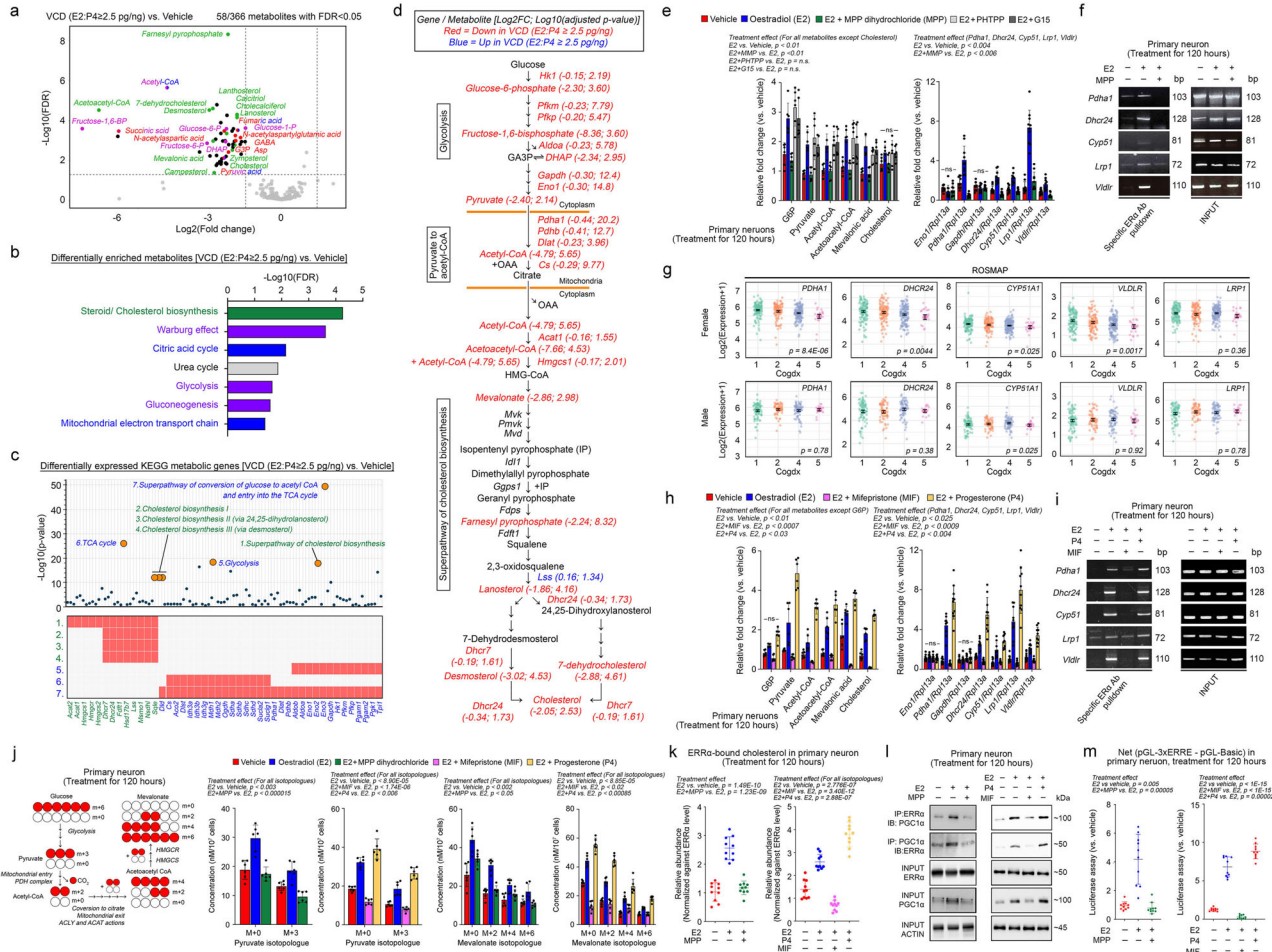

**Fig. 4 | ERα signalling regulates the downstream ERRα-PGC1α axis by maintaining cholesterol homoeostasis. a** Volcano plot summarising the trend of changes in significantly altered metabolites (n = 6; two-sided Welch multiple unpaired *t*-test with Benjamini, Krieger, and Yekutieli correction). **b** Metabolite set enrichment analysis was conducted. Colour codes of the metabolites in (**a**). are matched to the corresponding pathways (global test with Bonferroni correction). **c** Pathway enrichment analysis of differentially expressed KEGG metabolic genes extracted from the transcriptome dataset presented in Fig. 3a, b. (Vehicle: n = 3; VCD: n = 4) performed on Enrichr[164] (two-sided Fisher's exact test with correction). **d** Diagram illustrating the link between glycolysis and cholesterol metabolism. The gene expression of the enzymes in (**c**) and the metabolites in (**a**) are shown. **e, h** Targeted analysis of key metabolites (n = 6) and qPCR analysis (n = 10) of key dysregulated genes involved in neuronal cholesterol metabolism (one-way ANOVA with Tukey's multiple comparisons test except. **e** *Lrp1* and Mevalonic acid; **h** G6P and *Gapdh* where the Kruskal–Wallis test with Dunn's multiple comparisons test was used). **f, i** ChIP–PCR analysis with an ERα-specific antibody. Quantifications in

Supplementary Fig. 7c. **g** Scattered plots of the expression levels of ERα-targeted metabolic genes in ROSMAP brain samples against Cogdx scores (ND: male, n = 133; female, n = 219; LOAD: male, n = 88; female, n = 173; Kruskal–Wallis test).
**j** Schematic representation of glucose fate for the biosynthesis of mevalonate. Mass isotopologue analysis of pyruvate and mevalonate (n = 6; one-way ANOVA with Šídák's multiple comparisons test). **k** Quantitative measurements of cholesterol in ERRα coimmunoprecipitated (n = 10; one-way ANOVA with Šídák's multiple comparisons test for the left panel, while Tukey's multiple comparisons test was used for the right panel). **l** Representative immunoblots reveal that treatment affects the protein–protein interaction between ERRα and PGC1α, and their total levels. Quantifications in Supplementary Fig. 10. **m** Luciferase reporter assay for ERRα nuclear activity (i.e. pGL-3xERRE), normalised against those expressed pGL-Basic (n = 10, Kruskal–Wallis test with Dunn's multiple comparisons test for the left panel; one-way ANOVA with Tukey's multiple comparisons test for the right panel). N represents biological replicates. The values represent the mean ± s.d. Source data are provided as a Source Data file.

using glucose-$^{13}C_6$ further validated these findings (Fig. 4j). Isotopologue profiling revealed that P4 facilitated E2-induced robust glucose-derived pyruvate flux into the mevalonate pathway for cholesterol biosynthesis. This effect was blocked by MIF or MPP dihydrochloride, confirming the role of P4-guided ERα signalling in cholesterol metabolism (Fig. 4j). Downstream of these events, the relative abundance of ERRα-bound cholesterol (Fig. 4k), protein–protein interactions between ERRα and PGC1α (Fig. 4l and Supplementary Fig. 10), and ERRα nuclear transcriptional activity (Fig. 4m) were regulated in a similar manner. Together, these findings demonstrated that progesterone-guided ERα signalling sustained cholesterol homeostasis, which is essential for the function of the ERRα-PGC1α axis.

## Loss of downstream ERRα dysregulates the neuronal NAAG metabolic axis

ERRα is a key regulator of cellular energy metabolism[68]; however, the detailed metabolic reprogramming events and their effects on neuronal resistance to energetic stress following ERRα loss remain unclear. To investigate this interaction, an adeno-associated virus (AAV) expressing a microRNA-30 (miR30)-based short hairpin RNA (shRNA) under the control of a human synapsin promoter was injected into the prefrontal cortex of C57BL/6 mice to specifically knock down neuronal *Esrra*[69] (Fig. 5a, b and Supplementary Fig. 11a). Validation of the knockdown results revealed reduced *Esrra* expression at both the mRNA and protein levels, along with significant neurite loss in regions where the shRNA was robustly expressed (Fig. 5b and Supplementary

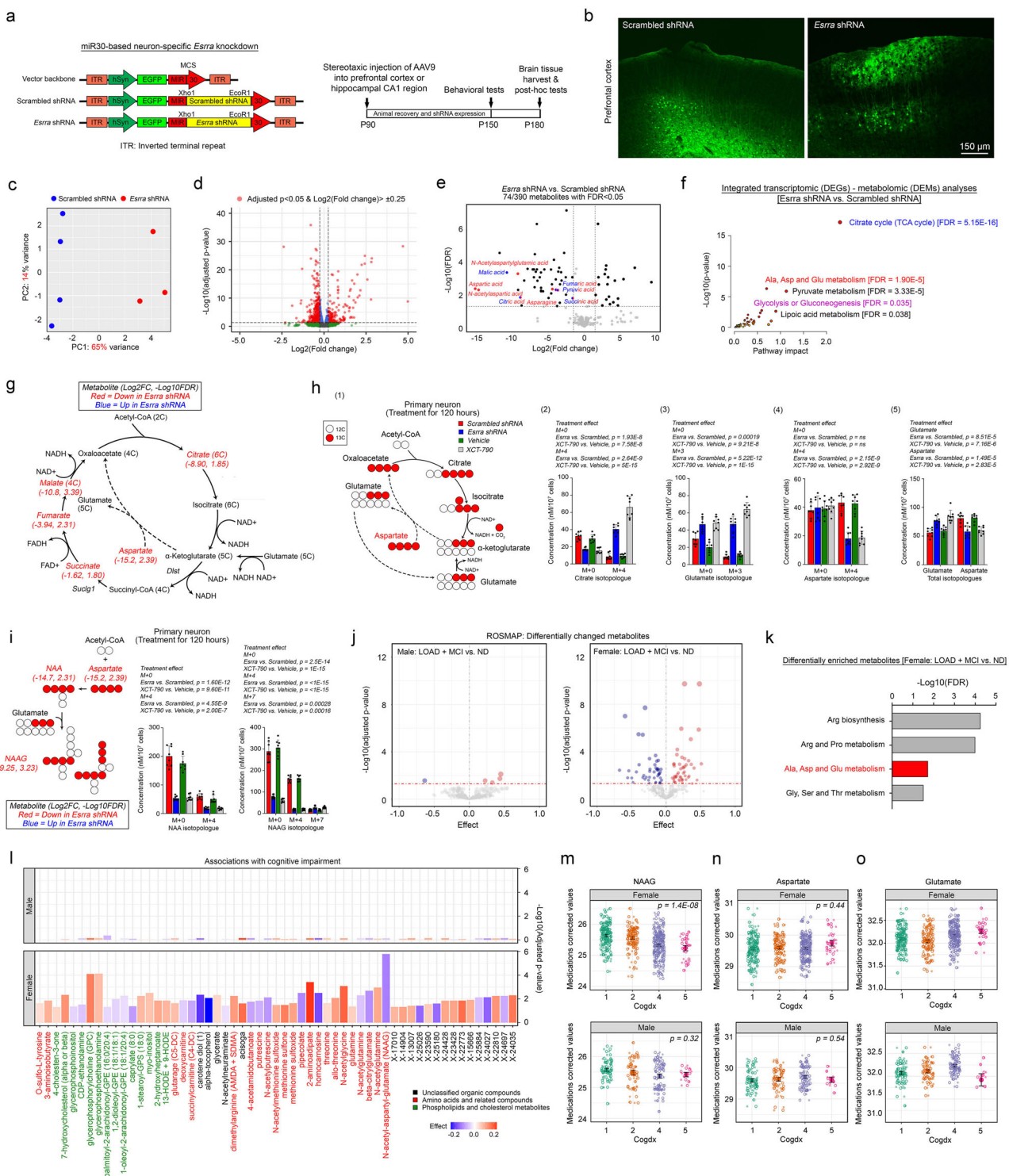

Fig. 11a–d). Behaviourally, this knockdown led to mild impairments in memory and cognitive function (Supplementary Fig. 11e–h).

Transcriptomic and metabolomic profiling of microdissected GFP-positive neurons revealed that the TCA cycle was the most affected pathway, followed by "alanine, aspartate, and glutamate metabolism" (Fig. 5c–f and Supplementary Fig. 12). Specific reductions in the levels of TCA cycle intermediates, including succinate, fumarate, and malate, were observed (Fig. 5g). These findings suggested that a rewiring of metabolic pathways might compensate for the truncated TCA cycle. In particular, aspartate aminotransferase might mediate the transfer of amino groups from aspartate to α-ketoglutarate, allowing the latter to re-enter the TCA cycle via glutamate dehydrogenase. Consistently, a significant reduction in aspartate levels was observed (Fig. 5g). To validate this metabolic reprogramming, isotopologue tracing with labelled $^{13}C_4$-aspartate was performed in primary neuronal cultures. The analysis confirmed that neurons lacking functional ERRα engaged a "mini" TCA cycle, where aspartate contributed carbons to α-ketoglutarate, allowing downstream regeneration of citrate and glutamate [Fig. 5h (1)]. This rewiring was observed in neurons with *Esrra* knockdown via shRNA or those treated with the ERRα inverse agonist XCT-790 (400 nM)[70]. Elevated levels of M + 4 citrate, M + 3 glutamate, and corresponding isotopologues were observed, indicating a reliance

**Fig. 5 | Loss of ERRα dysregulates the neuronal NAAG metabolic axis.**
**a** Schematic of the in vivo experimental flow and design of AAV-shRNA constructs.
**b** Representative images of GTP-labelled neurons in the prefrontal cortex ($n = 5$).
**c** PCA indicating that samples from the scrambled shRNA ($n = 3$) and *Esrra* shRNA ($n = 4$)-treated groups were distinctly clustered. **d** Volcano plot revealing 214 upregulated and 379 downregulated DEGs were found (adjusted $p$-value $< 0.05$; log2(fold-change)$> |0.25|$) (two-sided Wald test with Benjamini−Hochberg correction). **e** Volcano plot summarising the pattern of changes in 74 significantly changed metabolites in brain tissues ($n = 6$; two-sided Welch multiple unpaired $t$-test with Benjamini, Krieger, and Yekutieli correction). **f** Integrated transcriptomics and metabolomics analysis of significantly changed DEGs identified from (**d**) and metabolites from (**e**) in the *Esrra* shRNA-treated group conducted on MetaboAnalyst[167]. The colour codes of the metabolites in (**e**) are matched to the corresponding pathways (Globaltest with Bonferroni correction[168]). **g** Diagram illustrating TCA cycle metabolites with reference to the data presented in (**e**). **h** (1) Schematic representation of aspartate fate in the TCA cycle. (2–4) Mass isotopologue analysis of citrate, glutamate and aspartate. (5) Total glutamate and

aspartate levels ($n = 8$; one-way ANOVA with Holm−Šídák's multiple comparisons test). **i** Schematic representation of the role of aspartate in NAAG biosynthesis. Mass isotopologue analysis of NAA and NAAG ($n = 8$; one-way ANOVA with Holm−Šídák's multiple comparisons test). **j** Volcano plots illustrating differentially altered metabolites curated from the comparison between LOAD + MCI (i.e. disease-affected; male ($n = 62$); female ($n = 161$)) vs ND (male ($n = 86$); female ($n = 191$)) samples in a biological sex-specific manner from the ROSMAP study (Limma with Benjamini and Hochberg corrections). **k** Differentially changed metabolites were clustered by MetaboAnalyst[167] (global test with Bonferroni correction[168]). **l** Bar plots reveal changes in metabolites that are significantly associated with cognitive impairment (Gaussian linear regression corrected with Benjamini−Hochberg). **m−o** Scatter plots reveal the levels of selected metabolites in ROSMAP brain samples with different cognitive scores (Cogdx) [i.e. LOAD: male ($n = 62$); female ($n = 161$); ND: male ($n = 86$); female ($n = 191$); Kruskal−Wallis test]. N represents biological replicates. The values represent the mean ± s.d. Source data are provided as a source data file.

on aspartate to sustain this minicycle [Fig. 5h (2–3)]. Interestingly, while synthetic M + 4 aspartate levels decreased as expected, the total levels of unlabelled M + 0 aspartate and glutamate increased [Fig. 5h (4–5)], suggesting the activation of a compensatory mechanism to replenish these amino acids.

Unbiased metabolomics suggested that the breakdown and/or reduced synthesis of NAAG could serve as a source for replenishing aspartate and glutamate. NAAG, a neuron-specific dipeptide synthesised from N-acetyl-aspartate (NAA) and glutamate[71], was significantly reduced in ERRα-deficient neurons. Consistently, $^{13}C_4$-aspartate isotopologue tracing experiments revealed that the contributions of both aspartate and glutamate carbons to the formation of NAA and NAAG were greatly reduced in neurons deficient in functional ERRα (Fig. 5i), regardless of their source (i.e. endogenous [M + 0 glutamate and M + 0 aspartate] or exogenous [M + 3 glutamate and M + 4 aspartate]−Fig. 5h). While the diminished levels of preexisting unlabelled forms of NAA and NAAG also supported a procatabolic shift, the relatively lower levels of these newly synthesized heavily labelled neuropeptides also indicated diminished anabolic synthesis (Fig. 5i). Such a procatabolic shift likely replenished unlabelled aspartate and glutamate pools, as reflected in isotopologue studies [Fig. 5h (3–4)]. In support of these findings, the VCD-induced AOF mouse model, characterised by diminished ERRα signalling, also displayed reduced succinate, fumarate, NAA, and NAAG levels (Supplementary Fig. 13).

Clinically, higher NAAG levels are associated with better outcomes in patients with neurological conditions such as psychosis[72], schizophrenia[73] and multiple sclerosis[74]. Analysis of human brain samples from the dorsolateral prefrontal cortex of the ROSMAP cohort ($n = 339$ for AD + MCI, $n = 153$ for no dementia) revealed biological sex-specific differences in metabolite levels[75]. Compared with males, female subjects exhibited more dramatic changes in amino acid, phospholipid, and cholesterol-related metabolites, even when sample sizes were adjusted via permutation analysis (Fig. 5j and Supplementary Fig. 14). Metabolite set enrichment analysis revealed that most of the differentially altered metabolites were involved in amino acid metabolism, including aspartate and glutamate metabolism (Fig. 5k). Notably, NAAG levels were significantly correlated with cognitive decline in females but not in males (Fig. 5l, m). These findings underscore the potential role of NAAG metabolism in driving female-biased vulnerability to LOAD. Compared with neuronal changes observed in vitro, changes in total brain glutamate and aspartate levels, which are derived from multiple cell types, were less consistent. However, glutamate levels were elevated and positively correlated with cognitive decline, particularly in females (Fig. 5n, o). These trends were more pronounced and consistent in females across different cognitive scores.

## Loss of ERRα leads to increased spontaneous postsynaptic activity, bioenergetic incompetence, and heightened neuronal vulnerability to excitotoxic insults

Previous studies have demonstrated that NAAG has neuroprotective effects against NMDA receptor (NMDAR)-mediated excitotoxicity by acting as a partial NMDAR antagonist[76]. However, the loss of ERRα activity appears to drive NAAG hydrolysis, providing the aspartate required for metabolic compensation while increasing free glutamate levels−the primary excitatory neurotransmitter in neurons. This metabolic alteration may deplete the inhibitory effects of NAAG on NMDARs while simultaneously amplifying the excitatory activity of the neuronal network. To investigate how the loss of ERRα affects neuronal electrophysiology, whole-cell patch-clamp recordings were performed on CA1 pyramidal neurons in hippocampal slice cultures (Fig. 6a). Miniature excitatory postsynaptic currents (mEPSCs) were recorded in the presence of bicuculline (a GABA_A receptor antagonist) and tetrodotoxin (TTX, a sodium channel blocker) to inhibit inhibitory neurotransmission and action potentials, respectively. Neurons with *Esrra* knockdown exhibited significantly increased mEPSC frequencies compared with scrambled shRNA controls, whereas mEPSC amplitudes, the resting membrane potential (RMP), and input resistance remained unchanged (Fig. 6b). Conversely, *Esrra* knockdown did not alter miniature inhibitory postsynaptic currents (mIPSCs) in terms of frequency or amplitude (Fig. 6c). These findings suggested that the loss of ERRα selectively enhanced glutamatergic transmission without affecting GABAergic activity in CA1 pyramidal neurons. The elevated mEPSC frequency suggested an increased probability of presynaptic glutamate release. This hypothesis was confirmed by paired-pulse ratio experiments in which evoked excitatory postsynaptic currents (eEPSCs) were recorded in CA1 pyramidal neurons following paired electrical stimulation of Schaffer collaterals. Neurons with *Esrra* knockdown displayed a notable reduction in paired-pulse ratios, indicating an increased probability of presynaptic glutamate release (Fig. 6d). Similar increases in mEPSC frequency were observed in primary neurons treated with the ERRα inverse agonist XCT-790, further supporting a presynaptic mechanism (Fig. 6e, f). Consistent with these findings, the expression of the presynaptic vesicular glutamate transporter VGLUT1 was upregulated in ERRα-deficient neurons, whereas the expression of the postsynaptic marker PSD95 remained unchanged. Moreover, the colocalization of VGLUT1 puncta and PSD95 puncta, which are indicators of mature synapses, was significantly increased (Fig. 6g). These findings suggested that VGLUT1 upregulation enhanced glutamate packaging into synaptic vesicles[77], facilitating spontaneous glutamate release and leading to an increase in mEPSC frequency.

The restoration of membrane potential following mEPSCs is an energy-intensive process[78]. While healthy neurons typically have

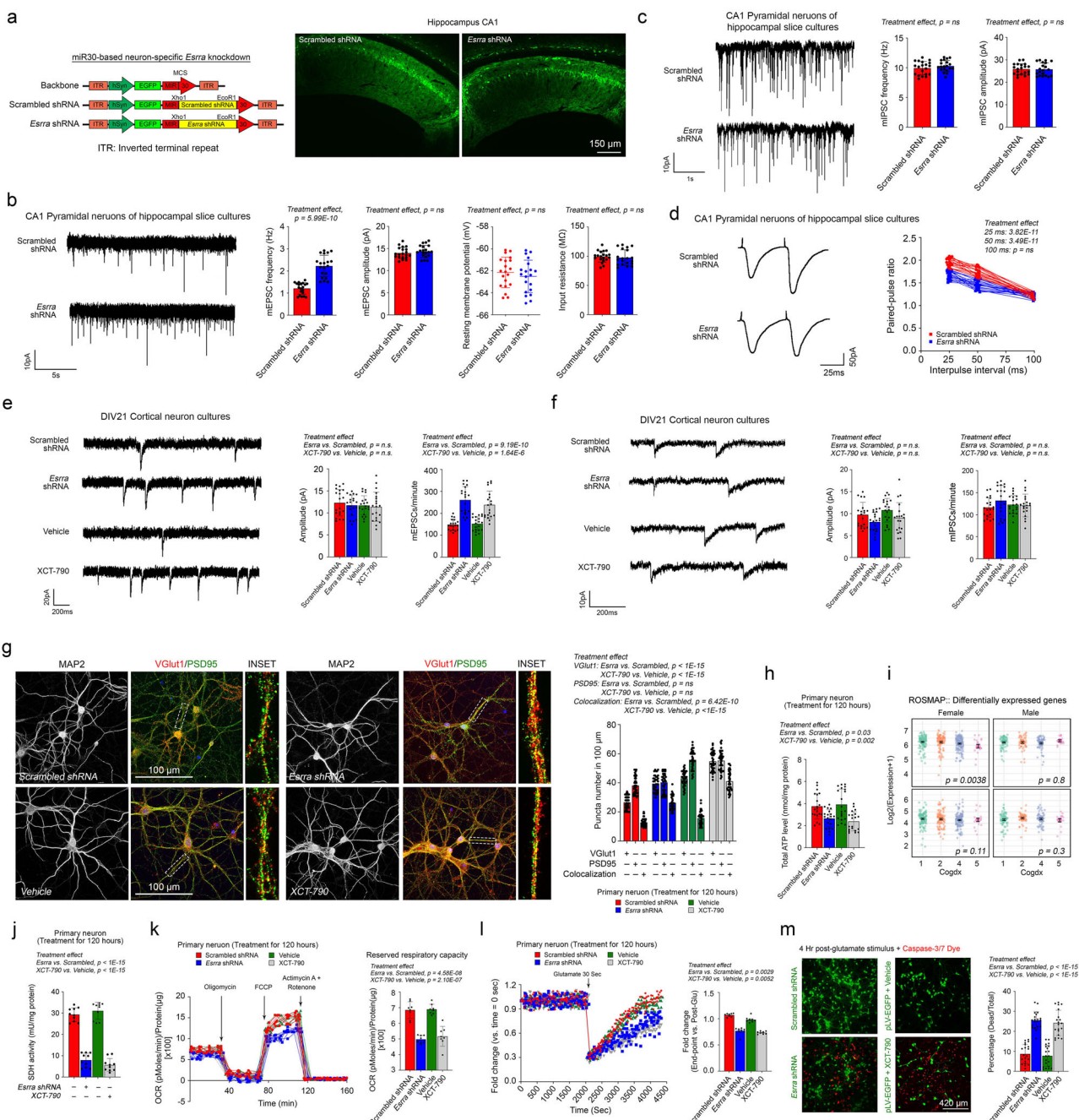

**Fig. 6 | Increased spontaneous postsynaptic activity, coupled with bioenergetic incompetence due to the loss of ERRα, confers heightened neuronal vulnerability to excitotoxic insults. a** Schematic of the AAV-shRNA construct designs. Representative images of GTP-labelled neurons in the hippocampal CA1-2 region (*n* = 5). **b, c** Sample records of (**b**). mEPSCs and **c** mIPSCs in hippocampal slice cultures. Mean frequencies and amplitudes are shown (*n* = 5; 4 experimental replicates; two-tailed unpaired *t*-test). **d** Representative traces of pair-pulse stimulation. Paired-pulse ratios are plotted against interstimulus intervals (*n* = 5; 4 experimental replicates; two-way ANOVA with Šídák's multiple comparisons test). **e, f** Sample records of **e** mEPSCs and **f** mIPSCs in primary cortical neurons. Mean frequencies and amplitudes during the 1–3 min of monitoring are shown (*n* = 5 biological replicates, 4 experimental replicates; one-way ANOVA with Tukey's multiple comparisons test). **g** Immunocytochemistry analysis on the abundance of VGLUT1 and PSD95 signals (n = 40; one-way ANOVA with Holm–Šídák's multiple comparisons test except that for colocalised VGlut1 + PSD95 + , where the Kruskal–Wallis test with Dunn's multiple comparison test was used). **h** Total baseline ATP levels in primary cortical neurons (*n* = 20; Kruskal–Wallis test with

Dunn's multiple range test). **i** Scattered plots reveal biological sex-specific comparisons of the expression levels of *SDHA* and *SDHD* in the ROSMAP brain samples of different cognitive scores (Cogdx) (ND: male *n* = 133 and female *n* = 219; LOAD: male *n* = 88 and female *n* = 173; Kruskal–Wallis test). **j** Succinate dehydrogenase enzymatic activity in primary cortical neurons (*n* = 10; one-way ANOVA with Holm–Šídák's multiple comparison test). **k** Mito Stress assay in primary cortical neurons. Reserved respiratory capacity was calculated (*n* = 8; one-way ANOVA with Tukey's multiple comparisons test). **l** Live monitoring of dynamic ATP signals in response to temporal synaptic activity-dependent energy consumption. The relative abundance of ATP signals after recovery (Signal ratio: Endpoint$_{4270s}$/Post-Glu$_{135s}$) was calculated (*n* = 8, Kruskal–Wallis test with Dunn's multiple comparisons test). **m** Representative images of the AAV-GFP signals and caspase-3/7 signals of primary cortical neurons at 4 h post-glutamate challenge are shown in (**h**). (*n* = 20, one-way ANOVA with Tukey's multiple comparisons test). N represents biological replicates. The values represent the mean ± s.d. Source data are provided as a source data file.

sufficient reserve respiratory capacity to meet increased metabolic demands[79], ERRα-deficient neurons displayed reduced ATP levels at rest, making them particularly vulnerable to energy crises during heightened activity (Fig. 6h). Earlier findings from TCA cycle analyses (Fig. 5g, h) indicated that the activity of succinate dehydrogenase (SDH), a key enzyme that links the TCA cycle to the electron transport chain (ETC), was reduced in ERRα-deficient neurons. Transcriptomic data further revealed decreased expression of *Sdha* and *Sdhd*, which encode key SDH subunits, in both *Esrra*-knockdown and XCT-790-treated neurons (Fig. 5c, d and Supplementary Figs. 12c–e and 15b). In agreement with the predictions from the ENCODE and ChIP-atlas (Supplementary Fig. 15a) databases, ChIP–PCR confirmed that these genes were direct ERRα targets (Supplementary Fig. 15c), and enzyme activity assays verified the reduced SDH function in ERRα-deficient neurons (Fig. 6j). These findings gained additional significance through an analysis of the ROSMAP cohort, which revealed a significant correlation between the expression of SDHA−the gene encoding the major catalytic subunit of SDH−and cognitive decline. Notably, this association was observed exclusively in females (Fig. 6i). This biological sex-specific correlation strongly suggested that impaired SDH function might play a critical role in driving the heightened vulnerability of females to LOAD.

The dual role of SDH as a TCA cycle enzyme and Complex II of the ETC makes it a critical determinant of mitochondrial reserve respiratory capacity−a key determinant of the ATP production rate in response to a sudden surge in energy demand[80]. Measurements of the mitochondrial oxygen consumption rate (OCR) revealed that both basal respiration and reserve respiratory capacity were significantly impaired in ERRα-deficient neurons (Fig. 6k and Supplementary Fig. 15d). This bioenergetic inefficiency, combined with enhanced spontaneous postsynaptic activity, likely accounted for the reduced ATP levels observed at baseline (Fig. 6h). To test the ability of the neurons to recover from bioenergetic stress, we challenged them with glutamate to induce transient sodium and potassium fluxes and stimulate the Na$^+$/K$^+$-ATPase pump, a major consumer of ATP[81]. Real-time mitochondrial ATP measurements using ATP-Red dye revealed that while the effects of glutamate on ATP depletion were similar across all groups during the initial 30-second challenge, the recovery of ATP levels was significantly slower in ERRα-deficient neurons than in control neurons after 30 min (Fig. 6l). Four hours after the glutamate challenge, approximately 16% of the ERRα-deficient neurons underwent apoptosis, as evidenced by caspase-3/7 activation (Fig. 6m). Taken together, these findings revealed that the loss of ERRα led to increased glutamatergic transmission, increased spontaneous postsynaptic activity, and impaired bioenergetic capacity. The reduced mitochondrial reserve capacity and inefficient ATP recovery rendered ERRα-deficient neurons highly vulnerable to excitotoxic challenges, contributing to their heightened susceptibility to bioenergetic crises and excitotoxic insults. These mechanisms might underlie the neuronal dysfunction associated with ERRα deficiency and its potential role in neurodegenerative diseases.

## P4 supplementation during perimenopausal state transition restores cholesterol-bioenergetic homoeostasis and neuronal resilience against excitotoxicity in a mouse model of AD

Our findings suggested that hormonal imbalances during the perimenopausal state, characterised by elevated E2 and reduced P4 levels, disrupted P4-regulated ERα signalling and its downstream metabolic pathways, thereby increasing neuronal vulnerability to excitotoxicity. To investigate how this imbalance contributed to female susceptibility to AD, we applied the VCD-induced AOF paradigm to the 3xTg-AD mouse model at postnatal day 30 (P30), an age at which AD-related pathology is not yet apparent[82] (Fig. 7a, left panel). Similar to findings in C57BL/6 mice (Fig. 2), VCD-induced AOF treatment in 3xTg-AD mice led to increased plasma FSH levels (Supplementary Fig. 16a) and

prolonged oestrous cycles beginning around the 6th–7th cycle post-treatment (Fig. 7a, right panel). These changes were absent in vehicle-treated 3xTg mice, confirming that the observed effects were not influenced by the three transgenes (Fig. 7a and Supplementary Fig. 16a).

At Cycle 14, consistent reductions in E2 and P4 levels were observed in all VCD-induced AOF 3xTg mice. However, around Cycles 6–7, some mice exhibited a pronounced increase in E2 or a decrease in P4 levels, resulting in an elevated E2:P4 ratio (≥2.5 pg/ng), indicating a high oestradiol-low progesterone imbalance (Fig. 7b and Supplementary Fig. 16a). This hormonal imbalance was associated with impaired spatial learning (Fig. 7c and Supplementary Fig. 16b, d) and short-term working memory (Fig. 7d and Supplementary Fig. 16c). Transcriptomic analysis of the cerebral cortex from mice with an E2:P4 concentration ≥ 2.5 pg/ng revealed significant downregulation of genes involved in cholesterol homoeostasis, oestrogen responses, and glycolysis (Fig. 7e–g), which was consistent with findings in C57BL/6 mice (Figs. 3 and 4 and Supplementary Fig. 6). These changes were accompanied by a reduction in the expression of cholesterol-bound ERRα during Cycle 7 (Fig. 7h) and a subsequent decrease in the expression of OXPHOS pathway genes during Cycle 14 (Supplementary Fig. 17a, b).

As previously observed (Fig. 4), activated PR signalling plays a coregulatory role in guiding ERα signalling to regulate the ERRα-PGC1α axis via cholesterol homoeostasis. The 3xTg mice exhibited a high E2:P4 ratio, and a decreased ERα-PR interaction was accompanied by increased levels of Aβ monomers, dimers, and oligomers, as well as phosphorylated tau (S202/T205), in cortical tissues (Fig. 7i and Supplementary Fig. 17c). These findings suggested that reversing the high E2:P4 ratio during the early perimenopausal state (Cycles 7–8) by elevating P4 levels could mitigate the adverse effects of hormonal imbalance on subcellular signalling, AD-related pathology, and cognitive decline. Pharmacokinetic profiling after P4 administration revealed a short plasma half-life following intraperitoneal (i.p.) injection (4–8 mg/kg), with peak levels occurring at 15 min and a rapid decline thereafter[83] [Supplementary Fig. 18a(2), b(2)]. Although P4 injections had no effect on endogenous E2 levels [Supplementary Fig. 18a(3), b(3)], they effectively reduced the E2:P4 ratio below the defined imbalance threshold (≥2.5 pg/ng) for approximately 8 h [Supplementary Fig. 18a, b(4)]. This time window allowed for the implantation of subcutaneous minipumps, which maintained effective P4 concentrations over an extended period. Using a protocol combining i.p. injection (4 mg/kg) followed by subcutaneous infusion (11.7 ± 0.7 mg/kg over 48 h), we found that P4 treatment normalized plasma E2:P4 ratios (Fig. 7k and Supplementary Fig. 18c) and restored P4 levels in brain tissues (Fig. 7k and Supplementary Fig. 18d). Subsequent extended P4 supplementation over 60 days, which coincided with Cycles 14–15, improved memory and cognitive functions in 3xTg mice with an initial high E2:P4 ratio, whereas no significant effects were observed in vehicle-treated mice with balanced E2:P4 ratios (Fig. 7l and Supplementary Fig. 19a, b). Brain tissues harvested from P4-treated 3xTg mice showed significant improvements in neurite length (Fig. 7m), reduced levels of phosphorylated tau (Fig. 7m, n and Supplementary Fig. 19c), and decreased Aβ immunoreactivity, accompanied by enhanced ERα−PR interactions (Fig. 7n and Supplementary Fig. 19d). Transcriptomic and metabolomic profiling revealed that P4 supplementation restored cholesterol and NAAG biosynthetic networks (Fig. 7o and Supplementary Fig. 20a–g). P4 treatment upregulated the expression of cholesterol biosynthetic genes (*Dhcr7, Dhcr24*) and pyruvate decarboxylation genes (*Pdha1*), which are key targets of P4-regulated ERα signalling (Fig. 7o). This phenomenon re-established the relative abundance of cholesterol-bound ERRα (Fig. 7p, left panel) and preserved the expression of SDH, a critical metabolic enzyme involved in TCA cycle activity (Fig. 7p, right panel). Neurophysiological analyses further demonstrated that P4 supplementation improved the

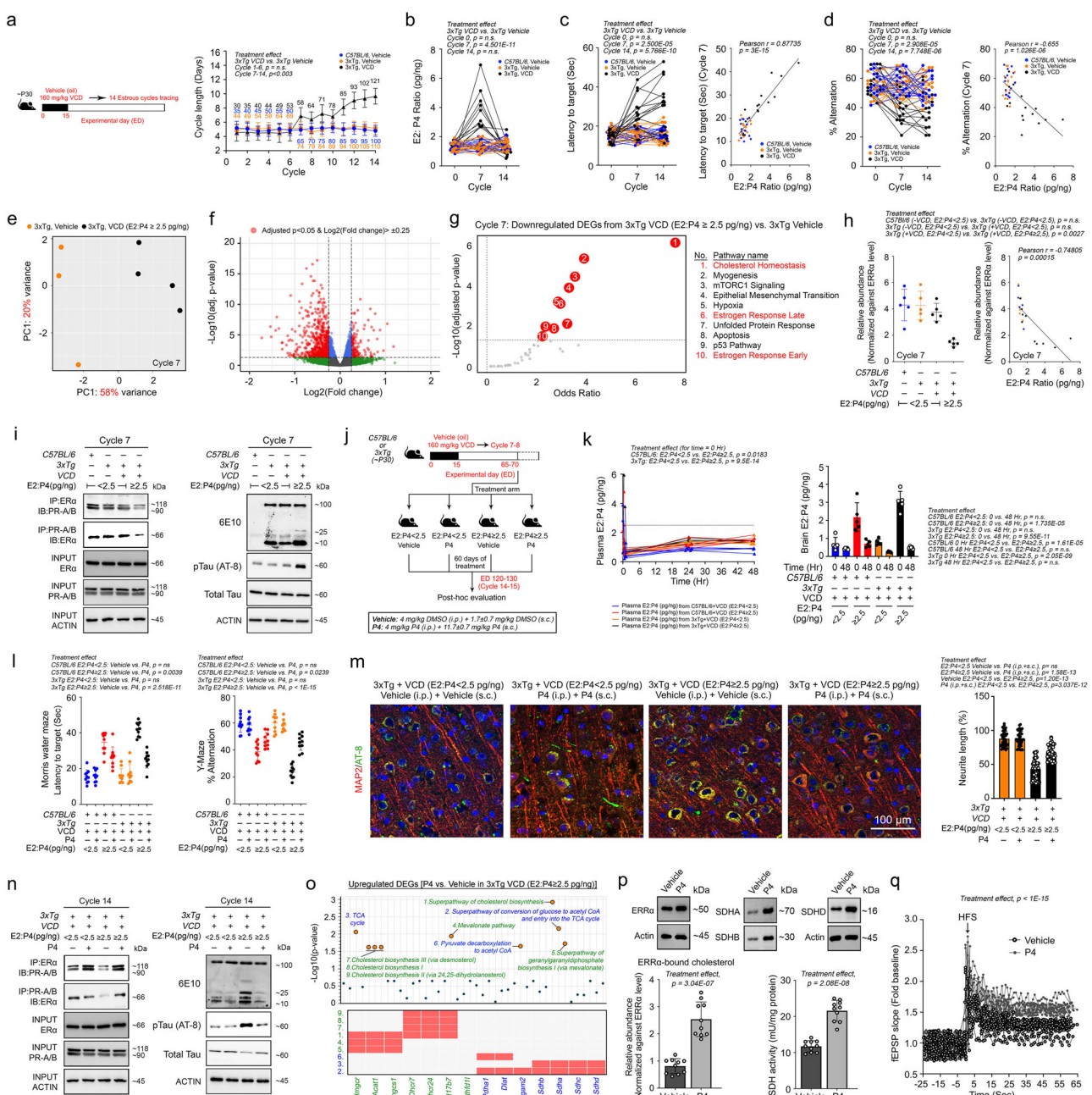

**Fig. 7 | P4 supplementation during the perimenopausal-state transition restores cholesterol-bioenergetic homoeostasis and neuronal resilience against excitotoxicity in a mouse model of AD. a** Schematic of VCD administration procedures in 3xTg mice. Average oestrous cycle length evaluated by vaginal cytology and **b** changes in the circulating E2:P4 ratio at various proestrus phases (*n* = 15; two-way ANOVA with Šídák's multiple comparisons test). **c** Probe trial latency to target in MMW and **d** percentage of alternations in the Y-maze paradigm. Correlations to E2:P4 ratios are shown (*n* = 15; two-way ANOVA with Tukey's multiple comparisons test and Pearson correlation test). **e** PCA of the transcriptomic profiles of the vehicle (*n* = 3) and VCD-treated (*n* = 4) samples. **f** Volcano plot revealing 160 up- and 655 downregulated DEGs (two-sided Wald test with Benjamini−Hochberg correction). **g** Functional enrichment analysis of all downregulated DEGs performed on Enrichr[164] (two-sided Fisher's exact test with correction). **h** Quantitative measurements of ERRα-bound cholesterol levels and their correlations with individual E2:P4 ratios (*n* = 5; one-way ANOVA with the Holm−Šídák multiple comparisons test and Pearson correlation test). **i** Treatment

effects on ERRα, PGC1α, p-Tau and Aβ at Cycle 7 (*n* = 5). **j** Schematics of VCD administration, followed by P4 supplementation in 3xTg. **k** Changes in plasma and brain E2:P4 ratios (n = 5; two-way ANOVA with Šídák's multiple comparisons test). **l** Probe trial latency to the target of the MMW and percentage of alternations in the Y-maze paradigm (*n* = 10; one-way ANOVA with Tukey's multiple comparisons test). **m** Representative micrographs revealing dendrite integrity and p-Tau signals (*n* = 8; 5 technical repeats; one-way ANOVA with Tukey's multiple comparisons test). **n** Effects of P4 treatment on ERRα, PGC1α, p-Tau and Aβ levels at Cycle14 (*n* = 8). **o** Pathway enrichment analysis of upregulated DEGs presented in Supplementary Fig. 20a, b on Enrichr[164] (two-tailed Fisher's exact test with correction). **p** Left: relative abundance of ERRα-bound cholesterol (*n* = 10; two-tailed unpaired *t*-test). Right: Total protein levels and activities of SDH (*n* = 10; two-tailed unpaired *t*-test). **q** fEPSPs during the LTP experiment in acute hippocampal slices (Vehicle: *n* = 8; P4: *n* = 7; two-way ANOVA). N represents biological replicates. The values represent the mean ± s.d. Source data are provided as a Source Data file.

maximal respiratory capacity and ATP-linked respiration in the brain (Supplementary Fig. 20h). Additionally, fEPSPs in the Schaffer collateral pathway were significantly enhanced (Fig. 7q), indicating restored synaptic function.

## Discussion

The precise mechanisms underlying biological sex biases in brain diseases remain poorly understood but are likely driven by a complex interplay between the endocrine system and nervous system. Our findings shed light on how age-related changes in the relative production of E2 and P4, originating peripherally, directly influence female brain physiology and function. Notably, our study reveals a critical regulatory role of P4-guided ERα signalling in maintaining cholesterol homeostasis in neurons—a relationship previously suggested in cancer biology[84–86]. Previous studies have reported that the cholesterol biosynthesis pathway is upregulated in oestrogen-sensitive cancer cell lines, contributing to cell growth and apoptosis. Here, we extend these findings to neurons, demonstrating that P4-activated PR modulates ERα promoter occupancy, which is essential for the regulation of cholesterol homeostasis. This mechanism involves not only the upregulation of key lipoprotein receptors that facilitate cholesterol uptake but also the activation of genes directly involved in cholesterol biosynthesis.

The loss of this P4-guided regulatory effect—whether due to the natural or artificial induction of perimenopausal state transition (e.g. VCD-induced AOF model) or through pharmacological inhibition of PR and downstream ERα signalling—likely contributes to the observed reductions in brain glucose metabolism in females[87]. Moreover, the results of this study reveal that through its regulatory role in cholesterol homoeostasis, P4-guided ERα signalling unexpectedly, in turn, affects ERRα signalling. Disruption of ERRα signalling, in turn, not only impairs neuronal mitochondrial bioenergetics but also enhances the catabolism of NAAG into aspartate and glutamate, two key excitatory neurotransmitters in the brain[88]. While aspartate carbons may partially sustain a rewired mini-TCA cycle under conditions of mitochondrial dysfunction, the accumulation of free glutamate leads to heightened stochastic neurotransmitter release at synaptic regions. This increase in intrinsic excitability and basal bioenergetic demand, especially in neurons with already compromised mitochondrial function, renders them highly vulnerable to additional excitotoxic insults. Such insults may include the age-related intraneuronal accumulation of amyloid precursor protein (APP)[89,90] or Aβ[91–94]. These mechanisms collectively contribute to female-biased susceptibility to the development and progression of LOAD.

Cerebral hypometabolism is a common feature of the prodromal stage of LOAD and is often associated with grey matter atrophy and cognitive decline[95,96]. Our findings provide mechanistic insights into why these metabolic changes may be more pronounced in women than in men. In our VCD-induced AOF mouse model, the extent of neurodegeneration and functional decline in the brain strongly correlated with the E2:P4 ratio, emphasising the intimate connection between brain health and endocrine status. Steroid hormones such as E2 and P4 freely cross the BBB, suggesting that their effects on the ageing brain are both direct and systemic. Importantly, our data indicated that the plasma E2:P4 ratio, a surrogate marker of high oestradiol-low progesterone imbalance, could serve as a potential biomarker to identify ageing women at elevated risk of cognitive and memory decline during the perimenopausal transition. Population-based studies[23,24,97–101] and meta-analyses[21] have indicated that hormonal shifts in women often begin earlier than anticipated, sometimes as early as the mid-thirties, even in women who are still menstruating regularly. These shifts are not limited to a decrease in E2 levels but include increased fluctuations in E2, diminished P4 levels (due to short luteal phases or anovulatory cycles), and dysregulation of the ovarian–pituitary–hypothalamic axis[22]. While E2 is typically perceived

as the primary reproductive hormone, P4 plays a critical and often underappreciated role in female physiology[102]. Despite this phenomenon, P4 is produced in substantially greater amounts (nanomolar range) than E2 (picomolar range) and increases by 1400% during the luteal phase, compared with only a 220% increase in E2 levels during the midcycle peak[103]. From an evolutionary perspective[104], this disproportionate increase in P4 levels for approximately 10 days per cycle likely serves to balance the growth-promoting effects of E2 by supporting cell maturation and differentiation[13,105]. This complementary relationship between E2 and P4 is evident in various physiological contexts. E2 stimulates cervical glands to produce mucus that facilitates sperm passage, whereas P4 counteracts this phenomenon by inhibiting mucus production. Similarly, E2 drives breast gland development during puberty (Tanner stages I–III), but under the influence of P4, it promotes the final maturation of breast tissue to Tanner stages IV–V[102]. In the brain, E2 has neuroexcitatory effects, whereas P4 counteracts these effects by promoting sleep, reducing anxiety, and mitigating addictive behaviours[106]. Given that neuronal mitochondrial OXPHOS is closely linked to the degree of cellular differentiation and maturation[107], our findings offer new insights into the pivotal role of P4 and its activated receptor in preserving the postmitotic identity of neurons. Specifically, P4 appears to modulate ERα promoter occupancy, which supports downstream cholesterol homeostasis in neurons, a critical factor for maintaining neuronal bioenergetics and functional stability.

Cholesterol is a critical component of neuronal physiology, and its depletion in neurons disrupts synaptic vesicle exocytosis, impairs neuronal activity, and ultimately leads to dendritic spine and synapse degeneration[95,108,109]. In addition to its essential role in maintaining neuronal structure and function, dysregulated brain cholesterol metabolism is closely linked to the risk and progression of LOAD. Numerous polymorphisms in genes involved in cholesterol transport, including *APOE, TREM2, ABCA7, INPP5D, CLU, SPI1*, and *SORL1*, have been identified as genetic risk factors for LOAD[110]. Our study highlights the underappreciated role of cholesterol as a regulator of mitochondrial bioenergetics, acting as an endogenous ligand for the transcription factor ERRα. While neuronal cholesterol is predominantly acquired exogenously via lipid carriers such as ApoE, de novo cholesterol biosynthesis also contributes significantly to total cholesterol levels, particularly under chronic stress conditions. For example, neurons in an APOE4 microenvironment[111] or those experiencing myelin degeneration rely more heavily on endogenous cholesterol production[64]. Consistent with prior findings[58,112], our data reveal that the binding of cholesterol to ERRα enhances its interaction with the cofactor PGC1α, thereby activating the nuclear activity of genes that regulate mitochondrial bioenergetics and OXPHOS functions. These effects are further modulated by an upstream PR-guided, ERα-mediated cholesterol biosynthetic network, supporting the notion that female subjects are more vulnerable to ERRα dysfunction[113]. This cholesterol–ERRα linkage underscores the importance of crosstalk between two seemingly unrelated neuroprotective hormones, P4 and E2, in regulating neuronal metabolism by enhancing glucose carbon utilisation. The *APOE4* allele, the strongest genetic risk factor for LOAD, is known to have greater penetrance in females[114]. We propose that hormonal imbalances, particularly disruptions in the E2:P4 ratio during the perimenopause state, may exacerbate neuronal cholesterol depletion and mitochondrial dysfunction, contributing to heightened biological sex-biased vulnerability to LOAD in female *APOE4* carriers[4]. This hypothesis was supported by our pilot data (Fig. 1f and Supplementary Fig. 2c, d), which showed that compared with noncarriers, samples from *APOE4* carriers exhibited greater depletion of nuclear PGC1α and ERRα. As ApoE is crucial for cholesterol transport from astrocytes to neurons, the impaired function of the *APOE4* variant further exacerbated cholesterol deficiency in neurons. This decrease in cholesterol availability is linked to reduced nuclear localisation and

activity of ERRα, as evidenced by the lower number of MAP2+ nuclear ERRα+ neurons in the brains of females with and without dementia than in the brains of their male counterparts (Fig. 1f and Supplementary Fig. 2c, d). Future research utilising mouse models with distinct humanised APOE variants (e.g. *APOE2, APOE3*, and *APOE4*) will be essential to further elucidate these mechanisms. Such studies address the limitations of using early-onset familial AD mouse models in the current work and provide valuable insights into how APOE genotype and hormonal imbalances interact to influence neuronal cholesterol homeostasis, mitochondrial bioenergetics, and LOAD susceptibility.

The critical role of ERRα in regulating mitochondrial metabolism and oxidative OXPHOS is well established. Our findings expand this understanding by demonstrating how mature neurons adapt to the loss of ERRα function. Specifically, our data revealed that the TCA cycle was the most affected metabolic pathway upon *Esrra* knockdown, primarily because of decreased expression of specific subunits in the SDH complex. This disruption led to the emergence of a "truncated TCA cycle," which was compensated for by the use of aspartate as an alternative carbon source to sustain metabolic flux. Similar adaptations have been reported in cancer cells with SDH deficiency[115] and under physiological conditions, such as in normal prostate tissues, to sustain fertility[116]. Our findings further highlighted that the reliance of neurons on aspartate metabolism affects the biosynthesis of NAAG, a retrograde neurotransmitter that acts on inhibitory Gi/Go-coupled metabotropic glutamate receptor 3 at presynaptic terminals[117]. NAAG provides a feedback mechanism to postsynaptic neurons, preventing excessive glutamate signalling. The decline in NAAG biosynthesis observed in our study led to an increase in the intracellular glutamate pool, resulting in a significant increase in the frequency of spontaneous mEPSCs. This phenomenon likely arose from both reduced feedback inhibition by NAAG and enhanced stochastic presynaptic glutamate release, as evidenced by electrophysiological changes in these neurons. The increased frequency of spontaneous firing imposes a substantial metabolic burden, as spontaneous neuronal activity accounts for 60–80% of the total energy consumption of the brain[118,119]. Consequently, elevated mEPSCs increase basal energy demands even in the absence of external stimuli, rendering neurons particularly vulnerable to metabolic inefficiencies. This vulnerability is compounded by the dual role of SDH as both an essential component of the TCA cycle and Complex II of the OXPHOS system. Impaired SDH activity compromises ATP production, heightening susceptibility to energy crises during periods of increasing energy demands[81,120]. Pathological damage to glutamatergic neurons, particularly within the cell bodies and neurites located in layers III and IV of the neocortex and the hippocampus, is a hallmark of LOAD[121]. Our findings suggest that the decrease in P4-guided ERα signalling, which mediates neuronal de novo cholesterol biosynthesis, contributes to this vulnerability. This decline can result from either a significant reduction in P4 or erratic elevations in E2 levels during the perimenopausal state[102], offering a new metabolic perspective on why neurons in early middle-aged female individuals are particularly susceptible to excitotoxic insults.

Our findings suggest that progesterone (P4) supplementation therapy could address the metabolic reprogramming events in neurons caused by a higher oestradiol-lower progesterone imbalance. However, for maximum neuroprotective efficacy, such therapy must be introduced early during the onset of perimenopause. In our study, we demonstrated that sustained P4 supplementation, which was delivered via osmotic pumps for more than 60 days, provided significant neuroprotective effects through the restoration of the physiological E2:P4 ratio. Consistent with previous reports[83], we accounted for the inherent delay in drug release associated with minipumps by administering an initial i.p. loading dose of P4 to achieve the desired concentrations promptly. Owing to its lipophilic and steroidal properties, P4 crosses the BBB efficiently and accumulates within brain tissue[122,123]. However, the peak effect following a single i.p. dose was transient, and the corrected E2:P4 ratio, when reduced below the critical threshold of 2.5 pg/ng, persisted for only approximately 8 h. The ideal therapeutic profile for P4 supplementation would maintain its concentration and the E2:P4 ratio within a narrow physiological range over an extended period. This phenomenon was achieved in our study using a subcutaneously implanted minipump drug delivery system. P4 has been shown to be both safe and effective in various clinical applications, such as HRT and treatment for acute traumatic brain injury[124–126]. However, our findings and those of prior studies indicate that increasing P4 doses beyond the optimal range does not necessarily enhance neuroprotective benefits. In agreement with previous research[127], we observed that compared with the recommended ideal dose of 8 mg/kg, chronic administration of P4 at a slightly higher dose (11.7 ± 0.7 mg/kg) resulted in significant neuroprotective effects. This dosing effectively mitigated the metabolic and molecular impacts of the VCD-induced AOF paradigm, inducing a higher oestradiol-lower progesterone imbalance, in a familial AD mouse model. Interestingly, our in silico analysis of a published dataset suggested that the effects of P4-guided ERα signalling on cholesterol homoeostasis were specific to the natural form of the hormone and were not replicated by synthetic progestins (e.g. R5020). This distinction between natural progesterone and synthetic progestins aligned with findings from the French E3N longitudinal observational study. In more than 80,000 menopausal women with a mean follow-up time of more than eight years, this study revealed that oestrogen-alone therapy increased the risk of breast cancer by 29%, whereas oestrogen combined with synthetic progestins heightened the risk by 69%. In contrast, oestrogen combined with natural progesterone did not significantly increase the risk of developing breast cancer[128]. Moreover, a recent randomised controlled trial demonstrated that daily oral intake of micronised natural progesterone significantly reduced the severity of night sweats and daytime hot flashes, improved sleep quality, and did so without inducing depression[129]. Although we selected natural progesterone as the preferred pharmacological intervention in this study, further investigation into the effects of clinically approved synthetic progestins would be highly interesting.

Collectively, our findings reveal an unexpected interplay between P4 and E2, which is critical for maintaining cholesterol balance and mitochondrial energy production in neurons. These results also highlight how a higher oestradiol-lower progesterone imbalance during the perimenopausal transition increases the susceptibility to AD/ADRD among ageing women. Notably, while the VCD-induced AOF mouse model has provided some insights into female perimenopause, the use of a chemical to induce such changes may not perfectly replicate all aspects of natural menopausal transition in humans[130]. Furthermore, systemic off-target effects on the liver, kidney[131] and cardiovascular system have been reported in long-term studies[132], which could interfere with the findings and should be monitored in future investigations.

## Method

### Ethics and safety regulation compliance statement

All experimental protocols involving wet laboratory procedures were reviewed and approved by the University Safety Office at The Chinese University of Hong Kong (CUHK). Experiments involving mice were conducted using colonies maintained and bred at the Laboratory Animal Service Centre (LASEC) of CUHK, under the approved protocol number 19-243-GRF. Animal care and handling were carried out in full compliance with institutional guidelines and the regulations stipulated by the Hong Kong Special Administrative Region.

### Study design, statistical analysis, and reproducibility

The aim of this study was to investigate the persistent molecular and metabolic alterations caused by an imbalance in the progesterone (P4)

to oestradiol (E2) hormonal ratio. This imbalance underpins its sustained impact on brain physiology, thus establishing it as a female sex-biased biological risk factor for age-related dementia. To achieve this, complementary transcriptomic, metabolomic, and pharmacological approaches were utilised to elucidate the effects of progesterone-directed loss of ERα signalling both in vivo and in vitro. The effects of the perimenopausal state were simulated using a VCD-induced AOF model, which was employed to assess animal behaviour, metabolic reprogramming, neuronal integrity, and functional changes. These evaluations were conducted on wild-type mice, 3xTg familial AD mice, and mice with neuron-specific knockdown of oestrogen-related receptor alpha (Esrra/ERRα) in the forebrain region. In human studies, the research involved re-analysing the ROSMAP brain tissue single-nuclei RNA-sequencing dataset and bulk transcriptomics and metabolomics datasets, released from published studies (for details, please refer to the section titled: ROSMAP single-nucleus RNA sequencing (snRNA-seq), bulk RNA sequencing and metabolomics dataset analyses). Additionally, immunostaining analyses were performed on brain tissues from patients diagnosed with LOAD and ND individuals to corroborate the findings.

Sample sizes for animal experiments were determined based on prior studies, ensuring sufficient statistical power. A sample size of 12–15 animals per group was calculated to achieve 80% power to detect a 24% difference between group means at a significance level of 0.05 using an unpaired t-test. As differences smaller than 25% are not deemed biologically relevant, a minimum of 12–15 animals per group was recommended to ensure sufficient statistical power for detecting meaningful differences in the experimental outcomes. For experiments involving genetic modification of Esrra and subsequent quantification of outcome measures, littermates manipulated with scrambled shRNA were used as controls. For experiments using commercially obtained mice, group allocation was randomised to minimise bias. Cell-based experiments were conducted at least three times, with a minimum of three replicates per condition to ensure reproducibility. Wherever practical, investigators were blinded to treatment assignments and group information to avoid bias in data collection and analysis. All animal and experimental protocols were reviewed and approved by the relevant authorities at CUHK, as detailed above.

### Reagents, RNA interference, and plasmids
Unless otherwise stated, all chemicals and reagents were obtained from Sigma-Aldrich. Comprehensive details of antibodies, specialised reagents, assay kits, sequence-based reagents, analytical software, oligonucleotide sequences, and a list of unique reagents are provided in Supplementary Data 2. Unique reagents generated during this study will be made available upon reasonable request to the lead contact, subject to the completion of a Materials Transfer Agreement.

### Allen Brain Map database
The relative expression levels of target genes of interest in various brain cell types within the whole cortex and hippocampus regions of the mouse brain were queried using the Allen Brain Map Cell Types Database. The Whole Cortex and Hippocampus−Smart-Seq (2019) datasets were employed for this analysis. Scatter plots generated via the Transcriptomics Explorer were utilised to visualise the relative expression levels of target genes across different brain cell types[133].

### ROSMAP single-nucleus RNA sequencing (snRNA-seq), bulk RNA sequencing, and metabolomics dataset analyses
The datasets utilised in this study were accessed with appropriate permissions from the Accelerating Medicines Partnership® Program for Alzheimer's Disease (AMP® AD) platform under a signed data use agreement. All meta datasets were derived from the ROSMAP cohort and included: (1) Single-nucleus RNA sequencing (snRNA-seq)−Data

from the prefrontal cortex region (snRNAseqPFC_BA10, Syn18485175) (https://www.synapse.org/Synapse:syn18485175) as described in the original study[52]; (2) Bulk RNA sequencing−Data from the dorsolateral prefrontal cortex (DLPFC), posterior cingulate gyrus (PCG), and anterior cingulate (AC) regions (syn3388564) (https://www.synapse.org/Synapse:syn3388564)[134]; and (3) Non-targeted metabolomics−Data from the DLPFC region (syn3157322) (https://www.synapse.org/Synapse:syn3157322)[135]. Key clinical and pathological features of the de-identified subjects included in these studies are detailed in Supplementary Data 1. For the snRNA-seq dataset, the definitive disease status (i.e. LOAD vs ND) was provided in the original published paper and was adopted in this study. For bulk transcriptomics and metabolomics samples, disease status was defined using the Ward D2 hierarchical clustering method, with reference to the following neuropathological and clinical measures: CERAD score (a semi-quantitative measure of neuritic plaques); Braak staging score (a semi-quantitative measure of neurofibrillary tangles); Cogdx score (a clinical consensus diagnosis of cognitive status at the time of death); and Dcfdx score (a clinical diagnosis of cognitive status).

For the analysis of the single-nucleus RNA sequencing (snRNA-seq) dataset, a total of 70,634 high-quality nuclei were processed using the Seurat 4 pipeline[136]. Dimensionality reduction and visualisation were performed using t-distributed stochastic neighbour embedding (t-SNE), with the top 30 principal components used as input. Cell types were annotated based on the provided cell annotation file and further validated using established cell type-specific markers. Cell type-specific gene expression patterns were visualised using the DotPlot and VlnPlot functions in the Seurat package. To estimate the significance of differences between the LOAD and ND groups, the Find-Markers function was applied using the MAST method. The analysis incorporated post-mortem interval (PMI), age, and biological sex of the samples as covariates. A generalised linear model was employed to identify significantly altered genes or metabolites between LOAD and ND groups, correcting for the aforementioned covariates to ensure robust and unbiased results.

### Post-mortem human brain samples
Frozen post-mortem brain samples from diseased ataxia-telangiectasia (A-T) patients and age-matched controls were obtained from the NeuroBioBank, National Institutes of Health (NIH). All samples were fully anonymised prior to distribution. The US Human Studies Board classified these tissues as exempt. All studies involving human brain tissue were conducted in strict compliance with the ethical standards of the NIH. Frozen brain sections were prepared from brain blocks and utilised for immunohistochemistry. Tissue sections were examined and imaged using a fluorescent microscope (Olympus BX53 with DP80 camera) equipped with a 20 × objective lens (UPlanSApo; 0.75 NA) and an X-Cite 120Q light source (Excelitas Technologies Corp.).

### Animal maintenance and brain tissue harvesting
**Animal maintenance, ethics approval, and general information.** C57BL/6J mice (Jackson Laboratory strain #: 000664) and 3xTg-AD mice on a C57BL/6 J background (Jackson Laboratory strain #: 033930; B6.Cg-Tg(APPSwe,tauP301L)1Lfa Psen1tm1Mpm/2J) were procured from the Jackson Laboratory. The mouse colonies were maintained and bred at the LASEC of the CUHK under the approved protocol number 19-243-GRF. All procedures related to animal care and handling adhered to institutional guidelines and the regulatory standards of the Hong Kong Special Administrative Region. As the study focused on the perimenopausal state, only female mice were included. The ages of the animals used in specific experiments are detailed in the text or indicated in the schematic diagrams of treatment protocols in the main figures. All animals were housed in a specific pathogen-free (SPF) environment under a 12-h light/dark cycle at room temperature

(22 ± 2 °C) with constant humidity levels (50–70%). Food and water were provided ad libitum. Female mice were randomly selected from the available colonies under these conditions. Unless otherwise specified, mice were euthanised via carbon dioxide exposure, with euthanasia confirmed by bilateral thoracotomy.

**Brain tissue harvesting.** Mice were first anaesthetised via i.p. administration of 1.25% ($v/v$) Avertin at a dose of 30 ml/kg body weight. After anaesthesia, the heart was surgically exposed, the left cardiac chamber catheterised, and the right atrium opened. Chilled physiological saline was perfused transcardially for 3 min to remove blood from the body. Following perfusion, the cranial bones were opened, after which the cortex and cerebellum tissues were harvested. The harvested tissues were snap-frozen in liquid nitrogen and stored at −80 °C until required for further use.

### The VCD-induced AOF mouse model for perimenopause and menopause states

The VCD model is a well-established and widely used system to induce a "menopause-like" state in rodents. VCD is well-tolerated by animals and selectively depletes ovarian follicles, resulting in ovarian failure without significant systemic toxicity[54,137]. To minimise the confounding effects of chronological ageing, the impact of the menopause state was assessed in 3-month-old mice. Test subjects received i.p. injections of VCD (160 mg/kg body weight) diluted in sesame oil or a sesame oil vehicle (sham group) for a total of 14 injections over 15 days. Daily body weight measurements were recorded throughout the treatment period. The oestrous cycles of each animal were monitored daily via vaginal cytology to determine cycling stages and the point at which cycles ceased, indicating ovarian failure. The average length of the oestrous cycle was calculated, where the cycle length was defined as the time (in days) between two successive proestrus phases. On each day of proestrus, blood samples were collected to assess plasma levels of FSH) and oestradiol (E2). Mice were briefly immobilised (<1 min) in a restraint tube (Braintree Scientific, Braintree MA), and a small nick was made at the end of the tail using a scalpel to collect blood. Samples were centrifuged at room temperature for 15 min at 16,000 × $g$ to isolate the plasma fraction. Plasma levels of FSH, progesterone (P4), and E2 were determined using commercial ELISA kits (details provided below). Following the completion of the final behavioural tests at the conclusion of the treatment paradigm, ~6-month-old animals were sedated, and brain tissue harvesting was performed as previously described in other sections.

### Progesterone administration

In the dosage testing trial, mice were administered i.p. injections of either 4 mg/kg or 8 mg/kg progesterone (USP) (Sigma). Progesterone was prepared as a 16 mg/ml solution dissolved in 100% dimethylsulphoxide (DMSO). Mice were injected while under isoflurane anaesthesia (induction at 4%; maintenance at 1.5% in a $NO_2/O_2$ mixture of 70/30%) (RWD Life Science Co, Shanghai, China) and were allowed to recover fully from anaesthesia after administration. For sustained delivery, Alzet minipumps (Model 2004) with an infusion rate of 0.25 μl/h and a 28-day delivery capacity were used. Minipumps were loaded with a progesterone solution (50 mg/ml dissolved at 37 °C in 100% DMSO) and primed by submerging them in sterile 0.9% saline at 37 °C overnight to ensure immediate drug delivery upon implantation. Minipumps were implanted subcutaneously behind the neck under isoflurane anaesthesia. Immediately after implantation, mice received a progesterone loading dose (4 mg/kg i.p., dissolved in DMSO) and were allowed to recover from anaesthesia. To maintain continuous progesterone delivery, preconditioned new pumps were replaced every 20 days under anaesthesia, sustaining the treatment for a total of 60 days. Following this treatment period, post-hoc behavioural analyses and brain tissue harvesting were performed.

### Intracerebroventricular AAV injection into mouse brains

Adeno-associated virus serotype 9 (AAV9) particles containing the hSyn-EGFP-MIR-ESRRA shRNA shuttle and scrambled control were purchased from Shanghai GeneChem. Mice aged P30–P45 were anaesthetised with isoflurane (2–5%) (RWD Life Science Co, Shanghai, China) and their scalps immobilised in a stereotaxic apparatus (RWD). After a precise craniotomy, AAV viral particles were loaded into a Hamilton syringe and injected into specific brain regions. The injection sites were targeted as follows: (1) dorsal hippocampus CA2 (Left: M/L: −1.70 mm, A/P: −1.55 mm, D/V: −1.70 mm); (2) medial prefrontal cortex (Left: M/L: −0.5 mm, A/P: 2.10 mm, D/V: −2.0 mm). Approximately $10^{12}$ vg/ml (~100 nl) of AAV particles were injected into each region using a syringe pump (KD Scientific, Holliston, USA) at a controlled rate of 10 nl/min. Following the injection, the needle was left in place for 20 min before being gradually withdrawn to minimise backflow. To prevent hypothermia during and after anaesthesia, mice were placed on a heating pad throughout the procedure and monitored until fully awake. Injected mice were observed daily for recovery, body weight, and signs of infection. A recovery period of 30 days was allowed before conducting any behavioural tests or collecting tissue samples.

### Animal behavioural tests

All tests were conducted as previously described, with slight modifications[138]. Behavioural testing was performed during the light phase of the circadian cycle, between 09:00 and 17:00, in a counterbalanced order across different treatment groups. Previous studies, including our own, have demonstrated that the behaviours assessed in this study are not influenced by the time of day during testing or by single housing of C57BL/6J mice, the wildtype background strain[139,140]. Prior to each test, mice were allowed to habituate to the testing rooms to minimise stress. All experiments were conducted blind to the treatment groups to reduce bias. Behavioural tests were recorded using an overhead camera and analysed with the Smart 3.0 Video Tracking system (Panlab), as detailed below.

**MWM test.** The MWM test was performed as previously described, with slight modifications[138]. A blue circular tank [90 cm in diameter and 35 cm in height] was filled with water maintained at approximately 22 °C. A circular platform (10 cm in diameter) was submerged 1 cm beneath the water surface in a designated target quadrant. Bright and contrasting shapes were placed on the walls surrounding the tank to serve as spatial reference cues. Training was conducted over six consecutive days, with four trials per day and an inter-trial interval of 1–1.5 min. For each trial, mice were randomly placed into one of the four starting locations around the tank. Each mouse was allowed to swim until it located the hidden platform. If the platform was not located within 60 s, the experimenter gently guided the mouse to the platform. Once on the platform, mice remained there for 15 s before being returned to their home cage. Daily performance data were averaged across the four trials. On day 7, a probe trial was conducted. The hidden platform was removed, and mice were placed in the pool to swim freely for 60 s. The time spent in each of the target quadrants and the latency to reach the former platform location were recorded.

**Rotarod test.** The rotarod test was performed as described previously, with slight modifications. Mice were placed on a stationary rotarod device (IITC Life Sciences) in a well-lit room. The rotarod was then activated and accelerated from 0 to 45 revolutions per minute (rpm) over a duration of 5 min[138]. The latency to fall off the rod was recorded for each mouse. Each mouse underwent four trials in total, with inter-trial intervals of 30 min. All trials were conducted within a single day to assess motor coordination and balance.

**Y-maze test.** The short-term working memory was assessed in the Y-maze spontaneous alternation test using a grey opaque Perspex

Y-maze with three arms each containing a visual cue (arm dimensions: 15 × 10 × 10 cm) as previously described in ref. 138. In this discrete trial procedure, there are two phases to each trial: a sample phase (the information-gathering stage, where the animal runs to one goal arm of the maze and a memory trace of this event is formed) and a choice phase (the animal's choice between the sampled and unsampled arms may or may not be guided by the memory of recently visiting the former). The sample phase began with the placement of the mouse in the designated starting arm of the maze. The mouse could freely explore two of the three arms for 5 min. The mouse was then returned to the home cage for 30 min before the start of the choice phase. The choice phase began with the placement of the mouse in the same starting arm as the sample phase and was allowed to freely explore all three of the arms for another 5 min. An arm entry was defined as four limbs within the arm. The percentage number of alternations was calculated as the number of actual alternations divided by the maximal number of alternations (the total number of arm entries minus 2). The total number of moves was also recorded as an index of ambulatory activity.

**Open field test.** This test was for the evaluation of anxiety and locomotion, and it was conducted as previously described in ref. 141. Rodents show distinct aversions to large, brightly lit, open and unknown environments. It is assumed that they have phylogenetically been conditioned to see these types of environments as dangerous. In the experiments, mice were placed in the centre of an open-field arena 50 cm (length) × 50 cm (width) × 38 cm (height) that was made from white high-density and non-porous plastic. Free and uninterrupted movement of the mouse was allowed for 5 min and movements were videotaped. Locomotor activity was measured using the number of crossed grids, while exploratory activity was measured using the number of rearing on the hind feet. The total travel distance and time spent in the outer vs the inner zone areas of the field were computed using a Smart 3.0 video tracking system (Panlab, Harvard Apparatus).

**Enrichment of neuronal cells from the adult mouse brain**

**Isolation of neurons from the adult mouse brain.** The tissue dissociation protocol was performed according to the manufacturer's instructions with slight modifications. All reagent volumes provided were optimised for 20–30 mg of adult murine brain tissue, with proportional upscaling for multiple tissues. Buffers were degassed, stored on ice, and only pre-cooled solutions were used throughout the procedure. To maintain the integrity of the samples, vortexing was strictly avoided at all steps. This approach ensured efficient tissue dissociation while preserving cellular and molecular components.

**Brain tissue harvesting and dissociation.** After euthanasia by carbon dioxide exposure, each mouse was perfused with chilled PBS for 3 min. Brain tissue was carefully harvested and dissected into 8–10 sagittal slices using a murine brain matrix (Ted Pella). Prefrontal cortex tissues, where stereotaxic injection of Esrra shRNA was performed, were pooled into a petri dish filled with D-PBS (Dulbecco's Phosphate Buffered Saline with calcium, magnesium, 1 g/L glucose, and 36 mg/L sodium pyruvate) and kept on ice until downstream processing. Tissue dissociation was carried out using the Adult Brain Dissociation Kit for mice (Miltenyi Biotec) according to the manufacturer's instructions. Prefrontal cortex tissues from each mouse were transferred to a gentleMACS C Tube (Miltenyi Biotec) containing enzyme mixes from the kit for enzymatic digestion. Mechanical enzymatic dissociation was performed using the gentleMACS Octo Dissociator with Heaters (Miltenyi Biotec) under program 37C_ABDK_01. Following dissociation, the brain tissue homogenate was passed through a 70 μm cell strainer (Corning, Corning, MA, USA) to ensure a uniform cell suspension.

**Debris and red blood cell removal.** Debris and red blood cell removal were performed according to the protocol provided with the Adult Brain Dissociation Kit (Miltenyi Biotec), with slight modifications. During centrifugation of the density gradient, the brake was turned off to allow more precise separation of the three phases: the bottom cell suspension, intermediate myelin layer, and top supernatant containing debris of lower density than the targeted cells. This adjustment ensured the reliable removal of all myelin residues. Residual red blood cells remaining after whole-body perfusion were eliminated via an osmotic gradient using the Red Blood Cell Removal Solution at a 1:10 dilution in double-distilled water (ddH$_2$O). The remaining cells were washed and resuspended in 80 μL of PBS buffer [Dulbecco's Phosphate Buffered Saline (1 ×) without calcium and magnesium, supplemented with 0.5% bovine serum albumin] per brain homogenate. For experiments involving more than one mouse per condition, up to two brain homogenates were pooled (maximum neural tissue weight of 1000 mg), with reagent volumes appropriately scaled according to the manufacturer's instructions.

**Enrichment of neurons vs non-neuronal cells.** The following steps were performed as quickly as possible using pre-cooled solutions to prevent antibody capping on the cell surface and reduce non-specific cell labelling. To isolate neurons from the mixed cell population, the Non-Neuronal Cells Biotin-Antibody Cocktail was added to the resulting cell suspension to label non-neuronal cells with biotin-conjugated antibodies. The mixture was gently mixed and incubated for 5 min in the dark at 2−8 °C. After incubation, cells were washed in D-PBS/BSA buffer, centrifuged, and resuspended in the same buffer. Subsequently, anti-Biotin MicroBeads were added to the cell suspension, mixed thoroughly, and incubated for 10 min in the dark at 2−8 °C. The single-cell suspension was then loaded onto a pre-rinsed LS Column (Miltenyi Biotec) placed within the magnetic field of a MACS Separator (Miltenyi Biotec). The flow-through containing unlabelled neuronal cells was collected. The LS Column was then removed from the separator and placed in a new collection tube to elute magnetically labelled non-neuronal cells by pushing the plunger into the column. To assess the purity of the enriched neuronal and non-neuronal fractions, immunoblot-based purity analysis was performed using known intracellular markers (to avoid cross-reactivity with antibodies targeting cell surface proteins included in the Non-Neuronal Cells Biotin-Antibody Cocktail). Once confirmed, the enriched neuronal fraction was sent for metabolite profiling.

**Bulk RNA sequencing**

Frozen whole cortex tissues from the test subjects of the VCD-induced AOF model, laser microdissection-enriched GTP-positive cortex and hippocampal tissues from the AAV-study model, or enriched neuronal populations were sent to Novogene for total RNA extraction and RNA sequencing using the Illumina HiSeq X Ten platform. The resulting FASTQ files were analysed for quality control using FastQC (v. 0.11.9) from Babraham Bioinformatics. Once the quality of the data was confirmed, genome indexes were generated with the STAR software using mouse annotation and genome assembly from Gencode Release 28 (GRCm39, May 2021). The cleaned datasets were then aligned to the genome indexes using the STAR alignment software. Quantification of sequencing data was performed with FeatureCounts. Data extraction, matrix construction, and differential gene expression analysis were carried out using the DESeq2 package in R, following standard procedures. Pathway analysis of the DEGs was conducted using Enrichr (https://maayanlab.cloud/Enrichr/)[142]. The original FASTQ files and metadata files were deposited on GEO Omnibus.

## Untargeted and targeted metabolome analysis by capillary electrophoresis time-of-flight mass spectrometry (CE-TOFMS) and liquid chromatography (LC)-TOFMS

Metabolome analyses were performed in mouse frontal cortex tissue, primary neuron and astrocyte cultures using CE-TOFMS for both cationic and anionic metabolites on the basis of a service purchased from Human Metabolome Technologies' standard library. Samples were sent to HMT, where their weight were first measured. For CE-TOFMS preparation, samples were mixed with 1500 μl of 50% acetonitrile in water ($v/v$) containing internal standards (10 μM) and homogenised by a homogeniser (1500 rpm, 120 s × 1 time). The supernatant (400 μl) was then filtered through a 5-kDa cut-off filter (ULTRAFREE-MC-PLHCC, HMT) to remove macromolecules. The filtrate was centrifugally concentrated and resuspended in 50 μl of ultrapure water immediately before measurement. Whereas for LC-TOFMS preparation, weighted samples were mixed with 300 μl of 1% formic acid in acetonitrile ($v/v$) containing internal standards (10 μM) and homogenised by a homogeniser (1500 rpm, 120 s × 2 times). The mixture was yet again homogenised after adding 100 μl of Milli-Q water and then centrifuged (2300 × $g$, 4 °C, 5 min). After the supernatant was collected, 300 μl of 1% formic acid in acetonitrile ($v/v$) and 100 μl of MilliQ water were added to the precipitation. The homogenization and centrifugation were performed as described previously, and the supernatant was mixed with the previously collected one. The mixed supernatant was filtered through a 3-kDa cut-off filter (NANOCEP 3 K OMEGA, PALL Corporation, Michigan, USA) to remove proteins and further filtered through a column (Hybrid SPE phospholipid 55261-U, Supelco, Bellefonte, PA, USA) to remove phospholipids. The filtrate was desiccated and resuspended in 200 μl of 50% isopropanol in Milli-Q water ($v/v$) immediately before the measurement.

**CE-TOFMS measurement.** The compounds were measured in the Cation and Anion modes of CE-TOFMS-based metabolome analysis in the following conditions, as previously reported in ref. 143. Samples were diluted in a 2-fold dilution for measurement to improve the quality of the CE-MS analysis.

Peaks detected in both CE-TOFMS and LC-TOFMS were extracted using automatic integration software (MasterHands ver. 2.17.1.11 developed at Keio University) in order to obtain peak information, including $m/z$, migration time (MT) in CE, retention time (RT) in LC, and peak area. The peak area was then converted to relative peak area by the following equation. The peak detection limit was determined based on a signal-to-noise ratio = 3.

**Relative peak area = metabolite peak area/internal standard peak area × sample amount**

Putative metabolites were then assigned from HMT's standard library and Known-Unknown peak library on the basis of $m/z$ and MT or RT. The tolerance was ±0.5 min in MT and ±0.3 min in RT, ±10 ppm (CE-TOFMS) and ±25 ppm (LC-TOFMS) in $m/z$. If several peaks were assigned the same candidate, the candidate was given the branch number.

**Mass error (ppm) = (measured value − theoretical value)/measured value × 10^6**

Subsequent absolute quantification was performed in target metabolites. All the metabolite concentrations were calculated by normalising the peak area of each metabolite with respect to the area of the internal standard and by using standard curves, which were obtained by single-point (100 μM or 50 μM) calibrations. Significantly changed metabolites ($p < 0.05$) were enriched and analysed by the Metabolite Set Enrichment Analysis (MESA) or the Joint Pathway Analysis module (with KEGG metabolic gene expression data) on MetaboAnalyst (https://www.metaboanalyst.ca/MetaboAnalyst/ModuleView.xhtml).

## Stable-isotope labelled glucose and aspartate metabolite tracing

Metabolic fate and competitive metabolic flux of glucose and ethanol were studied. In primary neurons, glucose-$^{13}C_6$ isotope alone or aspartate-$^{13}C_4$ incubation for 4 h was first performed, followed by tracing by the capillary electrophoresis-time of flight mass spectrometer (CE-TOF/MS). DIV12-14 neurons pre-treated with various drug treatment procedures were incubated in the glucose-free medium supplemented with glucose-$^{13}C_6$ isotope (Cambridge Isotope, CLM-1396) or a custom-ordered aspartate-free medium supplemented with aspartate-$^{13}C_4$ isotope (Cambridge Isotope, CLM-1801-H-PK). Cell lysates were collected at 4-h post-incubation by being washed twice with 10 ml of chilled PBS solution, and subsequently incubated with 1 ml of methanol containing 25 μm internal standards (methionine sulfone, 2-(N-morpholino)-ethanesulfonic acid (MES) and D-camphor-10-sulfonic acid) for 10 min. Four hundred microliters of the extracts were mixed with 200 μl Milli-Q water and 400 μl chloroform and centrifuged at 10,000 × $g$ for 3 min at 4 °C. Subsequently, 400 μl of the aqueous solution was centrifugally filtered through a 5-kDa cut-off filter to remove proteins. The filtrate was centrifugally concentrated and dissolved in 50 μl of Milli-Q water that contained reference compounds (200 μm each of 3-aminopyrrolidine and trimesate) immediately before metabolome analysis.

The relative concentrations of all the charged metabolites in samples were measured by CE-TOFMS, following the methods as previously reported in ref. 144. In brief, a fused silica capillary (50 μm internal diameter × 100 cm) was used with 1 m formic acid as the electrolyte. Methanol: water (50% $v/v$) containing 0.1 μm hexakis (2,2-difluoroethoxy) phosphazene was delivered as the sheath liquid at 10 μl min−1. Electrospray ionisation (ESI)-TOFMS was performed in positive-ion mode, and the capillary voltage was set to 4 kV. Automatic recalibration of each acquired spectrum was achieved using the masses of the reference standards [(13 C isotopic ion of a protonated methanol dimer (2 MeOH + H)]+, $m/z$ 66.0632) and ([hexakis (2,2-difluoroethoxy) phosphazene + H]+, $m/z$ 622.0290). Quantification was performed by comparing peak areas to calibration curves generated using internal standardisation techniques with methionine sulfone. The other conditions were identical to those described previously in ref. 144. To analyse anionic metabolites, a commercially available COSMO (+) (chemically coated with cationic polymer) capillary (50 μm internal diameter × 105 cm) (Nacalai Tesque) was used with a 50 mm ammonium acetate solution (pH 8.5) as the electrolyte. Methanol: 5 mM ammonium acetate (50% $v/v$) containing 0.1 μm hexakis (2,2-difluoroethoxy) phosphazene was delivered as the sheath liquid at 10 μl min−1. ESI-TOFMS was performed in negative ion mode, and the capillary voltage was set to 3.5 kV. For anion analysis, trimesate and CAS were used as the reference and the internal standards, respectively. Other conditions were identical to those described previously in ref. 145. MPE of isotopes, an index of isotopic enrichment of metabolites, was calculated as the percentage of all atoms within the metabolite pool that are labelled according to the established formula[146].

## In silico protein structure modelling and docking analyses

To obtain the structure ERRα with a complete and open LBD, an opened-yet-incomplete LBD was first extracted from the crystal structure from the Protein Data Bank (PDB: 7E2E)[147]. The missing alpha helix segment in this extracted structure was modelled and added to 7E2E using AlphaFold2 on the ColabFold v1.4.0 server with default settings[61]. The complete alpha-helix was then subsequently transferred to the structure PDB: 2PJL[148], and energy minimisation was conducted using a MMFF94 force field[149].

To perform ligand-receptor (i.e. cholesterol-ERRα) docking, ligand structures were first obtained from PubChem and subjected to energy minimisation using a MMFF94 force field[149]. The ERRα LBD

structures were prepared with AutoDockTools[150] by adding both polar and non-polar hydrogens and computing Gasteiger charges. Ligand docking simulation on AutoDock Vina[151] was centred at $(x, y, z) = ((21.39, 55.82, 10.94)$ with a box size of $30 \times 30 \times 30$ at exhaustiveness of 16. ERRα-ligand complexes were then refined by solvated molecular dynamics simulation under a TIP3P model[152].

To perform ERRα-PGC1α docking, the PGC1α peptide consisting of the 3rd LXXLL motif (amino acid residues 208-216) was first prepared by AlphaFold2 on the ColabFold v1.4.0 server with default settings[61]. The peptide was then docked onto the refined ERRα-ligand complex using HADDOCK2.4[62]. Ambiguous interaction restraints, namely the active site residues, derived from protein–protein interaction analysis of experimentally resolved structures, were used to drive the docking. Residues were regarded as accessible if they fulfilled the minimum 15% of relative solvent accessibility, whereas passive residues were defined within the 6.50 Å radius of active site residues. This allows a more definitive and accurate sampling of protein–protein interaction, instead of a more random selection based on conformational energies. Nonpolar hydrogens were removed during the docking. Semi-flexible regions were automatically defined. A three-step docking with default settings was conducted, starting with rigid docking, semi-flexible docking, and lastly structural refinement in the presence of water. The HADDOCK scores and binding energies calculated for the top three clusters were compared. The cutoff distances of proton-acceptor (hydrogen bonds) and carbon–carbon (hydrophobic contacts) were set at 2.5 Å and 3.9 Å, respectively.

Protein-peptide and protein–ligand interactions were modelled by Protein–Ligand Interaction Profiler (PLIP)[153], PyMOL and BIOVIA Discovery Studio. Representative images were created on PyMOL.

## Mouse primary neuronal culture

Mouse embryonic cortical neurons were isolated by standard procedures. Gravid females were killed on the 16th day of gestation, and E16.5 embryos were collected in ice-cold PBS-glucose. The cortical lobes were removed, following which the meninges were removed and the cortices were placed in 1× trypsin-EDTA solution for 10 min, with manual shaking for 5 min. After digestion, an equal volume of DMEM with 10% (v/v) FBS was added to inactivate the trypsin. Samples were then centrifuged at $200 \times g$ for 5 min. Supernatant was removed, followed by transferring the pellet to fresh Neurobasal medium supplemented with B-27, penicillin/streptomycin (1×) and L-glutamine (2 mM; GlutaMAX) prior to gentle resuspension. Tissue was triturated ten times through a 5 ml pipette and allowed to settle to the bottom of a 15-ml conical tube. Dissociated cells in solution above the pellet were removed. Surviving cells were identified by trypan blue exclusion and counted before plating on poly-L-lysine-coated (0.05 mg ml⁻¹) glass coverslips. Enrichment of neuronal culture was performed using both a previously reported method. 1.5 μM 5-fluoro-2′-deoxyuridine (FdU) (Sigma) was added at DIV4 and incubated for 24 h to kill any proliferating cells in the dish. Medium containing FdU was then replaced with fresh medium without FdU (old to new complete Neurobasal medium in a 1:1 ratio). Half of the medium was then replaced every 3 days for maintenance[154]. Unless otherwise specified, cells were plated in 24-well plates at 50,000 cells per well and allowed to mature for over 7–10 days in vitro (DIV) before transfection or AAV induction. For other experiments, cells were grown for a minimum of 14-16 DIV (DIV14) before any drug-treatment experiments. Cultures subjected to drug treatment were refreshed with new conditioned medium containing fresh drug compounds every 24 h, such that the drug dosage was kept approximately consistent over time.

## Transfection and transduction

Primary neurons were isolated from E16 embryos as described above. DNA constructs were transfected with Lipofectamine LTX in the presence of Plus Reagent (ThermoFisher, 15338100) into both types of primary culture following the manufacturer's protocol. At 5 h after transfection, cells were refreshed with new culture medium (or conditioned medium for primary neurons) and further incubated for 48–72 h to allow recovery and ectopic expression of transfected constructs.

For the transduction of commercially purchased AAV particles or in-house generated lentiviral particles, dissociated primary neuronal cultures were allowed to mature until DIV6-7 prior to the initiation of the transduction process at an approximate multiplicity of infection (MOI) $1.5 \times 10^5$. Cautions were taken on the volume of vector added was which shall not exceed about 1/10 of the medium volume. At DIV21, neurons were almost ~90% transduced, as evaluated by the GFP signals.

## Quantitative real-time PCR (qPCR) analysis

Total cellular RNA was purified from brain tissues or cultured cells using the RNease kit (Qiagen) following the manufacturer's protocol. For qPCR, RNA was reverse-transcribed using the High-Capacity cDNA Reverse Transcription Kit (Applied Biosystems) according to the manufacturer's instructions. The resulting cDNA was analysed by qRT-PCR using SYBR Green PCR Master Mix (Applied Biosystems). All reactions were performed in a Roche LightCycler (LC) 480 instrument using the following protocol: pre-incubation at 95 °C for 15 min (1 cycle); denaturation at 94 °C for 15 s, annealing and extension at 55 °C for 30 s (40 cycles), melting at 95 °C for 5 s, 65 °C for 60 s and 95 °C continues (1 cycle) followed by cooling at 40 °C for 30 s. The specificity of the primers was confirmed by observing a single melting point peak. qPCR efficiency was calculated from the slope between 95 and 105% with a coefficient of reaction $R^2 = 0.98$-0.99. A total of 7–9 biological replicates × at least three technical replicates were performed for each treatment group. Normalisation of results to stabilised expressed reference gene *Rpl13* in the brain was performed[155]. Data was analysed using the comparative Ct method (ΔΔCt method). Target gene primers are listed in Supplementary Data 2.

## Coimmunoprecipitation and SDS-PAGE Western Blotting

Isolated brain tissues or cell pellets were homogenised in RIPA buffer (Millipore) with 1× complete protease inhibitor mixture (Roche) and 1× PhosSTOP phosphatase inhibitor mixture (Roche) on ice, then centrifuged for 10 min at $18,400 \times g$ to remove large debris. The protein concentration of the supernatant was determined by Bradford Assay (Bio-Rad). For coimmunoprecipitation, 1 mg of the total cell lysates was first incubated with control IgG (Santa Cruz Biotechnology) for 30 min, precleared with 50 μl Dynabeads Protein G (Invitrogen), and then incubated with various antibodies overnight at 4 °C, using the suggested dilutions from the product datasheets. Beads bound with immune complexes were collected by DynaMag-2 (Life Technologies) and washed three times before elution in 90 μl of buffer containing 0.2 M Glycine-HCl, pH 2.5, which was neutralised with 10 μl of neutralisation buffer (1 M Tris-HCl, pH 9.0). The eluates were subjected to 9–15% SDS-PAGE and Western blot analysis.

For SDS-PAGE (polyacrylamide gel electrophoresis), 100 μg of proteins derived from cell or tissue lysates were prepared in 5× sample buffer [10% w/v SDS; 10 mM beta-mercaptoethanol; 20% v/v glycerol; 0.2 M Tris-HCl, pH 6.8; 0.05% Bromophenol Blue]. With a Bio-Rad system, separating gels of different acrylamide percentages (6–15%) were prepared with the following components in double-distilled water: acrylamide/bis-acrylamide (30%/0.8% w/v); 1.5 M Tris (pH = 8.8); 10% (w/v) SDS; 10% (w/v) ammonium persulfate and TEMED. and a 5% stacking gel (5 ml prep: 2.975 ml Water; 1.25 ml 0.5 M Tris-HCl, pH6.8; 0.05 ml 10% (w/v) SDS; 0.67 ml acrylamide/bis-acrylamide (30%/0.8% w/v); 0.05 ml 10% (w/v) ammonium persulfate and 0.005 ml TEMED) were prepared. Samples were run in SDS-containing running buffer [25 mM Tris-HCl; 200 mM Glycine and 0.1% (w/v) SDS] until the dye front and the protein marker reached the foot of the glass plate. Standard immune-blotting procedures were used, which include

protein transfer to polyvinylidene difluoride (PVDF) membranes, blocking with non-fat milk, and incubation with primary and secondary antibody, followed by visualisation with the SuperSignal West Dura/Femto Chemiluminescent Substrate (ThermoFisher Scientific). Full scan blots can be found in "Source Data".

## Extraction and measurement of ERRα-bound cholesterol

Cerebral cortex samples collected from test subjects were homogenised on ice in radioimmunoprecipitation assay (RIPA) buffer. Co-immunoprecipitation of endogenous ERRα proteins was conducted with Dynabead Protein G beads prebound and crosslinked with ERRα antibody (ThermoFisher Cat # PA5-28749), at 4oC with continuous shaking for 24 h. Cholesterol, the ERRα ligand, was analysed using LC-tandem mass spectrometry (LC-MS/MS). The analysis was conducted using an Agilent 1260 Infinity LC system equipped with a 1260 Infinity Diode Array Detector HS (Agilent Technologies) and coupled to an Impact HD mass spectrometer (Bruker, Milton, ON, Canada). Chromatographic separation was performed on an Agilent Zorbax Eclipse Plus C18 column (4.6 × 10 mm, 3.5 μm). The mobile phase consisted of two components: mobile phase A (methanol/water/0.1% formic acid, 3:1, $v/v$) and mobile phase B (100% isopropanol/0.1% formic acid; Millipore, Sigma-Aldrich). The elution gradient was programmed as follows: 40% mobile phase B for 0.5 min, ramping from 40% to 90% B over 4.5 min, held at 90% B from 4.5 to 6.5 min, decreased from 90% to 40% B over 0.1 min, and maintained at 40% B until 7 min. The flow rate of the mobile phase was set at 1 ml/min, with an injection volume of 20 μl. For mass spectrometry, electrospray ionisation was conducted in positive and total scan modes. The following parameters were used: capillary voltage of 4000 V, fragmentor voltage of 500 V, nebuliser gas pressure of 73 psi, drying gas temperature of 350 °C with a flow rate of 12 l/min, and an $m/z$ detection range of 150–800 Daltons. A fragment with an $m/z$ value of 369.4, representing a cholesterol-derived daughter ion, was detected. The relative intensities of this daughter ion (369.4 $m/z$) were quantified and correlated with ERRα protein levels and fold changes in cholesterol ion levels.

## Immunocytochemistry and immunohistochemistry

For immunocytochemistry, primary neuronal cultures were grown on 13-mm coverslips in 24-well plates, whereas for immunohistochemistry, 10 μm cryo-sections of frozen mouse brains were used. Samples were fixed with fresh 4% ($w/v$) paraformaldehyde (Sigma-Aldrich) for 10 min, washed and followed by permeabilisation with 0.3% Triton-X100 in PBS for 10 min. After blocking with 5% ($w/v$) BSA in PBS for 1 h, primary antibodies were added and incubated overnight at 4 °C. The following day, coverslips were washed three times (10 min each) with PBS. After rinsing, secondary antibodies were applied for 1 h at room temperature, followed by three additional washes with PBS. The coverslips were then inverted and mounted on glass slides with ProLong Gold Antifade Reagent (Life Technologies). Immuno-florescence was analysed, and Z-stack maximum projected images were photographed using a TCS SP8 confocal microscope (Leica Microsystems Inc.).

For quantification of synaptic protein presented in Fig. 6g, Immunoreactive puncta were identified on -100 μm MAP-positive dendrite segments from each neuron using the NIH ImageJ intensity threshold function and quantified with the Analyse Particles module of NIH ImageJ with respect to puncta density (per micron of analysed dendrite length) and puncta area. The threshold intensity for each antigen was set as the mean plus 3 standard deviations of the background dendritic cytoplasm signal sampled from any random 5 neurons. Identical background measurements and intensity thresholds were employed in all treatment groups.

To estimate neurite integrity, MAP2 staining or ectopic expression of cytosolic GFP signals was performed on brain sections for subsequent analysis. For each condition, ten images were captured using an epifluorescence microscope (Nikon) from five biological replicates,

focusing on the prefrontal cortex and hippocampal regions (i.e. CA1-3). The images were binarised, and objects were analysed in FIJI (Version 1.57q) to determine the average neurite length in the aforementioned regions. Total neurite length was normalised to the values obtained from the corresponding control treatment groups.

For the VCD study (Fig. 2j and Supplementary Fig. 3i, j), five cohorts of mice were tested. The number of mice in each cohort showing signs of neurite loss—characterised by fragmentation and/or reduced MAP2 signal density within the same unit area compared to the vehicle control group—was counted (Supplementary Fig. 3i) to determine which CA region was most severely affected. Additionally, the normalised ratio of total neurite length in VCD-treated groups relative to the control treatment group was calculated and presented (Supplementary Fig. 3j).

Similarly, for the *Esrra* knockdown study (Fig. 5b and Supplementary Fig. 11c, d), four cohorts of mice were tested. The number of mice in each cohort exhibiting neurite loss—characterised by fragmentation and/or reduced densities of cytosolic GFP signals within the same unit area compared to the scramble shRNA control group—was counted (Supplementary Fig. 11c) in the prefrontal cortex and CA1-2 hippocampal regions. Additionally, the normalised ratio of total neurite length in the *Esrra* knockdown group relative to the scramble shRNA control group was calculated and presented (Supplementary Fig. 11d).

## Enzyme-linked immunosorbent assays (ELISAs) for cholesterol, oestradiol (E2), progesterone (P4) and follicle-stimulating hormone measurements (FSH)

The ERRα-bounded cholesterol levels were measured using ELISA kits purchased from Abcam (ab285242). Plasma levels of E2 were measured using the Mouse/Rat Oestradiol ELISA Kit purchased from Sigma-Aldrich (SE120084). Plasma levels of P4 were measured by the Mouse Progesterone ELISA kit from Novusbio (NBP2-60125-1Kit). Plasma levels of FSH were measured using the FSH (Rodent) ELISA Kit purchased from Abnova (KA2330).

## SDH/Complex II activity assay

Activities of SDH/Complex II were measured using kits purchased from BioVision/ Abcam (K660-100/ ab228560). Measurements were conducted by strictly adhering to the manufacturer's instructions.

## Cell mito-stress-test assay

Growth medium was removed, leaving 50 μl in each well, after which cells were rinsed twice with 400 μl of prewarmed assay medium, consisting of XF base medium supplemented with 25 mM glucose, 2 mM glutamine, and 1 mM sodium pyruvate buffered to pH 7.4. Following the rinses, 475 μl assay medium and 50 μl conditioned medium (525 μl final) were added to each well. Cells were then incubated in a 37 °C incubator without $CO_2$ for 1 h, after which prewarmed oligomycin, FCCP, rotenone, and antimycin A solutions were loaded into injector ports A, B, and C of the sensor cartridge to achieve final concentrations of 0.5 μM for oligomycin, 1 μM for FCCP, and 1 μM for rotenone and antimycin A. The cartridge was calibrated with the XF24 analyser, and the assay was performed as described in ref. 156.

The OCR was measured, and changes from baseline rates were tracked following the sequential addition of oligomycin, FCCP, and the rotenone + antimycin A mixture. This allowed for calculation of the basal respiration rate, proton leakage, maximal respiration, spare respiratory capacity, nonmitochondrial respiration, and ATP production[156] by Seahorse XF24 software version 1.8. At the end of each assay, cells were washed once with an excess of room temperature PBS, lysed with ice-cold radioimmunoprecipitation assay buffer (0.15 M NaCl, 1 mM EDTA, 1 mM EGTA, 0.5% sodium deoxycholate, 0.1% SDS, 1% Triton X-100, and 50 mM Tris-HCl, pH 8, with added protease and phosphatase inhibitor cocktails). Protein content was estimated by

the Bio-Rad DC protein assay (Bio-Rad), using a Molecular Devices SoftMax M3 microplate reader (Sunnyvale). Data from each well were then normalised for total protein content. Normalised Seahorse XF$^e$24 measurements were used to calculate a mean (±SEM) for each density group and each plate using XF24 software version 1.8 (Seahorse Bioscience).

## ATP colourimetric assay

The cellular contents of ATP were measured using kits purchased from BioVision/ Abcam (K354-100/ ab83555).

## Live cell imaging of intracellular ATP dynamics

The BioTracker ATP-Red dye (Millipore, SCT045) is a live cell imaging dye for detecting cellular ATP specifically localised to mitochondria. According to the manufacturer's protocol, 10 μM ATP-Red dye was added to DIV21 primary neuronal cultures pre-treated with designated treatments. After an incubation time for 30 min at 37 °C, excessive dyes were washed away with PBS buffer. Neurons were placed in a custom chamber which was continuously perfused with a bathing solution (2 ml/min; 32–34 °C) containing 120 mM NaCl, 3 mM KCl, 2 mM CaCl2, 1 mM MgCl2, 3 mM NaHCO3, 1.25 mM NaH$_2$PO$_4$, 15 mM Hepes, 5 mM glucose, 0.2 mM pyruvate, and 0.5 mM GlutaMax, adjusted to pH 7.4. We determined a baseline reporter signal by imaging a resting cell for at least 25 min. After ensuring a stable signal, 50 μM glutamate was applied for 30 s to strongly excite the neurons. Following the depolarisation, the cell would need considerable amounts of ATP to drive the Na+/K+-ATPase (sodium pump) to be able to restore the resting ion gradients. We followed the drop in ATP and its subsequent restoration by the mitochondria for 30–45 min after the glutamate stimulus. Time-dependent changes of dye signal intensities (i.e. Excitation 510 nm; Emission 570 nm) were live captured by a TCS SP8 confocal microscope (Leica Microsystems Inc.).

## Live cell apoptosis assay

The CellEvent™ Caspase-3/7 Red dye (ThermoFisher, C10430) is are fluorogenic substrate for activated caspase-3/7 in live cells during apoptosis. DIV21 primary neuronal cultures pre-treated with designated drug or AAV shRNA or lentiviral transduction treatments (for GFP signal that indicates neuronal cell morphology) were subjected to glutamate stimulus as mentioned above (in the section: Live cell imaging of intracellular ATP dynamics). After washing away the stimulus, 10 μM of the Caspase-3/7 Red dye was added to the culture. At 4 h after the glutamate stimulus, cells were live imaged for the Caspase-3/7 Red dye (i.e. Excitation 590 nm; Emission 610 nm). GFP signals resulting from AAV or lentiviral vectors were also captured by a TCS SP8 confocal microscope (Leica Microsystems Inc.).

## Luciferase assay for nuclear activities of ERRα

Nuclear activities of endogenous ERRα were analysed by the pGL-3xERRE-luciferase reporter system (Addgene #37851)[157,158]. To normalise transfection efficiency in the reporter assays, cells were co-transfected with the pRL-TK plasmid, which contains a functional Renilla luciferase gene cloned downstream of a herpes simplex virus thymidine kinase promoter (Promega). The assay was carried out as described above with the pGL-Basoc as a negative control. Luminescence was measured using a Bright-Glo™ Luciferase Assay System (Promega) on a luminometer (Berthold Technologies) and normalised to the control Renilla Luciferase signal. Luciferase activity was calculated against the negative control signals, and fold differences were compared among groups in separate assays.

## Electrophysiology
### In the primary cortical neuronal culture
**mEPSCs and mIPSCs.** These were recorded according to previously published procedures[159]. The cultures were transferred to a recording

chamber placed in a Nikon inverted microscope and washed with standard recording medium containing 129 mM NaCl, 4 mM KCl, 1 mM MgCl$_2$, 2 mM CaCl$_2$, 10 mM glucose, 4-(2-hydroxyethyl)-1-piperazineethanesulfonic acid (HEPES) 10, pH was adjusted to 7.4 with NaOH, and osmolarity to 320 mOsm with sucrose. TTX (0.5 μM) and picrotoxin (20 μM) were also added to this medium for the recording of spontaneous mEPSCs. Neurons were recorded at room temperature with patch pipettes containing 140 mM K-gluconate, 2 mM NaCl, 10 mM HEPES, 0.2 mM ethyleneglycol-bis(2-aminoethylether)-N,N,N′,N′-tetraacetic acid, 0.3 mM Na-GTP, 2 mM Mg-ATP, 10 mM phosphocreatine and pH 7.4, having a resistance in the range of 6–12 MΩ. Miniature "inhibitory" synaptic currents (mIPSCs) were recorded with a patch pipette where CsCl replaced K-gluconate and with the extracellular recording medium containing DNQX (20 μM) and DL-2-Amino-5-phosphonopentanoic acid (APV) (50 μM). Signals were amplified with Axopatch 200 A (Axon Instruments Inc., Foster City, CA) and were stored on an IBM PC. PClamp analysis software was used for the off-line analysis of voltage/current protocols. For the analysis of mEPSCs, a 600-Hz low-pass filter was first applied (in Clampfit analysis), and the events were then analysed using Minianalysis software, with a threshold set at 9 pA currents.

### In hippocampal slices
**Slice preparation.** Mice were anaesthetised with isoflurane and subjected to cardiac perfusion with an ice-old oxygenated (95% O$_2$/5% CO$_2$) choline chloride-based cutting solution, containing: 120 mM choline chloride, 2.5 mM KCl, 7 mM MgCl$_2$, 0.5 mM CaCl$_2$, 1.25 mM NaH$_2$PO$_4$, 5 mM sodium ascorbate, 3 mM sodium pyruvate, 26 mM NaHCO$_3$, and 25 mM glucose. The mice were rapidly decapitated, and their brains were removed quickly and placed in a cutting solution. Hippocampal slices of 300 μm thickness were prepared with VT1000S Vibratome (Leica Microsystems). Slices were transferred to a storage chamber containing a cutting solution at 34 °C for 15 min, then translocated to the incubator with artificial cerebrospinal fluid (ACSF) (124 mM NaCl, 2.5 mM KCl, 2 mM MgSO$_4$, 2.5 mM CaCl$_2$, 1.25 mM NaH$_2$PO$_4$, 26 mM NaHCO$_3$, and 10 mM glucose) at room temperature (25 ± 1 °C) for at least 1 hr before recording.

**mEPSCs and mIPSCs.** Slices that are ready for recording were placed in a recording chamber superfused (2 ml/min) with ACSF at 32–34 °C. Pyramidal neurons in CA1 were visualised with infra-red optics using an upright fixed microscope equipped with a 40 × water-immersion lens (FN1, Nikon) and a CCD (Charge-Coupled Device) monochrome video camera (IR-1000, DAGE-MTI). Putative shRNA transduced neurons located reasonably deep within the slices (50-100 μm), which ensured better preserved morphological integrity, were further identified and selected under epifluorescence illuminations as those expressing EGFP. Patch pipettes (resistance of 3–5 MΩ) were prepared by a horizontal pipette puller (P-1000; Sutter Instruments).

For mEPSC recording, the selected EGFP-expressing pyramidal neurons were held at -70 mV in the presence of 20 μM bicuculline and 1 μM TTX, with the pipette solution containing: 125 mM K-gluconate, 5 mM KCl, 10 mM HEPES, 0.2 mM EGTA, 1 mM MgCl$_2$, 4 mM Mg-ATP, 0.3 mM Na-GTP and 10 mM phosphocreatine (pH 7.35, 290 mOsm). For mIPSC recording, pyramidal neurons were held at −70 mV in the presence of 20 μM CNQX, 50 μM DL-AP5 and 1 μM TTX, with the pipette solution containing: 130 mM CsCl, 10 mM HEPES, 0.2 mM EGTA, 1 mM MgCl$_2$, 4 mM Mg-ATP, 0.3 mM Na-GTP, 10 mM phosphocreatine and 5 mM QX314 (pH 7.35, 290 mOsm).

**Paired-pulse ratio recording.** EPSCs were evoked by stimulating the Schaffer collaterals (SC)-CA1 pathway at a holding potential of −70 mV in the presence of 20 μM bicuculline, with the pipette solution containing: 125 mM Cs-methanesulfonate, 5 mM CsCl, 10 mM HEPES, 0.2 mM EGTA, 1 mM MgCl$_2$, 4 mM Mg-ATP, 0.3 mM Na-GTP, 10 mM

phosphocreatine and 5 mM QX314 (pH 7.35, 290 mOsm). The first evoked EPSC was adjusted with an amplitude between 100 and 150 pA. The ratio was defined as the fraction of EPSC2/EPSC1 amplitudes.

**fEPSPs.** LTP studies were performed on hippocampal slices from test animals of the VCD-induced AOF treatment paradigm. Test animals were anaesthetised with isoflurane, decapitated, and their brains were rapidly removed in pre-ice cold ACSF [120 mM NaCl, 2.5 mM KCl, 1.25 mM $NaH_2PO_4$, 26 mM $NaHCO_3$, 2 mM $MgSO_4$, 10 mM D-glucose, 2 mM $CaCl_2$ with 95% $O_2$, 5% $CO_2$ (pH 7.4)]. Transverse hippocampal slices were cut at a thickness of 350 μm and placed on infusion chambers in ACSF. The fEPSPs were recorded from the CA1 by an extracellular borosilicate glass capillary pipette (resistance of 3–5 M Ω) filled with ACSF. Stimulation of Schaffer collaterals from the CA3 region was facilitated by a bipolar electrode. Signals were amplified using a MultiClamp 700 B amplifier (Axon), digitised using a Digidata 1440 Data Acquisition System (Axon) with 1 kHz low-pass filter and 2 kHz high-pass filter, and analysed using Clampex 10.7 software.

Data were acquired using the MultiClamp 700B amplifier and 1550 A digitiser (Molecular Devices). Series resistance was monitored throughout the experiments, and cells included in the analysis were of resistance <20 MΩ. Neurons would be rejected if membrane potentials were more positive than −60 mV, if the ratio of Rin to Rs <5 or if series resistance fluctuated >20% of initial values. Data were filtered at 3 kHz and sampled at 10 kHz. Data were analysed with GraphPad Prism software. The mEPSCs and mIPSCs were manually analysed by Mini Analysis (Synaptosoft). The paired-pulse ratio was analysed by Clampfit (Molecular Devices). Data were presented as mean ± SD.

**ChIP assay**
The relevance of ERα or ERRα transcription factors to the target genes of interest was first queried on their respective ENCODE and ChEA datasets located on Harmonizome[160]. The binding of these two transcription factors at the promoter regions of the target genes was then validated on the ChIP-Atlas, and the gross genome binding regions were visualised on the IGV genome browser[161,162]. Conserved ERα or ERRα binding sites located within regions spanning the 4 kb centred on their transcription starting sites [−2kb, +2 kb] were predicted by the JASPAR database (https://jaspar.elixir.no)[163] with a relative profile score threshold 75%.

A total of $10^7$ primary cortical neurons subjected to different combinations of treatment (according to their corresponding figures) for 120 h were cross-linked with 3.7% formaldehyde (Sigma) at room temperature for 10 min. Cells were incubated with 0.125 M glycine to terminate cross-linking, washed twice with PBS and lysed with SDS nuclear lysis buffer (1% SDS, 10 mM EDTA and 50 mM Tris-HCl, pH 8.1) for 10 min on ice. Sonicated lysates were diluted in ChIP dilution buffer (0.01% SDS, 1.1% Triton X-100, 1.2 mM EDTA, 167 mM NaCl and 16.7 mM Tris-HCl, pH 8.1), and incubated with 10 μg of rabbit IgG (Santa Cruz Biotechnology) and Dynabeads Protein G (ThermoFisher, 10003D) overnight at 4 °C with gentle shaking. The cleared supernatants were mixed with either 100 μl of anti-ERα (ThermoFisher, MA1-27107) or anti-ERRα (ThermoFisher, PA5-28749) antibody or with pre-immune rabbit IgG overnight at 4 °C. Antibody–protein–DNA complexes were co-precipitated with Dynabeads Protein G. Protein–DNA conjugates were eluted from the bead complexes with elution buffer (100 mM $NaHCO_3$ and 1% SDS) for 30 min. Cross-links were reversed in 5 M NaCl. RNA and protein were removed by incubation first with 10 μg DNase-free RNase-A at 37 °C for 1 h, and then with 20 μg proteinase K (Sigma) at 50 °C for 4 h. DNA was recovered by phenol–chloroform extraction and ethanol precipitation. DNA fragments encompassing the ERα or ERRα binding regions of the predicted target gene promoters were amplified using 35 cycles of PCR at 94 °C for 30 s, 55 °C for 30 s and 72 °C for 30 s. All amplified products were resolved on a 2% agarose gel. Primers (5′ to 3′) at the target gene promoters are listed in Supplementary Data 2. Full scan blots can be found in "Source Data.

**Quantification procedures and statistical analysis**
For each experiment, no statistical methods were used to predetermine sample sizes, but our sample sizes were similar to those reported in recent publications[138,141]. Data distribution was first tested for normality prior to statistical analysis. All samples were analysed, and the data collected were blinded to the experimental conditions. All experiments were performed on at least three independent occasions. Quantification of cellular morphology parameters was performed in a blinded manner. Analyses of the qPCR data were performed on Prism 8.0. Differences between groups were analysed using a two-tailed unpaired Student's $t$-test (for two groups) or a one-way ANOVA (for more than three groups) for normally distributed data or using a Wilcoxon signed-rank test or a Kruskal–Wallis test for skewed data, respectively. Pathway analysis was performed on Enrich R. All statistical analyses were performed using GraphPad Prism 8.0 software package. $p < 0.05$ was considered to indicate statistical significance. Exact $p$-values are provided in each figure panel, except when they are smaller than 1e-15 (the lower limit of the software), in which case '<1e-15' is reported.

**Reporting summary**
Further information on research design is available in the Nature Portfolio Reporting Summary linked to this article.

## Data availability
The bulk RNA-seq data generated in this study have been deposited in GEO Omnibus under the accession code GSE279885 (https://www.ncbi.nlm.nih.gov/geo/query/acc.cgi?acc=GSE279885). The metabolomics data generated in this study have been included in the Source Data file. Datasets utilised in this study were accessed with permission from the Accelerating Medicines Partnership® Program for Alzheimer's Disease (AMP® AD) platform under a signed agreement. The datasets were all generated from the ROSMAP cohort, these included the single cell transcriptomic data from the prefrontal cortex region (snRNA-seqPFC_BA10) (Syn18485175)[52] (https://www.synapse.org/Synapse:syn18485175); bulk RNA-seq data of dorsolateral prefrontal cortex (DLPFC), posterior cingulate gyrus (PCG), and anterior cingulate (AC) regions (syn3388564)[134] (https://synapse.org/Synapse:syn3388564); and the non-targeted metabolomics data form DLPFC region (syn3157322) (https://www.synapse.org/Synapse:syn3157322). Key clinical and pathological features of de-identified subjects included in these studies can be found in Supplementary Data 1. The definitive disease status (i.e. LOAD vs ND) of samples in the snRNA-seq dataset was provided by the original published paper and was adopted in this study[52]. For the bulk transcriptomics and metabolomics samples, definitive disease status was defined based on the Ward D2 hierarchical clustering method with reference to the CERAD score (i.e. a semi-quantitative measure of neuritic plaques); Braak staging score (i.e. semiquantitative measure of neurofibrillary tangles); Cogdx score (i.e. a clinical consensus diagnosis of cognitive status at time of death) and the Dcfdx score (i.e. a clinical diagnosis of the cognitive status). The ChIP-seq dataset of ER, PR and p300 in MCF7 breast cancer cell line treated with oestrogen with or without progestins was extracted from GSE68355 (inside the Super Series GSE68359)[13] (https://www.ncbi.nlm.nih.gov/geo/query/acc.cgi?acc=GSE68359). The protein structures used for docking analyses in this study can be found in the RCSB Protein Data Bank, including: 1XB7 for the X-ray structure of ERRα LBD in complex with a PGC1α peptide (https://doi.org/10.2210/pdb1XB7/pdb); 7E2E for the crystal structure of the ERRα LBD in complex with an agonist DS45500853 and a PGC-1α peptide (https://doi.org/10.2210/pdb7E2E/pdb) and 2PJL for the crystal structure of human ERRα in

complex with a synthetic inverse agonist (https://doi.org/10.2210/pdb2PJL/pdb). Source data are provided with this paper.

## Code availability

All original codes for each figure can be found at https://zenodo.org/records/17424482.

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

## Acknowledgements

The authors would like to gratefully acknowledge the following parties. The results published herein are in part based on data obtained from the AD Knowledge Portal related to the ROSMAP metadata sets. Details of the relevant datasets used in our analyses can be found in the Data availability statement. The human brain tissue samples utilised for immunohistology analyses were obtained from the University of Maryland Brain and Tissue Bank, which is a Brain and Tissue Repository of the NIH NueroBioBank, as requested by Dr Karl Herrup. The work was supported, in part, by grants from the following: The National Natural Science Foundation-Excellent Young Scientists Fund 2020 (Ref: 32022087) and Young Scientist Fund 2023 (32300643) (K.H.-M.C.); the Hong Kong Research Grants Council (RGC)-General Research Fund (GRF) (PI: ECS24107121, PI: 16100219 and PI: 11410824); CUHK-Improvement on Competitiveness in Hiring New Faculties Funding Scheme (PI: ref: 133), CUHK-Gerald Choa Neuroscience Institute (PI: Ref. 8425032) and School of Life Sciences Start-up funding to K.H.-M.C. All the funders played no role in the study design, data collection and analysis, decision to publish, or preparation of the manuscript.

## Author contributions

J.K.-L.S. performed the in vivo and in vitro studies and bioinformatics analyses. A.Z.P. performed the in vitro experiments. G.C.-N.W. performed the in silico simulations and docking studies. D.W. performed the bioinformatics analyses. R.P.H. contributed plasmid reagents, bioinformatics advice and technical support. K.H. contributed human brain samples. K.H-M.C. conceptualised and designed the research, conducted the majority of the in vivo and in vitro experiments and bioinformatics analyses, analysed all the data and wrote the paper.

## Competing interests

The authors declare no competing interests.
