## [Transparent Peer Review file · Nature Communications]

Perimenopausal State Oestradiol to Progesterone Imbalance Drives Alzheimer's Risk via $ERR\alpha$ Dysregulation and Energy Dyshomeostasis

Corresponding Author: Professor Kim Chow

Version 0:

Reviewer comments:

Reviewer #1

(Remarks to the Author)

Major Comment for Authors:

I am very gratified by your major and effective efforts to apply known clinical phenomena in midlife women to the question of why later life AD is more common in women. As a clinician, some of your methods and metabolism are not in my daily life, but you come back to human data, and your mouse accelerated perimenopause model and do tests there. I'm especially gratified because this well-supported notion of perimenopausal estrogen excess and declining progesterone is totally ignored by usual gynecological thought. Congratulations!

Specific necessary changes:

1. In discussing the ovarian hormonal imbalance in perimenopause, it is preferable to call it "higher Estradiol, lower Progesterone imbalance" (or $E2 < P4$) since "estrogen dominance" is a term widely used in the alternate care community to describe both menstrual cycles with ovulatory disturbances and also perimenopause. The term estrogen dominance is thus scoffed at by physicians of many stripes.
2. When first describing the hormonal changes of perimenopause it would be useful to quote the first demonstration and recognition in humans that estrogen is higher and progesterone lower. It is in the pioneering paper by Nanette Santoro (1). Another paper showing similar data is rarely quoted and it would be good to have it up front because it also shows both E2 and P4 changes (4).
3. The concept of balance between estradiol and progesterone is an important one. If you wish, I've done a review on this issue, including the difference in amounts and units for the two hormones that you might quote (6).
4. Finally, we now have published a randomized controlled trial of daily oral micronized progesterone for treatment of perimenopausal night sweats and hot flushes (5). It showed significantly improved night sweats, intensity of daytime hot flushes, improved sleep quality and decreased interference of perimenopause with daily life, although the primary outcome was not statistically significant despite almost double the size of menopause VMS trials. It importantly also showed no increase in depression.

Reference List

1. Santoro N, Rosenberg J, Adel T, et al. Characterization of reproductive hormonal dynamics in the perimenopause. *Journal of Clinical Endocrinology and Metabolism* 1996;81:4:1495-501.
4. O'Connor KA, Ferrell RJ, Brindle E, et al. Total and Unopposed Estrogen Exposure across Stages of the Transition to Menopause. *Cancer Epidemiol Biomarkers Prev* 2009;18(3):828-36.
5. Prior JC, Cameron A, Fung M, et al. Oral micronized progesterone for perimenopausal night sweats and hot flushes a Phase III Canada-wide randomized placebo-controlled 4 month trial. *Sci Rep* 2023;13(1):9082. doi: <https://www.nature.com/articles/s41598-023-35826-w> [published Online First: 20230605]
6. Prior JC. Women's Reproductive System as Balanced Estradiol and Progesterone Actions—a revolutionary, paradigm-shifting concept in women's health. *Drug Discovery Today: Disease Models* 2020;32:31-40. doi: <https://doi.org/10.1016/j.ddmod.2020.11.005>

Reviewer #2

(Remarks to the Author)

Sun et al have used a combination of human and animal studies to investigate the molecular and metabolic changes that result from disruption to the relative levels of key menopause-linked hormones (estradiol (E2) & progesterone (P4)) and how these changes can impact upon neuronal physiology and cognition. They convincingly display that such changes are significantly more impactful in female rather than male subjects. They link these changes to the pathogenesis of idiopathic Late Onset Alzheimer's Disease (LOAD) through the use of patient samples from the ROSMAP cohort and the 3xTg mouse model of familial AD, suggesting that this may underly the greater susceptibility for females to develop LOAD.

Specifically, they utilise transcriptomic and metabolomic approaches to show that female LOAD patients in the ROSMAP cohort exhibit more pronounced gene expression and metabolite changes than their male counterparts and that inducing accelerated ovarian failure (AOF) in WT mice recapitulates this phenotype to a remarkably similar degree. Importantly, AOF mice were shown to have impaired spatial learning and working memory through a behavioural test battery, with decline shown to be strongly correlated with the degree of hormonal disruption (as measured by E2:P4 ratio). These deficits were tied to reduced hippocampal excitatory postsynaptic potentials and diminished long-term potentiation, indicating that these transcriptomic/metabolomic changes can have downstream impacts that impair neuronal function.

The common outcome from both human and mouse experiments was disrupted expression of the targets of the transcriptional regulator estrogen-related receptor alpha (ERR1/ α), which they show to have diminished nuclear localisation following hormonal disruption. They therefore focus in on this receptor for proceeding experiments, demonstrating a reduction in the interaction between cholesterol and ERR α and hence disrupted cholesterol metabolism. To assess these changes in more detail they specifically knock-down the expression of ERR α in cortex and hippocampus of WT mice using shRNA, demonstrating a knock-on impact of reduced production of N-acetyl-aspartyl-glutamate (NAAG), a key regulator of neuronal activity. NAAG levels were also found to be correlated with cognitive decline in female LOAD patients. To more deeply probe the functional effects of these changes, patch clamp electrophysiology was performed in primary neuronal cultures exposed to ERR α targeted shRNAs or the ERR α antagonist XCT-790. This resulted in increased frequency of excitatory miniature postsynaptic potentials (mEPSCs) and deficits in ATP recovery rate that predispose neurons to undergo apoptosis, which the authors propose could underly the female-biased predisposition to LOAD.

Finally in an attempt to tie these changes to the pathophysiology of LOAD, the AOF paradigm was applied to 3xTg mice. This recapitulated the transcriptomic and behavioural effects observed in AOF WT mice, but they additionally demonstrate that chronic administration of P4 can effectively rescue these phenotypes.

I believe this manuscript provides compelling and novel insight and evidence into the downstream impacts that hormonal imbalance can have upon brain physiology and function. The transcriptomic, metabolomic and protein structure experiments used to derive these conclusions appear to be sound and appropriate (I note that a thorough critique of these techniques is beyond the scope of this reviewer). The electrophysiological techniques have been performed correctly and interpreted in a mostly appropriate manner. However, I believe that some minor outstanding points must be addressed before the work can be considered for publication by Nature Communications.

- Choice of using 3-month-old 3xTg mice

In an attempt to tie their results to the pathogenesis of LOAD, the authors apply their AOF paradigm to young 3xTg mice (3 month). They acknowledge that this timepoint is before the accumulation of AD related pathology occurs in this mouse line (which starts at approximately 6 months). They go on to show that this approach reproduces the phenotypes observed in C57BL6 animals. I would argue that without the added variable of the presence of AD pathology, this is simply a repeat confirmation of the C57BL6 experiment and provides no additional insight into the pathogenesis of LOAD. The authors need to at least add a clarifying statement for their rationale for choosing this time point. Alternatively (and preferably) an assessment of whether the AOF paradigm accelerates the appearance or increases the burden of amyloid or tau pathology in 3xTg mice e.g. through 6e10 and/or AT8 immunohistochemistry, would more definitively tie the hormonal imbalances described to LOAD pathogenesis.

Similarly, based on their work chronically administering P4 to 3xTg through implantation of minipumps, they conclude that P4 may be an effective supplementary therapy to address the impacts of disrupted hormonal imbalance upon neuronal physiology. While the rescue effects described are impressive, I believe this case could be strengthened by comparing data from AOF WT control mice undergoing the same treatment to that of the 3xTg mice. A relatively stronger degree of rescue in the 3xTg mice would again provide strong evidence of a direct interaction between hormonal dysregulation and LOAD. As it stands the use of the 3xTg model seems incidental rather than being more informative than the same experiment in WT mice would have been.

Finally, the 3xTg line is considered a model of early-onset familial AD rather than for LOAD. None of the transgenic mutations it carries are associated with the idiopathic, late-onset form of the disease. The authors should make this caveat clear in the manuscript. An interesting addition would be to repeat the AOF paradigm in a mouse line harbouring high genetic risk factors for LOAD specifically e.g. homozygous knock in APOE4 mice (available through JAX - <https://www.jax.org/strain/027894>), which would again provide more direct insights on the pathogenesis of LOAD.

- Confirmation of presynaptic effect of ERR α knock down.

The authors have performed patch clamp electrophysiology in primary cell cultures treated with ERR α targeted shRNA, demonstrating that this causes a specific increase in the frequency of miniature excitatory postsynaptic currents (mEPSCs). They conclude that this is due to an increase in presynaptic glutamate release driven by elevated levels of VGlut1, given that they observe no increases in the density of NR2B, which is used as a proxy marker for postsynaptic sites. While it is possible that this conclusion is correct, I don't believe the results presented fully exclude other possibilities. NR2B is not the best postsynaptic marker as not all excitatory synapses contain these receptors, and not all NMDARs are located at postsynaptic sites. A ubiquitous excitatory postsynaptic marker such as Homer1 or PSD95 would therefore be more appropriate for this analysis. Additionally, no attempt was made to colocalise presynaptic and postsynaptic signals in order to ensure that only mature functional synapses were included in the analysis. Repeating this analysis with these factors in place would give greater confidence to the conclusion that the increase in mEPSCs is driven by increased presynaptic release of glutamate.

Additionally, this conclusion can be confirmed through a relatively simple additional experiment consisting of paired-pulse ratio recordings. While not feasible in the primary cell cultures, the authors could utilise the hippocampal slice preps described elsewhere in the paper to compare the effect on Esrra and scrambled shRNA injected animals. A result displaying reduced paired-pulse facilitation in Esrra shRNA exposed neurons would definitively confirm the finding of enhanced presynaptic release probability.

- Quantification of neurite integrity/loss

The authors claim that the AOF paradigm induces a reduction in neurite integrity in mice that developed an imbalanced E2:P4 ratio (Fig.2 J) and that injection of Esrra shRNA causes neurite loss (Fig.5b). However, the only evidence given for this are individual representative images. In order to make these claims the authors should quantify a measure of neurite integrity across cohorts of mice and perform appropriate statistical comparisons, rather than relying on a single images from individual animals.

If these points are appropriately addressed, then I would recommend a revised manuscript for publication.

Reviewer #3

(Remarks to the Author)

This is an ambitious and interdisciplinary study that intends to connect the estradiol/progesterone imbalance occurring in perimenopause with the elevated vulnerability of women to Alzheimer's disease (AD) dementia. Here, the authors examined the transcriptional and behavioral effects and the underlying molecular mechanisms by which estradiol/progesterone imbalance (i.e, estrogen dominance), as that occurring in perimenopause, leads to memory loss in an accelerated ovarian failure (AOF) mouse model, and how the involved mechanisms may be linked to elevated susceptibility of dementia in women. Integrating the entire dataset into a comprehensive hypothesis systems model is not straightforward despite the variety of experimental models (mice, cultured neurons, human brain/transcriptomic data) and employed state-of-the art technical methodologies (sn-RNAseq data, metabolomics, AAV-mediated gene modulation).

The results obtained from the perimenopause mouse model fits with a model that estradiol/progesterone imbalance leads to memory and synaptic plasticity deficits by dysregulating essential metabolic pathways (i.e, oxidative phosphorylation, thermogenesis, TCA cycle). The study conclusively demonstrates that estradiol-mediated ER α signaling enhances genes encoding key metabolic enzymes (Pdha1, Dhcr24, Cyp51...) and related metabolites (pyruvate, acetyl-CoA, mevalonate...). The results suggest that disrupted ER α signaling may impact glycolysis and cholesterol biosynthesis, which potentially might impair cholesterol-bound ERR α -PGC1 α signaling in neurons of AOF mice and women with AD. However, direct evidence/s that altered ER α pathway disrupts ERR α -PGC1 α signaling (nuclear translocation, gene expression...), particularly in the context of AD (AD mice and brains) is lacking. In addition, the connection between estrogen dominance and the increased risk of dementia in postmenopausal women is not sufficiently addressed considering the challenges in translating results from the AOF mouse model to human transcriptomic and pathological findings (ROSMAP). Using a controlled non-dementia human cohort with both pre- and postmenopausal participants would have been useful to validate and reinforce some transcriptomic results/conclusions. In addition, the transcriptomic discrepancies on biological/metabolic pathways affected by genetic ERR α inactivation and perimenopausal mouse models do not seem to support the hypothesis.

Another critical issue is the diversity of brain regions analyzed (cortex vs hippocampus), some experiments with a very low number of samples (e.g, Fig. 1f, n=3) (human frontal cortex -Fig 1-, hippocampus or total cortex of AOF mice-Figs 2,3,4..), as well as the lack of cell-specific effects (bulk vs snRNA sequencing, metabolomics). It may be a good option to focus specifically in the estrogen dominance results and leave out AD. I also suggest several changes and considerations (see below) that could enhance the quality of the study. The abstract is quite confusing and should be more self-explanatory, for instance by describing, when necessary, the human samples and specific mouse models (age/sex). In conclusion, the study has several weaknesses that limit likely the global impact of the findings.

Specific comments

Fig.1. ERR α -regulated bioenergetic network in AD

Main issue: No clear quantification of ERR α levels/localization in AD brains.

Fig 1b/c. Comparison of transcriptome changes (i.e, DEGs) between AD females vs AD males are missing.

Fig 1e/f. To confirm Mathys et al (2019) results (Fig. 1e), levels of ESRRRA and PPARGC1A mRNA and protein should be quantified, at least in excitatory and inhibitory neurons, at cell-specific spatial resolution. Thus, ERR α and PGC1 α protein levels in excitatory and inhibitory neurons of AD and ND brains can be analyzed by double immunostaining with excitatory and inhibitory markers. Fig. 1f (lines 173-174): there is not a clear description how nuclear ERR α in MAP2+cells was quantified without using a nuclear marker. The authors should show the cytosolic, nuclear and/or nuclear/cytosolic ERR α levels in multiple ND vs AD male/female brain samples. Based in my own experience, n=3 seems is a very low n for human samples.

Fig. 2. Mouse model of perimenopause/AOF

Fig 2f. Latencies during 6 day training in the MWM should depicted to conclude that treatment causes spatial learning deficits.

Fig. 2j. Lines 212-218. The "neurite loss" caused by VCD treatment on neuronal dendritic morphology (MAP2 staining) should be quantified, and the specific hippocampal region/s analyzed should be indicated. Fig. 2k. State whether the "n" refers to number of slides, animals...

Fig. 3. Bulk transcriptome changes in the cortex of the AOF mouse model
Main issue: low number of samples and the use of total brain cortex.

Fig. 3a-d. a. The low number of samples (n=3-4) for the bulk transcriptomic analyses of the AOF model is quite moderate, which could be a concern considering the apparent transcriptomic variability of VCD samples compared to controls (according to PCA plot of Fig 3a). c-d. Besides metabolic pathways (oxidative phosphorylation, TCA...), the majority of affected pathways are related to neurodegenerative diseases raising the possibility that VCD treatment (>2.5 pg/ng) cause neurodegeneration leading to dendritic and/or neuronal loss and synaptic plasticity impairments (see Fig. 2j-k).

Fig. 3f. As in Fig. 1f, it is not clear how and in which cortical region/s) nuclear ERR α and PGC1 α levels are quantified. It is important to examine whether nuclear ERR α and PGC1 α change in excitatory and inhibitory neurons by using colocalization studies.

Fig. 3g. Co-immunoprecipitation experiments indicate that VCD treatment (>2.5 pg/ng) decreases ERR α /PGC1 α interaction. There is a clear decrease of PGC1 α protein levels that may affect this interaction. Input ERR α levels should be also depicted in the graph. As explained in the text, PGC1 α may be susceptible to degradation (line 244), so PGC1 α transcript levels may be quantified to confirm or not this possibility.

Fig 3i (lines 265-267) and Fig. 4k. I could not find how the ERR α -attached cholesterol is measured.

Fig. 4. ER α regulation of ERR α -PGC1 α signaling

Main issue: Lack of cell-specific metabolomics and no direct evidences that altered ER α pathway disrupts ERR α -PGC1 α signaling in neurons, and is altered in AD.

Based on the current data, the connection between ER α regulation of ERR α -PGC1 α signaling is not straight forward. Bulk transcriptomics (Figs. 3a,b) and metabolomics (Figs. 4a-d) showed downregulated genes clustered mainly to mitochondrial network (oxidative phosphorylation, TCA cycle...) and clear reduction of many metabolites of glycolysis, mevalonate and cholesterol biosynthesis pathways in total cerebral cortex of AOF mice. The main issue here is the lack of cell-specificity of metabolomic results: global metabolites/enzymes changes it does not mean cell-specific (i.e., neurons) changes.

Fig. 4e/h shows that primary cortical neurons treated with estradiol (E2) (effect blocked by a ER α antagonist) upregulates specific metabolites (pyruvate, acetyl-CoA, acetoacetyl-CoA, mevalonate) and their related enzymes (Pdha1, Dhcr24, Cyp51), a result reinforced by recruitment of ER α to these gene promoters (Fig. 4d). Since these results do not directly link ER α to ERR α dysfunction, it is essential to determine whether gain and loss of ER α function directly affects ERR α signaling.

Fig. 4g. Changes of metabolic-related ER α target genes are associated with cognitive impairment, an effect more evident in females of the ROSMAP cohort. Are there sex-differences (male vs females) in the ER α target genes within specific cognitive score groups ?

Fig. 5. ERR α regulates the TCA-NAAG pathway

Main issue: Transcriptomic discrepancies between genetic ERR α inactivation and perimenopausal mouse models

The authors describe that ERR α inactivation in mouse hippocampus and frontal cortex (are male and/or female mice ?) affects negatively memory (lines 350-351; Supl Fig. 7) and TCA metabolites (Fig. 5g) similar to VCD mice (Supl Fig. 8). First, there is no clear evidence that this AAV transduces neurons of the mouse brain. Second, according to data presented in Supl. Fig. 7c (lines 350-351), there is not a clear effect of ERR α inactivation in learning/memory in the Y-maze (% alternation is similar for most mice) and MWM (latencies during days 1-6 and % target quadrant occupancy during the probe trial should be shown). Third, based on the ERR α inactivation experiments, it is clear that ERR α upregulates TCA pathway genes/metabolites (Fig. 5g) in contrast to downregulation of glycolysis/TCA/mevalonate/cholesterol pathways in VCD-treated mice (Fig. 4).

When comparing Figs. 3c and 5f, the general picture that emerges highly differs: the most relevant metabolic pathways

affected in the perimenopause mouse model are oxidative phosphorylation, TCA and neurodegenerative diseases, whereas $ERR\alpha$ inactivation affects TCA, glycolysis, pyruvate and aminoacid metabolism. According to the authors hypothesis presented in previous figures, we should expect disruption of glycolysis/TCA/mevalonate/cholesterol pathways by neuronal $ERR\alpha$ inactivation. How do the authors explain that neuronal $ERR\alpha$ inactivation does not affect mevalonate/cholesterol biosynthesis ?

If $ERR\alpha$ is disrupted in the brain of AD females (Fig. 1), how is possible that metabolomics KEGG enrichment analyses does not show affectation of mevalonate/cholesterol-related metabolites in AD females (ROSMAP cohort)?. On the contrary, the main affectation is on aminoacid metabolisms/urea/ammonia cycle/lipids (Figs. 5j-l).

Fig. 6. Inactivation of $ERR\alpha$ enhances spontaneous postsynaptic activity and negatively affects respiratory capacity and neuron viability

Main issue: It is challenging to reconcile how TCA/mitochondrial deficits and ATP depletion lead to increased spontaneous glutamate release, and this ultimately may contribute to AD vulnerability in women.

Fig 6a,b. Instead of primary neurons, it seems more logical and relevant for the entire study to measure excitatory/inhibitory neurotransmission and synaptic plasticity (i.e, LTP) alterations in brain slices of genetic $ERR\alpha$ -inactivated mice (Fig. 5). In summary, are the electrophysiology results in primary neurons replicated in vivo ?

Fig. 7. Effect of VCD in 3xTgAD mice.

The analyses of E2:P4 ratio, behavior and cortical transcriptomic/metabolomics (cholesterol homeostasis/metabolites) indicate that vehicle-treated C57BL/6 and 3xTgAD mice behave similarly, whereas results in C57BL/6- and 3xTg AD-VCD treated mice (Figs. 2e-g, 4b-d, 7b-d) are almost identical (Behavior: 30-40 sec latency to platform target in the MWM and 20-40 % alteration in the T-maze).

Considering that results of VCD-treated C57BL/6 vs VCD-treated 3xTg-AD mice are missing is not possible to know whether VCD-treatment causes differential effects in control and AD pathological conditions. For that, these two groups should be compared.

Does E2:P4 imbalance (> 2.5 pg/ng) affect amyloid and tau pathologies in 3xTg AD mice at this age ?

Reviewer #4

(Remarks to the Author)

This manuscript by Ka-Li et al. describes a set of interesting experiments in human samples and mouse models to elucidate the pathways linking estrogen dominance in women in midlife with increased risk of cognitive decline.

Related to metabolomics:

1. The metabolomic profiling was performed using a well-validated commercial platform through Human Metabolome Technologies.
2. Generally, it would be helpful to provide more information related to metabolomic data analyses to get a clearer picture on how to interpret the findings from the metabolomics data.

For example, in the analyses presented in Figs. 4A-B and 5E:

- How many metabolites were examined in total? Are the metabolites presented in the heatmap only the subset that were significant at $p < 0.05$?
- Did you consider an adjustment for multiple testing (e.g., FDR, Bonferroni, etc.)? Many of these associations may be due to chance (related to the high number of statistical tests performed). Please justify in the methods if multiple testing correction was not considered.

In the analyses to determine sex-specific effects (shown in panels in Fig. 5):

- What statistical test was used to determine effect modification by sex? Supplementary Table 11 shows significant metabolites stratified by sex, but it is difficult to know how to compare these results without an actual test of effect modification/interaction by sex (e.g., do males have fewer significant metabolites than females because of a true biological effect or because the sample size is lower in the male group?).

Version 1:

Reviewer comments:

Reviewer #1

(Remarks to the Author)

This review is now much improved.

There is a single, still needed correction (that may have arisen because of a misuse of English)

Line 799--change the word "enhanced" before night sweats to "decreased"

Reviewer #2

(Remarks to the Author)

The authors have added a substantial amount of work to the revised manuscript which has improved it substantially. I congratulate them on their excellent work. Their response to my comments 1-4 appropriately address the concerns I had. I now only have two minor concerns that can be addressed with small edits to the text and figures.

1. A legend identifying experimental groups is missing from figure 6D
2. I could not find a clear definition of how they decided whether a brain region was displaying 'neurite loss' for figures S3 and S11. This should be expanded upon in the methods

After these minor points are addressed I would support the publication of this manuscript

Reviewer #3

(Remarks to the Author)

The revised manuscript has been greatly improved. The authors have satisfactorily addressed most of my previous concerns and questions. These included the comments related to transcriptomic comparison of female vs male AD groups, gene/protein validation of specific genes, neuronal specificity, low number of samples in specific experiments, spatial memory results, AAV cell-specificity, etc...

I still think that the abstract can be further improved. It should be more self-explanatory, for instance describing, when necessary, the human samples and specific mouse models (perimenopause/VCD, 3xAD-Tg...), including ages and sex.

REVIEWER COMMENTS

Reviewer #1 (Remarks to the Author):

Major Comment for Authors:

I am very gratified by your major and effective efforts to apply known clinical phenomena in midlife women to the question of why later life AD is more common in women. As a clinician, some of your methods and metabolism are not in my daily life, but you come back to human data, and your mouse accelerated perimenopause model and do tests there. I'm especially gratified because this well-supported notion of perimenopausal estrogen excess and declining progesterone is totally ignored by usual gynecological thought. Congratulations!

Specific necessary changes:

Comment 1—In discussing the ovarian hormonal imbalance in perimenopause, it is preferable to call it “higher Estradiol, lower Progesterone imbalance” (or $E2 < P4$) since “estrogen dominance” is a term widely used in the alternate care community to describe both menstrual cycles with ovulatory disturbances and also perimenopause. The term estrogen dominance is thus scoffed at by physicians of many stripes.

Response 1— We are grateful to the reviewer for her valuable suggestion regarding the appropriate terminology to describe ovarian hormonal imbalance in perimenopause within the medical community. We have now incorporated the recommended changes, replacing the term "estrogen dominance" with "higher estradiol-lower progesterone imbalance" (all highlighted in yellow) in the revised manuscript.

Comment 2—When first describing the hormonal changes of perimenopause it would be useful to quote the first demonstration and recognition in humans that estrogen is higher and progesterone lower. It is in the pioneering paper by Nanette Santoro (1). Another paper showing similar data is rarely quoted and it would be good to have it up front because it also shows both E2 and P4 changes (4).

Response 2—In response to the reviewer's suggestion, we have added these two references to the second paragraph of the introduction section (References 23, 24, 41), where the concept of “higher estradiol-lower progesterone imbalance” was also introduced in the revised manuscript.

Comment 3—The concept of balance between estradiol and progesterone is an important one. If you wish, I've done a review on this issue, including the difference in amounts and units for the two hormones that you might quote (6).

Response 3— In response to the reviewer's suggestion, we have added this reference to the second paragraph of the introduction section, where we mentioned “the importance of a dynamic and

balanced ratio of estrogen and progesterone in the maintenance of women's health, particularly brain health" (Lines 138-139, Reference 41).

Comment 4—Finally, we now have published a randomized controlled trial of daily oral micronized progesterone for treatment of perimenopausal night sweats and hot flushes (5). It showed significantly improved night sweats, intensity of daytime hot flushes, improved sleep quality and decreased interference of perimenopause with daily life, although the primary outcome was not statistically significant despite almost double the size of menopause VMS trials. It importantly also showed no increase in depression.

Response 4—In accordance with the reviewer's recommendation, we have included this reference in the last paragraph of the discussion section, emphasizing the positive findings regarding the supplementation of natural progesterone in human clinical trials (Line 797-800, Reference 130).

Reviewer #2 (Remarks to the Author):

General Comment—Sun et al have used a combination of human and animal studies to investigate the molecular and metabolic changes that result from disruption to the relative levels of key menopause-linked hormones (estradiol (E2) & progesterone (P4)) and how these changes can impact upon neuronal physiology and cognition. They convincingly display that such changes are significantly more impactful in female rather than male subjects. They link these changes to the pathogenesis of idiopathic Late Onset Alzheimer's Disease (LOAD) through the use of patient samples from the ROSMAP cohort and the 3xTg mouse model of familial AD, suggesting that this may underly the greater susceptibility for females to develop LOAD.

Specifically, they utilise transcriptomic and metabolomic approaches to show that female LOAD patients in the ROSMAP cohort exhibit more pronounced gene expression and metabolite changes than their male counterparts and that inducing accelerated ovarian failure (AOF) in WT mice recapitulates this phenotype to a remarkably similar degree. Importantly, AOF mice were shown to have impaired spatial learning and working memory through a behavioural test battery, with decline shown to be strongly correlated with the degree of hormonal disruption (as measured by E2:P4 ratio). These deficits were tied to reduced hippocampal excitatory postsynaptic potentials and diminished long-term potentiation, indicating that these transcriptomic/metabolomic changes can have downstream impacts that impair neuronal function.

The common outcome from both human and mouse experiments was disrupted expression of the targets of the transcriptional regulator estrogen-related receptor alpha ($ERR1/\alpha$), which they show to have diminished nuclear localisation following hormonal disruption. They therefore focus in on this receptor for proceeding experiments, demonstrating a reduction in the interaction between cholesterol and $ERR\alpha$ and hence disrupted cholesterol metabolism. To assess these changes in more detail they specifically knock-down the expression of $ERR\alpha$ in cortex and hippocampus of WT mice using shRNA, demonstrating a knock-on impact of reduced production of N-acetyl-aspartyl-glutamate (NAAG), a key regulator of neuronal activity. NAAG levels were also found to be correlated with cognitive decline in female LOAD patients. To more deeply probe the functional effects of these changes, patch clamp electrophysiology was performed in primary neuronal cultures exposed to $ERR\alpha$ targeted shRNAs or the $ERR\alpha$ antagonist XCT-790. This resulted in increased frequency of excitatory miniature postsynaptic potentials (mEPSCs) and deficits in ATP recovery rate that predispose neurons to undergo apoptosis, which the authors propose could underly the female-biased predisposition to LOAD.

Finally in an attempt to tie these changes to the pathophysiology of LOAD, the AOF paradigm was applied to 3xTg mice. This recapitulated the transcriptomic and behavioural effects observed in AOF WT mice, but they additionally demonstrate that chronic administration of P4 can effectively rescue these phenotypes.

I believe this manuscript provides compelling and novel insight and evidence into the downstream impacts that hormonal imbalance can have upon brain physiology and function. The transcriptomic, metabolomic and protein structure experiments used to derive these conclusions appear to be sound and appropriate (I note that a thorough critique of these techniques is beyond the scope of this reviewer). The electrophysiological techniques have been performed correctly and interpreted in a mostly appropriate manner. However, I believe that some minor outstanding points must be addressed before the work can be considered for publication by Nature Communications.

General response: We appreciate the reviewer's feedback and constructive criticism. We have now thoroughly addressed each of the individual comments provided.

Choice of using 3-month-old 3xTg mice

Comment 1—In an attempt to tie their results to the pathogenesis of LOAD, the authors apply their AOF paradigm to young 3xTg mice (3 month). They acknowledge that this timepoint is before the accumulation of AD related pathology occurs in this mouse line (which starts at approximately 6 months). They go on to show that this approach reproduces the phenotypes observed in C57BL6 animals. I would argue that without the added variable of the presence of AD pathology, this is simply a repeat confirmation of the C57BL6 experiment and provides no additional insight into the pathogenesis of LOAD. The authors need to at least add a clarifying statement for their rationale for choosing this time point. Alternatively (and preferably) an assessment of whether the AOF paradigm accelerates the appearance or increases the burden of amyloid or tau pathology in 3xTg mice e.g. through 6e10 and/or AT8 immunohistochemistry, would more definitively tie the hormonal imbalances described to LOAD pathogenesis.

Response 1—In response to the reviewer's suggestions, we opted for the suggestion to assess if the AOF paradigm accelerates the appearance of AD-related pathologies in 3xTg mice. In the revised manuscript, we elaborated our investigation further with immunoblotting experiments using specific antibodies targeting human APP/A β (6E10) and pTau (AT-8) (**Revised manuscript: Fig.7i, Supplementary Fig.17c**), as the reviewer suggested. In the updated result, robust elevated levels of phosphorylated tau at S202/T205 recognized by AT-8 antibody was found specifically in VCD-treated 3xTg mice with E2:P4 ≥ 2.5 pg/ng (**Revised manuscript: Fig.7i, Supplementary Fig.17c**). The signal intensities of AT-8 in VCD-treated 3xTg mice with E2:P4 < 2.5 pg/ng, however, remained indifferent from the vehicle-treated 3xTg control and C57BL/6 groups at this age (**Revised manuscript: Fig.7i, Supplementary Fig.17c**), suggesting the status of E2:P4 ≥ 2.5 pg/ng is a risk that associates with tau hyperphosphorylation¹. Progesterone drug treatment analysis reversely echoed the importance of E2:P4 status to pathological tau phosphorylation in neurons (**Revised manuscript: Fig.7m-n, Supplementary Fig. 19c-d**). In 3xTg mice exposed only to VCD and drug vehicle control, signals of intraneuronal AT-8 were more robustly observed among those with the E2:P4 status ≥ 2.5 pg/ng (**Revised manuscript: Figure 7m-n, Supplementary Fig. 19c-d**), and such increment was prevented if these animals were instead put

on progesterone (**Revised manuscript: Fig.7m-n, Supplementary Fig.19c-d**). A similar pattern of changes to the levels of A β monomer, dimer and oligomer recognized by the 6E10 antibody were observed as well (**Revised manuscript: Fig.7i, Supplementary Fig.17c; Fig.7m-n, Supplementary Fig.19c-d**). We reasoned that the positive correlations between the E2:P4 \geq 2.5 pg/ng status with AD pathologies are potentially related to the sustained elevation of spontaneous, excitatory firing among the neurons in these brains, as illustrated in **Fig.6b-g**. The positive relationships between excessive excitatory firing and these pathological changes were also supported by the previous literature²⁻⁶.

Comment 2—Similarly, based on their work chronically administering P4 to 3xTg through implantation of minipumps, they conclude that P4 may be an effective supplementary therapy to address the impacts of disrupted hormonal imbalance upon neuronal physiology. While the rescue effects described are impressive, I believe this case could be strengthened by comparing data from AOF WT control mice undergoing the same treatment to that of the 3xTg mice. A relatively stronger degree of rescue in the 3xTg mice would again provide strong evidence of a direct interaction between hormonal dysregulation and LOAD. As it stands the use of the 3xTg model seems incidental rather than being more informative than the same experiment in WT mice would have been.

Response 2— In light of the reviewer's recommendations, we have included a group of C57BL/6 mice in the drug treatment strategy trials (**Revised manuscript: Fig.7j**). In brief, this follows our confirmation that a successful maintenance of P4 levels in both plasma and brain can be achieved by utilizing a combination of intraperitoneal and subcutaneous P4 supplementation methods (**Revised manuscript: Fig 7k, Supplementary Fig.18**).

After completing the full 60-day treatment protocol, a subsequent evaluation of animal behavior was conducted using the Morris Water maze (specifically examining Latency to target) and Y maze (specifically Percentage alternation) paradigms (**Revised manuscript: Fig 7l, Supplementary Fig.19a-b**). In animals received vehicle-treatment, this assessment revealed that animals within the E2:P4<2.5 group, regardless of their genetic background (C57BL/6 or 3xTg), showed limited impacts from VCD (**Revised manuscript: Fig 7l, Supplementary Fig.19a-b**). Conversely, those in the E2:P4 \geq 2.5 group exhibited a notable impairment in these domains of behavioral performances, and such changes were particularly evident among the 3xTg mice (E2:P4 \geq 2.5) (**Revised manuscript: Fig 7l, Supplementary Fig.19a-b**).

In contrast, when P4 was administered, enhancements in functional performance were observed in the E2:P4 \geq 2.5 mice, and that more notable improvements were noted in the 3xTg subjects (**Revised manuscript: Fig 7l, Supplementary Fig.19a-b**). These changes could be in part attributed to reduced levels of pathological tau phosphorylation (AT-8) and the formation of monomeric, dimeric, and oligomeric amyloid- β (**Revised manuscript: Fig. 7m-n**,

Supplementary Fig.19c-d). It is worth noting that baseline levels of these pathological proteins were already elevated among the 3xTg animals with an E2:P4 status of ≥ 2.5 pg/ng before any rescuing P4 drug treatments were administered (**Revised manuscript: Fig. 7i, right panel; Supplementary Fig. 17c).**

Comment 3—Finally, the 3xTg line is considered a model of early-onset familial AD rather than for LOAD. None of the transgenic mutations it carries are associated with the idiopathic, late-onset form of the disease. The authors should make this caveat clear in the manuscript. An interesting addition would be to repeat the AOF paradigm in a mouse line harboring high genetic risk factors for LOAD specifically e.g. homozygous knock in APOE4 mice (available through JAX - <https://www.jax.org/strain/027894>), which would again provide more direct insights on the pathogenesis of LOAD.

Response 3— We acknowledge the reviewer's concern regarding the use of the early-onset familial Alzheimer's disease mouse model (3xTg), which is indeed a significant limitation of our study. The reviewer's suggestion to include a mouse line with the highest genetic risk factor for late-onset Alzheimer's disease (APOE4-KI) is valuable; however, the editor has recommended that we refrain from implementing this proposal. In response, we have added statements in the discussion section of the revised manuscript to recognize this limitation and to suggest the use of late-onset Alzheimer's disease-related mouse models in future research (Lines 729-732).

Confirmation of presynaptic effect of $ERR\alpha$ knock down.

Comment 4—The authors have performed patch clamp electrophysiology in primary cell cultures treated with $ERR\alpha$ targeted shRNA, demonstrating that this causes a specific increase in the frequency of miniature excitatory postsynaptic currents (mEPSCs). They conclude that this is due to an increase in presynaptic glutamate release driven by elevated levels of VGlut1, given that they observe no increases in the density of NR2B, which is used as a proxy marker for postsynaptic sites. While it is possible that this conclusion is correct, I don't believe the results presented fully exclude other possibilities. NR2B is not the best postsynaptic marker as not all excitatory synapses contain these receptors, and not all NMDARs are located at postsynaptic sites. A ubiquitous excitatory postsynaptic marker such as Homer1 or PSD95 would therefore be more appropriate for this analysis. Additionally, no attempt was made to colocalise presynaptic and postsynaptic signals in order to ensure that only mature functional synapses were included in the analysis. Repeating this analysis with these factors in place would give greater confidence to the conclusion that the increase in mEPSCs is driven by increased presynaptic release of glutamate.

Response 4— At the reviewer's request, we have substituted the figure with new Vglut1 and PSD95 staining, quantifying both the number of puncta for each and the number co-localized within 100 μ m neurites (**Revised manuscript: Fig.6g).** This quantification was conducted following a previously published protocol with slight adjustments, as detailed in the updated

methods and materials section⁷. Following these refined analyses, we observed heightened levels of VGlut1, while PSD95 levels remained unchanged when *Esrra* was knockdown or $ERR\alpha$ is inhibited. Additionally, in line with the reviewer's suggestion, we quantified the puncta where presynaptic VGlut1 and postsynaptic PSD95 co-localized (indicative of mature functional synapses). Interestingly, treatments that impacted $ERR\alpha$ function (specifically *Esrra* shRNA or XCT-790) led to an increase in the number of mature functional synapses. These findings further support the conclusion that the rise in mEPSC frequency is driven by an augmentation in presynaptic glutamate release.

Comment 5—Additionally, this conclusion can be confirmed through a relatively simple additional experiment consisting of paired-pulse ratio recordings. While not feasible in the primary cell cultures, the authors could utilize the hippocampal slice preps described elsewhere in the paper to compare the effect on *Esrra* and scrambled shRNA injected animals. A result displaying reduced paired-pulse facilitation in *Esrra* shRNA exposed neurons would definitively confirm the finding of enhanced presynaptic release probability.

Response 5— At the request of the reviewer, we conducted a series of electrophysiology experiments on hippocampal slices, including paired-pulse facilitation, as outlined below:

We initially delved into the impact of *Esrra* knockdown (**Revised manuscript: Fig.6a**) on glutamatergic transmission by conducting whole-cell voltage clamp recordings on CA1 pyramidal neurons. Miniature excitatory postsynaptic currents (mEPSCs) were recorded under the presence of bicuculline (a GABAA receptor antagonist) and tetrodotoxin (TTX, a blocker of voltage-gated sodium channels) to inhibit inhibitory neurotransmission and action potentials, respectively. Notably, mEPSC frequencies were significantly heightened in slices pre-treated with *Esrra* shRNA in comparison to the control group treated with scrambled shRNA (**Revised manuscript: Fig.6b**). Conversely, *Esrra* knockdown did not influence mEPSC amplitude when compared to the control group treated with scrambled shRNA (**Revised manuscript: Fig.6b**). Additionally, there were no discernible differences in passive membrane properties, such as resting membrane potential (RMP) and input resistance of pyramidal neurons between the two treatment groups as well (**Revised manuscript: Fig.6b**). These findings collectively suggest that *Esrra* knockdown augments glutamatergic transmission onto CA1 pyramidal neurons.

Furthermore, we explored the effects of *Esrra* knockdown on mIPSCs. Neurons were clamped at -70 mV, and miniature inhibitory postsynaptic currents (mIPSCs) were recorded in the presence of 20 μ M CNQX, 50 μ M DL-AP5 [antagonists of α -amino-3-hydroxy-5-methyl-4-isoxazolepropionic acid (AMPA) and N-methyl-D-aspartate (NMDA) receptors, respectively], and 1 μ M TTX, which hinders excitatory neurotransmission and action potentials. As depicted in the **revised Figure 6c**, *Esrra* knockdown did not alter mIPSC frequency. Moreover, there were no significant changes in mean amplitude observed. Collectively, these results indicate that *Esrra*

knockdown substantially enhances glutamatergic but not GABAergic transmission in CA1 pyramidal neurons.

The heightened frequency of mEPSCs suggests an increase in glutamatergic transmission owing to an elevated release probability. To substantiate this hypothesis, we assessed the paired-pulse response of evoked EPSCs (eEPSCs) at various intervals in CA1 pyramidal neurons (**Revised manuscript: Fig.6d**). Electrical stimuli were administered via a bipolar electrode positioned in the Schaffer collaterals (SC)-CA1 pathway. Neurons were maintained at -70 mV in the presence of 20 μ M bicuculline to impede GABAergic transmission. The paired-pulse ratio (PPR) was determined by the ratio of EPSC2 to EPSC1 amplitudes. As illustrated in **revised Figure 6d**, we observed a notable decrease in PPRs in the *Esrra* knockdown group in comparison to the scrambled shRNA control group. This outcome indicates that the suppression of *Esrra* expression led to an enhanced glutamate release probability from presynaptic terminals that project onto hippocampal CA1 pyramidal neurons.

Quantification of neurite integrity/loss

Comment 6—The authors claim that the AOF paradigm induces a reduction in neurite integrity in mice that developed an imbalanced E2:P4 ratio (Fig.2 J) and that injection of *Esrra* shRNA causes neurite loss (Fig.5b). However, the only evidence given for this are individual representative images. In order to make these claims the authors should quantify a measure of neurite integrity across cohorts of mice and perform appropriate statistical comparisons, rather than relying on a single image from individual animals.

Response 6— In compliance with the reviewer's inquiries, we have conducted the requested quantification analyses for Figure 2J (**Revised manuscript: Supplementary Fig.3h-j**) and Figure 5b (**Revised manuscript: Supplementary Fig.11c-d**) in the revised manuscript, encompassing all samples obtained from various cohorts. The revised figures now incorporate quantifications and statistical analyses of "percentage of samples within a cohort exhibiting neurite loss in distinct brain regions" and the "percentage of neurite length compared to corresponding controls". Additionally, details regarding the method for estimating neurite integrity have been included in the revised methods and materials section.

Reviewer #3 (Remarks to the Author):

General comment: This is an ambitious and interdisciplinary study that intends to connect the estradiol/progesterone imbalance occurring in perimenopause with the elevated vulnerability of women to Alzheimer's disease (AD) dementia. Here, the authors examined the transcriptional and behavioral effects and the underlying molecular mechanisms by which estradiol/progesterone imbalance (i.e, estrogen dominance), as that occurring in perimenopause, leads to memory loss in an accelerated ovarian failure (AOF) mouse model, and how the involved mechanisms may be linked to elevated susceptibility of dementia in women. Integrating the entire dataset into a comprehensive hypothesis systems model is not straightforward despite the variety of experimental models (mice, cultured neurons, human brain/transcriptomic data) and employed state-of-the art technical methodologies (sn-RNAseq data, metabolomics, AAV-mediated gene modulation).

The results obtained from the perimenopause mouse model fits with a model that estradiol/progesterone imbalance leads to memory and synaptic plasticity deficits by dysregulating essential metabolic pathways (i.e, oxidative phosphorylation, thermogenesis, TCA cycle). The study conclusively demonstrates that estradiol-mediated ER α signaling enhances genes encoding key metabolic enzymes (Pdha1, Dhcr24, Cyp51...) and related metabolites (pyruvate, acetyl-CoA, mevalonate...). The results suggest that disrupted ER α signaling may impact glycolysis and cholesterol biosynthesis, which potentially might impair cholesterol-bound ERR α -PGC1 α signaling in neurons of AOF mice and women with AD. However, direct evidence/s that altered ER α pathway disrupts ERR α -PGC1 α signaling (nuclear translocation, gene expression...), particularly in the context of AD (AD mice and brains) is lacking. In addition, the connection between estrogen dominance and the increased risk of dementia in postmenopausal women is not sufficiently addressed considering the challenges in translating results from the AOF mouse model to human transcriptomic and pathological findings (ROSMAP). Using a controlled non-dementia human cohort with both pre- and postmenopausal participants would have been useful to validate and reinforce some transcriptomic results/conclusions. In addition, the transcriptomic discrepancies on biological/metabolic pathways affected by genetic ERR α inactivation and perimenopausal mouse models do not seem to support the hypothesis.

Another critical issue is the diversity of brain regions analyzed (cortex vs hippocampus), some experiments with a very low number of samples (e.g, Fig. 1f, n=3) (human frontal cortex -Fig 1-, hippocampus or total cortex of AOF mice-Figs 2,3,4..), as well as the lack of cell-specific effects (bulk vs snRNA sequencing, metabolomics). It may be a good option to focus specifically in the estrogen dominance results and leave out AD. I also suggest several changes and considerations (see below) that could enhance the quality of the study. The abstract is quite confusing and should be more self-explanatory, for instance by describing, when necessary, the human samples and specific mouse models (age/sex). In conclusion, the study has several weaknesses that limit likely the global impact of the findings.

General response: We appreciate the reviewer's feedback and constructive criticism. We have now thoroughly addressed each of the individual comments provided.

Specific comments

Fig.1. ERR α -regulated bioenergetic network in AD

Main issue: No clear quantification of ERR α levels/localization in AD brains.

Comment 1—Fig 1b/c. Comparison of transcriptome changes (i.e, DEGs) between AD females vs AD males are missing.

Response 1— In response to the reviewer's request, we have integrated volcano plots to illustrate the comparison between non-dementia ND_{Female} and ND_{Male} (**Revised manuscript: Supplementary Fig. 1a**), and between disease-affected AD_{Female} and AD_{Male} (**Revised manuscript: Supplementary Fig. 1b**). Furthermore, a comprehensive list of differentially expressed genes (DEGs) has been appended to the corresponding spreadsheets within the Source Data File.

Subsequent to this update, we have identified a limited set of altered DEGs (Total: 558 genes, adj.p<0.05) in ND_{Female} and ND_{Male}, and the functional characterization of these DEGs did not reveal significant enrichment in any specific pathways (**Revised Manuscript: Source Data File**). This suggests that the brain transcriptomic profiles of non-demented female and male subjects exhibited general similarities. Notably, upon comparison made with male AD, genes that were downregulated in female AD brains were notably linked to functions associated with "oxidative phosphorylation" and "hedgehog signaling" (**Revised manuscript: Supplementary Fig.1d, Source Data File**). On the other hand, genes upregulated in female AD brains were involved in "xenobiotic metabolism," "IL-2/STAT5 signaling," "interferon gamma response," "p53 pathway," "estrogen response early and late," "complement," "hypoxia," and "epithelial mesenchymal transition" (**Revised manuscript: Supplementary Fig.1c, Source Data File**). These findings closely mirrored those observed in the comparison between AD_{Female} and ND_{Female} samples (**Revised manuscript: Fig.1c**), thereby validating the specificity of these observations to female AD samples.

Comment 2—Fig 1e/f. To confirm Mathys et al (2019) results (Fig. 1e), levels of ESRR α and PPARGC1A mRNA and protein should be quantified, at least in excitatory and inhibitory neurons, at cell-specific spatial resolution. Thus, ERR α and PGC1 α protein levels in excitatory and inhibitory neurons of AD and ND brains can be analyzed by double immunostaining with excitatory and inhibitory markers. Fig. 1f (lines 173-174): there is not a clear description how nuclear ERR α in MAP2+cells was quantified without using a nuclear marker. The authors should show the cytosolic, nuclear and/or nuclear/cytosolic ERR α levels in multiple ND vs AD male/female brain samples. Based in my own experience, n=3 seems is a very low n for human samples.

Response 2— At the reviewer's request, we conducted quantitative PCR to validate the transcript levels of *ESRRA* and *PPARGC1A* in human brain samples (**Revised manuscript: Supplementary Fig.2a, n = 4 subjects, 8 technical repeats**). Our analysis confirmed that there were no significant differences in the transcript levels of these genes between ND and AD samples of both sexes (**Revised manuscript: Supplementary Fig.2a**). This consistency was also observed in our analysis of specific cell types using single nuclear transcriptomic data, including astrocytes, excitatory neurons, and inhibitory neurons (**Revised manuscript: Supplementary Fig.2b**). However, our immunohistochemistry staining analyses revealed a notable finding: in female ND samples, the percentages of VGlut1+ excitatory neurons and GAD65+ inhibitory neurons showing positive nuclear signals for these two proteins were significantly lower compared to male ND samples (**Revised manuscript: Supplementary Fig.2c-d**). Further comparisons between sex-specific ND versus AD samples showed that such reductions were more evident in female samples (**Revised manuscript: Supplementary Fig.2c-d**), corroborating the broader observations depicted in the revised **Figure 1f**. The reductions in nuclear PGC1 α + and ERR α + populations were consistent across both excitatory and inhibitory neurons (**Revised manuscript: Supplementary Fig.2c-d**) (**n = 8 subjects, 4 experimental repeats**). All quantifications were conducted in the presence of a nuclear marker (i.e., DAPI).

Fig. 2. Mouse model of perimenopause/AOF

Comment 3—Fig 2f. Latencies during 6-day training in the MWM should be depicted to conclude that treatment causes spatial learning deficits.

Response 3— At the reviewer's request, we have reintroduced the latencies to target data during the 6-day training in the MWM paradigm conducted at time prior treatment, cycle 7 and cycle 14 back into the revised manuscript (**Revised manuscript: Supplementary Fig.3a**). Upon comparing the areas under the curve of the data, it became evident that a substantial number of subjects treated with VCD showed extended latencies to the target in the MWM paradigm, affirming that this treatment leads to spatial learning impairments.

Comment 4—Fig. 2j. Lines 212-218. The “neurite loss” caused by VCD treatment on neuronal dendritic morphology (MAP2 staining) should be quantified, and the specific hippocampal region/s analyzed should be indicated. Fig. 2k. State whether the “n” refers to number of slides, animals.

Response 4—As also requested by another reviewer (**Reviewer 2, Comment 6**), we performed the requested quantification analyses to Figure 2J (**Revised manuscript: Supplementary Fig.3h-j**) in the revised manuscript across all samples performed in different cohorts. Quantifications and statistical analyses related to “percentage of samples within a cohort with neurite loss in different

hippocampal regions (i.e., CA1, CA2 and CA3)” were performed as well. Detailed information regarding the “n” (i.e., animals) are added back to the figure legends.

Fig. 3. Bulk transcriptome changes in the cortex of the AOF mouse model

Main issue: low number of samples and the use of total brain cortex.

Comment 5—Fig. 3a-d. a.The low number of samples (n=3-4) for the bulk transcriptomic analyses of the AOF model is quite moderate, which could be a concern considering the apparent transcriptomic variability of VCD samples compared to controls (according to PCA plot of Fig 3a).c-d. Besides metabolic pathways (oxidative phosphorylation, TCA...), the majority of affected pathways are related to neurodegenerative diseases raising the possibility that VCD treatment (>2.5 pg/ng) cause neurodegeneration leading to dendritic and/or neuronal loss and synaptic plasticity impairments (see Fig. 2j-k).

Response 5— In light of the reviewer's feedback, we aim to address each sentence sequentially in our response:

Quote: “Fig. 3a-d. a.The low number of samples (n=3-4) for the bulk transcriptomic analyses of the AOF model is quite moderate, which could be a concern considering the apparent transcriptomic variability of VCD samples compared to controls (according to PCA plot of Fig 3a)”

Based on the findings from Figure 3a, the PCA plot, we understand that the "apparent transcriptomic variability" highlighted by the reviewer likely rooted from the observation that one sample of the VCD group appeared distinct from the rest of the group along the Y-axis (i.e., the bottom red dot). We contend that the variance along the Y-axis (15%) is notably smaller than that along the X-axis (59%). Therefore, given that the sample (the red dot at the bottom of the graph) aligns similarly along the X-axis, we do not agree with the comment that its position along the Y-axis signifies a significant deviation from the remainder of the group. Furthermore, supported by the volcano plot (**Revised manuscript: Figure 3b**), a total of 1,691 differentially expressed genes (DEGs) (adjusted $p < 0.05$) were identified. Achieving such a level of significance at this sample size would be challenging if there were substantial transcriptomic disparities among the samples (i.e., small transcriptomic profile differences between the control and treatment groups) under a well-controlled experimental condition in a laboratory setting.

Nevertheless, in response to this comment further, we repeated the experiment with a larger sample size (N = 8 per group) in a more robust manner, focusing on enriched neuronal populations extracted from these brains using magnetic-activated cell enrichment, as detailed in response to **Comment 9** from the same reviewer (**Revised manuscript: Supplementary Fig.12c-e**). This specific cell type enrichment approach was originally undertaken to fulfill the request for

conducting neuronal-specific metabolomics analysis (**Comment 9**). Following the enrichment and validation of the purity of enriched neurons through immunoblotting using antibodies against GFP (expressed from neuronal-specific shRNA constructs), NeuN (neuronal marker), and ERRA (validating neuronal *Esrra* knockdown) (**Revised manuscript: Supplementary Fig.12a**), bulk transcriptomic analysis was also carried out as well.

Upon comparing the treatment groups, a total of 941 upregulated and 761 downregulated differentially expressed genes (DEGs, adj $p < 0.05$, $\text{Log}_2\text{FC} > |0.25|$) were identified (**Revised manuscript: Supplementary Fig.12c-d, Source Data File**). While the upregulated DEGs did not show enrichment in meaningful pathways, the downregulated genes exhibited significant enrichment in pathways related to oxidative phosphorylation. This pathway was again commonly implicated in various neurodegenerative conditions such as Parkinson's disease, Prion disease, Huntington's disease, Alzheimer's disease and Amyotrophic Lateral Sclerosis (**Revised manuscript: Supplementary Fig.12e-f, Source Data File**). These findings align with those obtained from the original bulk tissue transcriptomic analysis, confirming that the observed changes at the tissue level primarily stemmed from alterations in neurons (**Revised manuscript: Fig.3a-e**).

Quote: “Besides metabolic pathways (oxidative phosphorylation, TCA...), the majority of affected pathways are related to neurodegenerative diseases raising the possibility that VCD treatment (>2.5 pg/ng) cause neurodegeneration leading to dendritic and/or neuronal loss and synaptic plasticity impairments (see Fig. 2j-k).”

Based on the information presented in **Figure 3c-d** and **Supplementary Fig.12c-f**, these panels depict unbiased pathway enrichment analyses with reference to the KEGG database renowned for its focus on metabolism, alongside other biologically significant pathways⁸. In an unbiased evaluation, it is common to observe a mixture of metabolic pathways and disease-related pathways if the differentially expressed genes enriched in these pathways exhibit high similarity, as is demonstrated in our data (**Revised manuscript: Source Data File**).

As pointed out by the reviewer, the pathways associated with neurodegenerative diseases could suggest that VCD treatment might directly lead to alterations in the expression of genes linked to dendritic loss or impairments in synaptic plasticity. While we acknowledge that these changes are commonly associated with neurodegeneration, our empirical data from the transcriptomic analysis indicate that majority of the differentially expressed genes enriched in these neurodegenerative pathways were, in fact, common to those enriched in the pathway of “oxidative phosphorylation” (**Revised manuscript: Source Data File and Fig.3d**). This underscores that the neurodegenerative modifications observed in Figure 2j-k (such as neurite loss and LTP impairment) could be a consequence of the loss of genes associated with oxidative phosphorylation function, specifically those involved in mitochondrial respiration and ATP production highlighted in our

experimental model, as later corroborated in **Figure 6**. Further analyses revealed that $ERR\alpha/ERR1$ serves as the primary common transcription factor governing the expression of these DEGs (**Revised manuscript: Fig.3e and Source Data File**) and that similar pathway enrichment findings were repeated in transcriptomic profiles of enriched neurons harvested in *Esrra* knockdown model (**Supplementary Fig.12c-f, Source Data File**).

Comment 6—Fig. 3f. As in Fig. 1f, it is not clear how and in which cortical region(s) nuclear $ERR\alpha$ and $PGC1\alpha$ levels are quantified. It is important to examine whether nuclear $ERR\alpha$ and $PGC1\alpha$ change in excitatory and inhibitory neurons by using colocalization studies.

Response 6— In our investigation, it is important to note that all analyses were specifically focused on the prefrontal cortex region of the samples, a detail that has now been included in the respective figure legends and methods section. Notably, we did not directly quantify the protein levels of $ERR\alpha$ and $PGC1\alpha$ in these analyses. Instead, our emphasis was on determining the proportion of nuclear $PGC1\alpha+$ or nuclear $ERR\alpha+$ neurons, identified through co-localization with neuronal markers like MAP2, VGlut1 (excitatory neurons), or GAD65 (inhibitory neurons), as illustrated in the corresponding data quantification panels (see below for more description).

Similar to our response to **Comment 2** from the same reviewer, we conducted additional immunohistochemistry staining analyses on the prefrontal cortex region of the corresponding mouse samples. In samples derived from VCD-treated animals exhibiting E2:P4 ratios ≥ 2.5 pg/ng, we noted significantly reduced percentages of VGlut1+ excitatory neurons and GAD65+ inhibitory neurons showing positive nuclear signals for $PGC1\alpha$ and $ERR\alpha$ proteins as compared to the vehicle control group and those treated with the same VCD regimen but having a E2:P4 ratios < 2.5 pg/ng status (**Revised manuscript: Supplementary Fig.4**). The declines in nuclear $PGC1\alpha+$ and $ERR\alpha+$ populations were consistently observed in both excitatory and inhibitory neurons, as depicted in **Supplementary Fig.4** of the revised manuscript. These observations are in line with the findings in human brain samples, as elaborated in **Supplementary Fig.2c**.

Comment 7—Fig. 3g. Co-immunoprecipitation experiments indicate that VCD treatment (>2.5 pg/ng) decreases $ERR\alpha/PGC1\alpha$ interaction. There is a clear decrease of $PGC1\alpha$ protein levels that may affect this interaction. Input $ERR\alpha$ levels should be also depicted in the graph. As explained in the text, $PGC1\alpha$ may be susceptible to degradation (line 244), so $PGC1\alpha$ transcript levels may be quantified to confirm or not this possibility.

Response 7—For the input $ERR\alpha$ level, it was already there in the original figure (**Revised manuscript: Fig.3g**). In response to the reviewer's next recommendation, we have integrated an evaluation of *Pparg1a* message levels in the same tissues and that no significant changes were found across all treatment groups (**Revised manuscript: Fig.3g, histogram located at the far**

right). This outcome provides additional support that the reduction in PGC1 α observed in the co-immunoprecipitation analyses was likely attributable to its vulnerability to degradation.

Comment 8—Fig 3i (lines 265-267) and Fig. 4k. I could not find how the ERR α -attached cholesterol is measured.

Response 8—In response to the reviewer's prompt, we have added back the revised methods for "**Extraction and measurement of ERR α -bound cholesterol**" (Lines 1299-1320) in the "**Materials and Methods**" section of the manuscript, referencing the previous publication⁹.

Fig. 4. ER α regulation of ERR α -PGC1 α signaling

Main issue: Lack of cell-specific metabolomics and no direct evidence that altered ER α pathway disrupts ERR α -PGC1 α signaling in neurons, and is altered in AD.

Comment 9—Based on the current data, the connection between ER α regulation of ERR α -PGC1 α signaling is not straight forward. Bulk transcriptomics (Figs. 3a,b) and metabolomics (Figs. 4a-d) showed downregulated genes clustered mainly to mitochondrial network (oxidative phosphorylation, TCA cycle...) and clear reduction of many metabolites of glycolysis, mevalonate and cholesterol biosynthesis pathways in total cerebral cortex of AOF mice. The main issue here is the lack of cell-specificity of metabolomic results: global metabolites/enzymes change it does not mean cell-specific (i.e, neurons) changes.

Response 9— In response to the reviewer's request, we have incorporated neuronal-specific metabolomics results into the revised manuscript. These additions are aimed at reinforcing the notion that the observed changes at the bulk tissue level in transcriptomic (**Revised manuscript: Fig.3a-b**) and metabolomic (**Revised manuscript: Fig. 4a-d**) analyses are specifically attributed to neurons rather than non-neuronal cell populations (**Revised manuscript: Supplementary Fig.6**). Additionally, we also added back this similar set of experiment to the enriched neuronal populations (i.e., GFP-positive, *Esrra* shRNA transduced neurons) to support the findings presented in **Figure 5a-f** at bulk tissue level was mainly attributed to changes in neurons (**Revised manuscript: Supplementary Fig.12, Source Data File**).

In brief, the isolation of neurons from adult mouse brain tissues was conducted following a modified protocol based on manufacturer instructions and a previous publication¹⁰. Brain tissues were harvested post-carbon dioxide termination, perfused with chilled PBS, and dissected into sagittal slices for further processing. Using the Adult Brain Dissociation Kit for mouse (Miltenyi Biotec), tissue dissociation was carried out in the gentleMACS C Tube with enzymatic digestion and mechanical dissociation. Debris and red blood cells were removed meticulously to obtain a purified cell suspension. Neurons were isolated from non-neuronal cells using a Biotin-Antibody Cocktail, followed by magnetic separation to collect the neuronal fraction. Immunoblot analysis

was performed for purity verification before the enriched neuronal fraction (NeuN+) or glia-enriched fraction (GFAP+, Iba+, Oligo2) was forwarded for metabolite profiling analyses (**Revised manuscript: Supplementary Fig.6a, 12a**), ensuring accurate neuronal-specific and/or glial-specific metabolomics results. The detailed protocol is now added to the methods and materials section of the revised manuscript.

Comment 10—Fig. 4e/h shows that primary cortical neurons treated with estradiol (E2) (effect blocked by a ER α antagonist) upregulates specific metabolites (pyruvate, acetyl-CoA, acetoacetyl-CoA, mevalonate) and their related enzymes (Pdha1, Dhcr24, Cyp51), a result reinforced by recruitment of ER α to these gene promoters (Fig. 4d). Since these results do not directly link ER α to ERR α dysfunction, it is essential to determine whether gain and loss of ER α function directly affects ERR α signaling.

Response 10—To address the reviewer's request on showing whether gain and loss of ER α function can directly affect ERR α signaling, one strategy is to measure how ER α signaling, in both progesterone-guided and non-guided scenarios, affect the nuclear activities of ERR α by a luciferase reporter assay. We indeed already included this analysis in **Figure 4m**. In the primary cortical neuron model, luciferase reporter for ERR α nuclear activities (i.e., pGL-3xERRE) was evaluated and normalized against the background signals of the vector (i.e., pGL-Basic). Robust luciferase signals were detected in neurons exposed to 100 nM estradiol cypionate (E2) alone for 120 hours (**Revised manuscript: Fig.4m, left panel**), and this induction was effectively abolished by co-treatment with 100 nM MMP dihydrochloride (MPP), a specific antagonist against ER α (**Revised manuscript: Fig.4m, left panel**). Furthermore, the E2-ER α -mediated ERR α -nuclear activities was also dependent on the guided-activity of progesterone receptor (PR), as the luciferase reporter signals were abolished by the co-treatment with 100 nM Mifepristone (MIF) a specific antagonist against PR (**Revised manuscript: Figure 4m, right panel**); but being enhanced by co-treatment with 100 nM P4 for 120 hours instead (**Revised manuscript: Figure 4m, right panel**).

Comment 11—Fig. 4g. Changes of metabolic-related ER α target genes are associated with cognitive impairment, an effect more evident in females of the ROSMAP cohort. Are there sex-differences (male vs females) in the ER α target genes within specific cognitive score groups?

Response 11— In response to the reviewer's request, the requested analysis was conducted (**Revised manuscript: Supplementary Fig.9**). Notably, no substantial difference was noted in the expression of these genes between male and female subjects with Cogdx scores of 1 and 2. However, these genes, with the exception of *SDHD*, exhibited a notable reduction in expression levels among female subjects compared to males as cognitive function declined (specifically, for Cogdx scores of 3 or 4).

Fig. 5. ERR α regulates the TCA-NAAG pathway

Main issue: Transcriptomic discrepancies between genetic $ERR\alpha$ inactivation and perimenopausal mouse models

Comment 12—The authors describe that $ERR\alpha$ inactivation in mouse hippocampus and frontal cortex (are male and/or female mice ?) affects negatively memory (lines 350-351; Supl Fig. 7) and TCA metabolites (Fig. 5g) similar to VCD mice (Supl Fig. 8). First, there is no clear evidence that this AAV transduces neurons of the mouse brain. Second, according to data presented in Supl. Fig. 7c (lines 350-351), there is not a clear effect of $ERR\alpha$ inactivation in learning/memory in the Y-maze (% alternation is similar for most mice) and MWM (latencies during days 1-6 and % target quadrant occupancy during the probe trial should be shown). Third, based on the $ERR\alpha$ inactivation experiments, it is clear that $ERR\alpha$ upregulates TCA pathway genes/metabolites (Fig. 5g) in contrast to downregulation of glycolysis/TCA/mevalonate/cholesterol pathways in VCD-treated mice (Fig. 4).

Response 12— Here we address the reviewer's comments systematically, one statement after another:

Quote: “First, there is no clear evidence that this AAV transduces neurons of the mouse brain.”

To investigate the relationship between brain function and neuronal $ERR\alpha$ levels, we utilized a targeted approach by stereotaxically injecting an adeno-associated virus (AAV) containing a microRNA 30 (miR30)-based short hairpin (shRNA) designed to silence *Esrria* specifically in neurons within the hippocampal CA1 region, as depicted in **Figure 5a-b and 6a**. This AAV construct enables the expression of shRNA under the control of the human synapsin (hSyn) promoter, facilitating cell- and region-specific gene silencing¹¹. This methodology, previously employed in our prior publication, has been successful in achieving precise gene knockdown in neurons¹². The specificity of neuronal expression of this shRNA and its efficacy in suppressing *Esrria* within these cells were further confirmed in additional experiments requested by the same reviewer, the one focusing on cell-specific metabolomics analysis in enriched neurons from these transduced brains (**Revised manuscript: Supplementary Fig.12a-b**). In that immunoblotting analysis after sub-fractionation of neuronal cells from non-neuronal ones, the fraction being positive for NeuN was also the same one enriched with GFP signals derived from the AAV construct, and the latter was absent from the non-neuronal fraction.

Quote: “Second, according to data presented in Supl. Fig. 7c (lines 350-351), there is not a clear effect of $ERR\alpha$ inactivation in learning/memory in the Y-maze (% alternation is similar for most mice) and MWM (latencies during days 1-6 and % target quadrant occupancy during the probe trial should be shown).”

To provide clarity on the statistical power and sample size considerations, we have reintegrated the following power analysis statement into the behavior analysis section: “For animal studies, a sample size of 12-15 per group will achieve an 80% power to detect a 24% difference between the means at a significance level of 0.05 using an unpaired t-test. Given that differences lower than 25% are not considered biologically relevant, the recommended minimum sample size for experiments will be 12-15 animals per group. This range ensures sufficient statistical power to identify significant differences in the experimental outcomes.” Additionally, alongside the inclusion of latency data from the training phase in the revised figure, we have also incorporated the percentage difference in mean values for each highlighted panel to better illustrate their distinctions (**revised manuscript: Supplementary Fig.11f-g**). All results indicate that the differences in mean values exceed 25% of the control group (scrambled shRNA), with p-values < 0.05, underscoring their statistical significance.

Quote: “Third, based on the $ERR\alpha$ inactivation experiments, it is clear that $ERR\alpha$ upregulates TCA pathway genes/metabolites (Fig. 5g) in contrast to downregulation of glycolysis/TCA/mevalonate/cholesterol pathways in VCD-treated mice (Fig. 4).”

It appears that there may have been some misunderstanding. Referring to the original **Figure 5g**, all highlighted metabolites were denoted in red, indicating their downregulation in $ERR\alpha$ shRNA group, while those labeled in black did not exhibit significant differences. The numbers in brackets represented the $\text{Log}_2(\text{Fold change})$ values and $-\text{Log}_{10}(\text{FDR})$ values, respectively, when comparing the *Esrra* shRNA versus Scrambled shRNA groups.

In the context of VCD-induced perimenopause, which initiates a state akin to the early phase of menopause characterized by the loss of progesterone-guided estrogen receptor signaling, substantial gene expression suppression linked to mitochondrial metabolism (including oxidative phosphorylation and the TCA cycle) were observed (**Revised manuscript: Fig.3c**). Our promoter transcription factor analysis indicated that these changes could be attributed to the diminished nuclear activities of $ERR\alpha$ (**Revised manuscript: Fig.3e**). These data suggested that $ERR\alpha$ may well act as one of the downstream effectors of progesterone-guided estrogen receptor signaling, rather than being solely responsible for all observed changes when hormonal signaling is disrupted in the model system. Further investigations into the mechanisms behind the loss of $ERR\alpha$'s nuclear activities, as demonstrated through immunostaining (**Revised manuscript: Fig.3f**), immunoblotting (**Revised manuscript: Fig.3g**), *in silico* modelling and ligand-binding analysis (**Revised manuscript: Fig. 3h-i**), hinted at a potential link to reduced cholesterol availability. The subsequent unbiased metabolomic analysis depicted in **Figure 4a-b** not only validated this idea but also unveiled a reduction in glycolytic metabolites that contribute carbon for cholesterol biosynthesis, as highlighted in **Figure 4j**.

Comment 13—When comparing Figs. 3c and 5f, the general picture that emerges highly differs: the most relevant metabolic pathways affected in the perimenopause mouse model are oxidative phosphorylation, TCA and neurodegenerative diseases, whereas $ERR\alpha$ inactivation affects TCA, glycolysis, pyruvate and amino acid metabolism. According to the authors hypothesis presented in previous figures, we should expect disruption of glycolysis/TCA/mevalonate/cholesterol pathways by neuronal $ERR\alpha$ inactivation. How do the authors explain that neuronal $ERR\alpha$ inactivation does not affect mevalonate/cholesterol biosynthesis?

Response 13— We would like to provide further clarification on a few points. Firstly, the gene set enrichment analysis in **Figure 3c** was derived from differentially expressed genes identified in the transcriptomic analysis following VCD-treatment, while the analysis in **Figure 5f** was based on metabolite set enrichment from the *Esrra*-knockdown integrated transcriptomic-metabolomic data. It is reasonable not to expect a complete overlap in the enriched pathways between these different analyses, as there is no known set of metabolites directly associated with any specific diseases. Additionally, as previously detailed in **Response 5, Quote 2** to the same reviewer, the genes enriched in various neurodegenerative pathways, and oxidative phosphorylation, were highly common to one another.

Regarding the explanation for why "neuronal $ERR\alpha$ inactivation does not affect mevalonate/cholesterol biosynthesis," as previously discussed in **Response 12**, we posited that $ERR\alpha$ acts as one of the downstream effectors of progesterone-guided estrogen receptor signaling. Therefore, it is not solely responsible for all the observed changes when hormonal signaling is disrupted in the model system. Furthermore, our data revealed that key genes involved in *de novo* biosynthesis and transport of cholesterol were targets of $ER\alpha$, not $ERR\alpha$ (**Revised manuscript: Fig.4h-i, Supplementary Fig.7**). On the other hand, our study indeed suggested neuronal $ERR\alpha$ inactivation primarily impacts mitochondrial TCA cycle and oxidative phosphorylation functionality observed during the peri-menopausal period when progesterone-guided estrogen receptor signaling is impaired (**Revised manuscript: Fig.5a-i**). In simplicity, we believe the sequence of regulatory events from upstream to downstream is: PR-guided $ER\alpha$ signaling \rightarrow mevalonate/cholesterol homeostasis \rightarrow $ERR\alpha$ activity \rightarrow mitochondrial function.

Comment 14—If $ERR\alpha$ is disrupted in the brain of AD females (Fig. 1), how is possible that metabolomics KEGG enrichment analyses does not show affectation of mevalonate/cholesterol-related metabolites in AD females (ROSMAP cohort)? On the contrary, the main affectation is on amino acid metabolisms/urea/ammonia cycle/lipids (Figs. 5j-1).

Response 14— Building upon the explanations provided in Response to **Comment 12**, we believe the conclusion that $ERR\alpha$ functions as one of the downstream effectors of progesterone-guided estrogen receptor signaling, implying it is not solely accountable for all the alterations observed upon disruption of hormonal signaling in our model system. As highlighted in **Figure 4e-f and h-j**, critical genes governing the mevalonate/cholesterol pathway are under the regulation of

progesterone-guided estrogen receptor (ER α) signaling, not that of ERR α . The subsequent impact of diminished cellular cholesterol *de novo* biosynthesis or reduced cholesterol import results in the downstream impairment of ligand-activated co-activator binding (such as PGC1 α interaction) with ERR α , consequently affecting its transcriptional activities within the nucleus, as illustrated in **Figure 4k-m**.

The absence of mevalonate/cholesterol-related metabolites in females with AD compared to non-demented females, as observed in contrast to the VCD-perimenopause model, could stem from the full-term menopausal status and the corresponding variations in progesterone-guided estrogen receptor signaling within the samples analyzed. Female subjects within the ROSMAP cohort, irrespective of their disease status, were typically over 85 years old at the stage of full-term menopause, where estrogen and progesterone levels naturally decline to minimal levels post-ovarian function cessation¹³. This contrasts with the observations in our laboratory test model at the early phase of menopause, where progesterone-guided estrogen receptor signaling remained active in the vehicle control group (in contrast to the VCD group with E2:P4 ratios ≥ 2.5 pg/ng), leading to distinct metabolite changes.

Notably, we would like to add here that intriguing findings related to the *APOE* genetic status emerged during the re-analysis of ERR α and PGC1 α immunostaining, despite the limited number of samples available in our laboratory (**Revised manuscript: Fig.1f, Supplementary Fig.2c-d**). Brain samples from individuals carrying the E4 variant allele generally exhibited reduced nuclear ERR α and PGC1 α signals, with a more pronounced decrease in samples harboring two E4 variant alleles. Given that ApoE plays a crucial role in transporting cholesterol from astrocytes to neurons, and its E4 variant is associated with impaired function, this further supports the hypothesis that diminished neuronal cholesterol availability is linked to decreased nuclear localization of ERR α and activities. The compounded effects of *APOE* E4 status alongside the loss of progesterone-guided estrogen receptor signaling were reflected in the lower number of brain nuclear ERR α + neurons in female subjects with and without dementia, when compared to their male counterparts, as evidenced by the lower number of MAP2+/Nuclear ERR α + neurons in these brain samples (**Revised manuscript: Fig.1f, Supplementary Fig.2c-d**).

The alterations observed in amino acid metabolisms, urea cycle, ammonia metabolism, and lipids could be a consequence of prolonged full-term menopause leading to the advancement of Alzheimer's disease. Our previous research has also suggested a potential upregulation in glutamine and branched-chain amino acid catabolism in the advanced stages of Alzheimer's disease, accompanied by an increase in urea cycle metabolism to detoxify ammonia generated in this process¹⁴. Regarding the lipid species detected in the ROSMAP metabolomics data, many were indeed cholesterol-related metabolites such as 4-cholesten-3-one¹⁵, 7-hydroxycholesterol (alpha or beta); and phospholipids indicated in cholesterol and lipid transport (e.g., glycerophosphoinositol¹⁶, 13-HODE + 9-HODE¹⁷, CDP-ethanolamine¹⁸,

Glycerophosphoethanolamine¹⁹, myo-inositol²⁰); whose levels could be influenced by neurodegeneration and cell death encountered in the advanced stages of the disease. These pieces of information are now updated to **Figure 5g** in the revised manuscript.

Fig. 6. Inactivation of ERR α enhances spontaneous postsynaptic activity and negatively affects respiratory capacity and neuron viability

Comment 15—It is challenging to reconcile how TCA/mitochondrial deficits and ATP depletion lead to increased spontaneous glutamate release, and this ultimately may contribute to AD vulnerability in women.

Response 15— It appears there was a misunderstanding, and the manuscript actually conveys the reverse relationship: the increase in free glutamate in these neurons, stemming from enhanced NAAG-catabolism as a surrogate for completing the TCA cycle (**Revised manuscript: Fig.5a-i**), leads to heightened spontaneous glutamate release, manifested as an escalation in the frequency of excitatory miniature postsynaptic potentials (mEPSCs), so as a notable decrease in paired-pulse facilitation, hence an over firing of neurons (**Revised manuscript: Fig.6a-g**). Both synaptic release and recycling²¹⁻²³, along with action potentials^{24,25}, are energy-intensive processes. When enhancement of these processes (i.e., increase in energy demand) is combined with the impairment of the TCA cycle and oxidative phosphorylation (i.e., reduction in energy supply) due to the loss of ERR α activity in perimenopausal subjects complicated by “higher estradiol, lower progesterone imbalance” (i.e., E2:P4 \geq 2.5 pg/ng), the imbalance between reduced energy supply but simultaneous heightened energy demand would result in ATP depletion and an energy crisis²⁶. This ultimately increases vulnerability to neurodegeneration and Alzheimer's disease in women.

Comment 16—Fig 6a,b. Instead of primary neurons, it seems more logical and relevant for the entire study to measure excitatory/inhibitory neurotransmission and synaptic plasticity (i.e, LTP) alterations in brain slices of genetic ERR α -inactivated mice (Fig. 5). In summary, are the electrophysiology results in primary neurons replicated in vivo ?

Response 16— In line with the primary aim of this figure, which is to verify the proposition that *Esrra* knockdown notably enhances glutamatergic, but not GABAergic, transmission by amplifying glutamate release from presynaptic terminals in CA1 hippocampal neurons, we have incorporated matching mEPSP and mIPSC analyses, along with pair-pulse facilitation in hippocampal slice culture, as also recommended by **Reviewer 2 (Comment 5)**, with the following findings:

We initially delved into the impact of *Esrra* knockdown (**Revised manuscript: Fig.6a**) on glutamatergic transmission by conducting whole-cell voltage clamp recordings on CA1 pyramidal neurons. Miniature excitatory postsynaptic currents (mEPSCs) were recorded under the presence of bicuculline (a GABA_A receptor antagonist) and tetrodotoxin (TTX, a blocker of voltage-gated

sodium channels) to inhibit inhibitory neurotransmission and action potentials, respectively. Notably, mEPSC frequencies were significantly heightened in slices pre-treated with *Esrra* shRNA in comparison to the control group treated with scrambled shRNA (**Revised manuscript: Fig.6b**). Conversely, *Esrra* knockdown did not influence mEPSC amplitude when compared to the control group treated with scrambled shRNA (**Revised manuscript: Fig.6b**). Additionally, there were no discernible differences in passive membrane properties, such as resting membrane potential (RMP) and input resistance of pyramidal neurons between the two treatment groups as well (**Revised manuscript: Fig.6b**). These findings collectively suggest that *Esrra* knockdown augments glutamatergic transmission onto CA1 pyramidal neurons.

Furthermore, we explored the effects of *Esrra* knockdown on mIPSCs. Neurons were clamped at -70 mV, and miniature inhibitory postsynaptic currents (mIPSCs) were recorded in the presence of 20 μ M CNQX, 50 μ M DL-AP5 (antagonists of α -amino-3-hydroxy-5-methyl-4-isoxazolepropionic acid (AMPA) and N-methyl-D-aspartate (NMDA) receptors, respectively), and 1 μ M TTX, which hinders excitatory neurotransmission and action potentials. As depicted in the **revised Figure 6c**, *Esrra* knockdown did not alter mIPSC frequency. Moreover, there were no significant changes in mean amplitude observed. Collectively, these results indicate that *Esrra* knockdown substantially enhances glutamatergic but not GABAergic transmission in CA1 pyramidal neurons.

The heightened frequency of mEPSCs suggests an increase in glutamatergic transmission owing to an elevated release probability. To substantiate this hypothesis, we assessed the paired-pulse response of evoked EPSCs (eEPSCs) at various intervals in CA1 pyramidal neurons (**Revised manuscript: Fig.6d**). Electrical stimuli were administered via a bipolar electrode positioned in the Schaffer collaterals (SC)-CA1 pathway. Neurons were maintained at -70 mV in the presence of 20 μ M bicuculline to impede GABAergic transmission. The paired-pulse ratio (PPR) was determined by the ratio of EPSC2 to EPSC1 amplitudes. As illustrated in **revised Figure 6d**, we observed a notable decrease in PPRs in the *Esrra* knockdown group in comparison to the scrambled shRNA control group. This outcome indicates that the suppression of *Esrra* expression led to an enhanced glutamate release probability from presynaptic terminals that project onto hippocampal CA1 pyramidal neurons.

Fig. 7. Effect of VCD in 3xTgAD mice.

Main issue: The analyses of E2:P4 ratio, behavior and cortical transcriptomic/metabolomics (cholesterol homeostasis/metabolites) indicate that vehicle-treated C57BL/6 and 3xTgAD mice behave similarly, whereas results in C57BL/6- and 3xTg AD-VCD treated mice (Figs. 2e-g, 4b-d, 7b-d) are almost identical (Behavior: 30-40 sec latency to platform target in the MWM and 20-40 % alteration in the T-maze).

Comment 16—Considering that results of VCD-treated C57BL/6 vs VCD-treated 3xTg-AD mice are missing is not possible to know whether VCD-treatment causes differential effects in control and AD pathological conditions. For that, these two groups should be compared.

Response 16— As also response to the other reviewer's suggestions (**Reviewer 2, Comments 1 and 2**), we also investigated the following during the revision:

1. Effect on VCD treatment on AD pathologies (pTau and A β)

First, we investigated if the AOF paradigm accelerates the appearance of AD-related pathologies in 3xTg mice. In the revised manuscript, we elaborated our investigation further with immunoblotting experiments with specific antibodies targeting human APP/A β (6E10) and pTau (AT-8) (**Revised manuscript: Fig.7i, Supplementary Fig.17c**), as the other reviewer (i.e., Reviewer 2) also suggested. In the updated result, robust elevated levels and immunoreactivity of phosphorylated tau at S202/T205 recognized by AT-8 antibody was found specifically in VCD-treated 3xTg mice with E2:P4 ≥ 2.5 pg/ng, whereas that in VCD-treated mice with the status of E2:P4 < 2.5 pg/ng remained indifferent from the vehicle-treated 3xTg control and C57BL/6 at this age, suggesting the status of E2:P4 ≥ 2.5 pg/ng is a risk that associates with pathological tau hyperphosphorylation. Progesterone drug treatment analysis further validated the importance of E2:P4 status to pathological tau phosphorylation in neurons (**Revised manuscript: Fig.7m-n, Supplementary Fig. 19a-c, 20**). In the 3xTg animals exposed only to VCD and drug vehicle treatment, signals of intraneuronal AT-8 were more robustly observed in ones with E2:P4 ≥ 2.5 pg/ng (**Revised manuscript: Figure 7m-n, Supplementary Fig. 19c-d**), and such observation was reversed in subjects that were instead treated with progesterone (**Revised manuscript: Fig.7m-n, Supplementary Fig.19c-d**). A similar pattern of changes to the level of A β monomer, dimer and oligomer detected by the 6E10 antibody was found as well (**Revised manuscript: Fig.7i, Supplementary Fig.17c; Fig.7m-n, Supplementary Fig.19c-d**). We reasoned that the positive correlation between the E2:P4 ≥ 2.5 pg/ng status with AD pathologies is potentially related to a sustained elevation of spontaneous excitatory firing among the neurons in these brains, as this phenomenon was also supported by the previous literature²⁻⁶.

2. Effect of P4 supplementation to VCD-treatment in C57BL/6 versus 3xTg mice

In this revised manuscript, we now added a group of C57BL/6 mice in the drug treatment strategy trials. As also requested by Reviewer 2 (**Revised manuscript: Figure 7j**). In brief, this follows our confirmation that a successful maintenance of P4 levels in both plasma and brain can be achieved by utilizing a combination of intraperitoneal and subcutaneous P4 supplementation methods (**Revised manuscript: Fig 7k, Supplementary Fig. 18**).

After completing the full 60-day treatment protocol, a subsequent evaluation of animal behavior was conducted using the Morris Water maze (specifically examining Latency to target) and Y maze (specifically Percentage alternation) paradigms (**Revised manuscript: Fig 7l, Supplementary**

Fig.19a-b). In animals received vehicle-treatment, this assessment revealed that animals within the E2:P4<2.5 pg/ng group, regardless of their genetic background (C57BL/6 or 3xTg), showed minimal behavioral changes from VCD (**Revised manuscript: Fig 7l, Supplementary Fig.19a-b**). Conversely, those in the E2:P4≥2.5 pg/ng group exhibited a notable decrease in their behavioral performances, particularly this effect was more evident among the 3xTg mice (**Revised manuscript: Fig 7l, Supplementary Fig.19a-b**).

In contrast, when P4 supplementation was administered, enhancements in functional performance were observed in the E2:P4≥2.5 mice, with more significant improvements noted in the 3xTg subjects (**Revised manuscript: Fig 7l, Supplementary Fig.19a-b**). These enhancements are likely attributed to reduced levels of pathological tau phosphorylation (AT-8) and the formation of monomeric, dimeric, and oligomeric amyloid-β (**Revised manuscript: Fig. 7m-n, Supplementary Fig.19c**). It is worth noting that baseline levels of these pathological proteins were already elevated among the 3xTg animals with an E2:P4 status of ≥ 2.5 pg/ng before any P4 treatments were administered (**Revised manuscript: Fig. 7i, right panel; Supplementary Fig. 17c**), highlighting that the protective effect of P4 was much pronounced than those observed in *C57BL/6* animals.

Comment 17—Does E2:P4 imbalance (> 2.5 pg/ng) affect amyloid and tau pathologies in 3xTg AD mice at this age?

Response 17—As also elaborated in our response to **Comment 16**, in the 3xTg animals exposed only to VCD and vehicle control treatment, signals of intraneuronal AT-8 were more robustly observed in ones with E2:P4 ≥ 2.5 pg/ng (**Revised manuscript: Figure 7m-n, Supplementary Fig. 19c**), and such observation was reversed in similar subjects that were instead treated with progesterone (**Revised manuscript: Fig.7m-n, Supplementary Fig.19c**). A similar pattern of changes to the level of Aβ monomer, dimer and oligomer detected by the 6E10 antibody was found as well (**Revised manuscript: Fig.7i, Supplementary Fig.17c; Fig.7m-n, Supplementary Fig.19c**). Moreover, progesterone treatment to 3xTg mice pre-exposed to VCD whose E2:P4 ratio were ≥ 2.5 pg/ng resulted in obvious improvements of both amyloid and tau pathologies (**Revised manuscript: Fig.7m-n, Supplementary Fig. 19c**).

Reviewer #4 (Remarks to the Author):

This manuscript by Ka-Li et al. describes a set of interesting experiments in human samples and mouse models to elucidate the pathways linking estrogen dominance in women in midlife with increased risk of cognitive decline.

Related to metabolomics:

The metabolomic profiling was performed using a well-validated commercial platform through Human Metabolome Technologies.

Comment 1—Generally, it would be helpful to provide more information related to metabolomic data analyses to get a clearer picture on how to interpret the findings from the metabolomics data. For example, in the analyses presented in Figs. 4A-B and 5E: How many metabolites were examined in total? Are the metabolites presented in the heatmap only the subsets that were significant at $p < 0.05$? Did you consider an adjustment for multiple testing (e.g., FDR, Bonferroni, etc.)? Many of these associations may be due to chance (related to the high number of statistical tests performed). Please justify in the methods if multiple testing correction was not considered.

Response 1— In response to the reviewer's request, we have made updates to **Figures 4a-b and 5e (Revised manuscript: Source Data File)**. Specifically, for the metabolomic dataset related to **Figure 4a-b**, a total of 366 metabolites were detected. We have refined the statistical analyses by conducting multiple unpaired t-tests with Welch correction on each metabolite without assuming consistent standard deviations. The two-stage step-up method (Benjamini, Krieger, and Yekutieli) was employed to manage the false discovery rate (FDR) during the multiple hypothesis testing where p-values may not be entirely independent. The updated analysis revealed 58 significantly dysregulated metabolites (all downregulated) with an $FDR < 0.05$ (**Revised manuscript: Fig.4a, Source Data File**). Among these, several metabolites linked to pathways such as "Steroid/cholesterol biosynthesis" (Green), "Warburg effect and glycolysis/gluconeogenesis" (Magenta), and "Mitochondrial electron transport chain and citrate cycle" (Blue) were enriched ($FDR < 0.05$) (**Revised manuscript: Fig.4b, Source Data File**). These re-analyses findings still align well with the results from the transcriptomic analyses (**Revised manuscript: Fig.4c-d**).

For the metabolomic dataset presented in **Figure 5e**, we conducted similar re-analyses. After adjusting for multiple testing, 74 out of 390 metabolites showed differential detection ($FDR < 0.05$) between the *Esrra* shRNA and scrambled shRNA-treated groups (**Revised manuscript: Fig.5e**). Many of these metabolites were enriched in "Ala, Asp and Glu metabolism" (Red), "Glycolysis/Gluconeogenesis" (Magenta) "Citrate cycle" (Blue) pathways ($FDR < 0.05$) (**Revised manuscript: Fig.5e-f, Source Data File**). These changes were further supported by subsequent metabolite fate tracing analyses presented in **Figure 5g-h**.

Comment 2—In the analyses to determine sex-specific effects (shown in panels in Fig. 5): What statistical test was used to determine effect modification by sex? Supplementary Table 11 shows significant metabolites stratified by sex, but it is difficult to know how to compare these results without an actual test of effect modification/interaction by sex (e.g., do males have fewer significant metabolites than females because of a true biological effect or because the sample size is lower in the male group?).

Response 2—In response to the reviewer’s comment, we have added in more details back to the manuscript, as laid out in **Supplementary Fig. 14**. We indeed began our analysis via performing a simple comparison between all LOAD versus ND samples regardless of the biological sex of the patients where the samples were harvested from (**Revised manuscript: Supplementary Fig.14a**). Analysis of differentially expressed metabolites (DEMs) was performed for a given metabolite by fitting a generalized linear model to estimate the association between the expression of a metabolite with its disease status, while adjusting for covariates, including their disease status, education levels, post-mortem intervals, age and biological sex using Gaussian linear regression, as advised by the original publication²⁷. Multiple testing correction was then performed on the calculated p-value using the Benjamini-Hochberg (BH) procedure, controlling the False Discovery Rate (FDR) to give the adjusted p-value (**Revised manuscript: Supplementary Fig.14b**). Notably, even in this simple comparison setting, N-acetylaspartylglutamate (NAAG) still ended up as the most significantly reduced metabolite in LOAD.

We then asked if this change was simply disease-specific or did it also dependent on biological sex. Thanks to the reviewer’s comment and reminder, we now realized that there is a significantly higher number of female samples in comparison to the male samples in both the LOAD (Male, 62; Female, 161) and ND groups (Male, 86; Female: 191) (**Revised manuscript: Supplementary Fig.14a**). To ensure the findings we reported was not due to such sample number difference but was a true biological effect, permutation analysis was performed by randomly downsizing the female sample numbers for 50 times to match the number of male-specific samples, and then the same statistical inferences mentioned above were applied to the randomized, downsized dataset²⁸. As presented in the **revised Supplementary Fig.14b**, histograms revealed the number of permutations round across different total numbers of upregulated and downregulated DEMs identified, with the red dotted line indicating the original number of DEMs before the downsizing of samples was conducted. While generally speaking, the original analysis did reveal a higher number of DEMs in comparison to the numbers of that identified from most of the permutation analyses (**Revised manuscript: Supplementary Fig.14c**), the empirical cumulative distribution function (ECDF) plot—which visually represents the distribution of a dataset via plotting the percentage of the significance cumulative against the $-\log_{10}(\text{adjusted p-value})$ of each metabolite—revealed that even though differences in metabolite hits did emerge from each round of permutation, these randomized datasets exhibited similar trends of distributions and cumulative probabilities at each value²⁹ (**Revised manuscript: Supplementary Fig.14c**). Metabolite wise,

the cumulative frequencies of the DEMs that found to be significant across all 50 permutations were plotted (**Revised manuscript: Supplementary Fig.14d-e, Source Data File**). Among all possible 297 significantly enriched (adjusted p-values < 0.05) DEMs identified in the 50 rounds of permutation analyses, key metabolites of our interest the N-acetyl-aspartyl-glutamate (NAAG), was found to be significantly downregulated across 86% of the permutations; whereas glutamate was significantly upregulated across 38% of all the permutation analyses as well (**Revised manuscript: Supplementary Fig.14d-e, Source Data File**). Together, these finding validated that our discovery was not due to biases from preexisting differences in sample numbers.

Reference list

- 1 Strang, K. H. *et al.* Generation and characterization of new monoclonal antibodies targeting the PHF1 and AT8 epitopes on human tau. *Acta Neuropathol Commun* **5**, 58 (2017). <https://doi.org/10.1186/s40478-017-0458-0>
- 2 Liang, Z., Liu, F., Iqbal, K., Grundke-Iqbal, I. & Gong, C. X. Dysregulation of tau phosphorylation in mouse brain during excitotoxic damage. *J Alzheimers Dis* **17**, 531-539 (2009). <https://doi.org/10.3233/JAD-2009-1069>
- 3 Rahman, A. *et al.* The excitotoxin quinolinic acid induces tau phosphorylation in human neurons. *PLoS One* **4**, e6344 (2009). <https://doi.org/10.1371/journal.pone.0006344>
- 4 Ismael, S., Sindi, G., Colvin, R. A. & Lee, D. Activity-dependent release of phosphorylated human tau from Drosophila neurons in primary culture. *J Biol Chem* **297**, 101108 (2021). <https://doi.org/10.1016/j.jbc.2021.101108>
- 5 Bero, A. W. *et al.* Neuronal activity regulates the regional vulnerability to amyloid-beta deposition. *Nat Neurosci* **14**, 750-756 (2011). <https://doi.org/10.1038/nn.2801>
- 6 Stargardt, A., Swaab, D. F. & Bossers, K. Storm before the quiet: neuronal hyperactivity and Abeta in the presymptomatic stages of Alzheimer's disease. *Neurobiol Aging* **36**, 1-11 (2015). <https://doi.org/10.1016/j.neurobiolaging.2014.08.014>
- 7 Plowey, E. D. *et al.* Mutant LRRK2 enhances glutamatergic synapse activity and evokes excitotoxic dendrite degeneration. *Biochim Biophys Acta* **1842**, 1596-1603 (2014). <https://doi.org/10.1016/j.bbadis.2014.05.016>
- 8 Kanehisa, M. & Goto, S. KEGG: kyoto encyclopedia of genes and genomes. *Nucleic Acids Res* **28**, 27-30 (2000). <https://doi.org/10.1093/nar/28.1.27>
- 9 Ghanbari, F., Mader, S. & Philip, A. Cholesterol as an Endogenous Ligand of ERRalpha Promotes ERRalpha-Mediated Cellular Proliferation and Metabolic Target Gene Expression in Breast Cancer Cells. *Cells* **9** (2020). <https://doi.org/10.3390/cells9081765>
- 10 Schroeter, C. B. *et al.* One Brain-All Cells: A Comprehensive Protocol to Isolate All Principal CNS-Resident Cell Types from Brain and Spinal Cord of Adult Healthy and EAE Mice. *Cells* **10** (2021). <https://doi.org/10.3390/cells10030651>
- 11 Liu, B., Xu, H., Paton, J. F. & Kasparov, S. Cell- and region-specific miR30-based gene knock-down with temporal control in the rat brain. *BMC Mol Biol* **11**, 93 (2010). <https://doi.org/10.1186/1471-2199-11-93>
- 12 Chow, H. M. *et al.* Age-related hyperinsulinemia leads to insulin resistance in neurons and cell-cycle-induced senescence. *Nat Neurosci* **22**, 1806-1819 (2019). <https://doi.org/10.1038/s41593-019-0505-1>

- 13 Chidi-Ogbolu, N. & Baar, K. Effect of Estrogen on Musculoskeletal Performance and Injury Risk. *Front Physiol* **9**, 1834 (2018). <https://doi.org:10.3389/fphys.2018.01834>
- 14 Chow, H. M. *et al.* Low-Density Lipoprotein Receptor-Related Protein 6 Cell Surface Availability Regulates Fuel Metabolism in Astrocytes. *Adv Sci (Weinh)* **8**, e2004993 (2021). <https://doi.org:10.1002/advs.202004993>
- 15 Elia, J. *et al.* 4-cholesten-3-one decreases breast cancer cell viability and alters membrane raft-localized EGFR expression by reducing lipogenesis and enhancing LXR-dependent cholesterol transporters. *Lipids Health Dis* **18**, 168 (2019). <https://doi.org:10.1186/s12944-019-1103-7>
- 16 Burgess, J. W. *et al.* Phosphatidylinositol increases HDL-C levels in humans. *J Lipid Res* **46**, 350-355 (2005). <https://doi.org:10.1194/jlr.M400438-JLR200>
- 17 Kammerer, I., Ringseis, R., Biemann, R., Wen, G. & Eder, K. 13-hydroxy linoleic acid increases expression of the cholesterol transporters ABCA1, ABCG1 and SR-BI and stimulates apoA-I-dependent cholesterol efflux in RAW264.7 macrophages. *Lipids Health Dis* **10**, 222 (2011). <https://doi.org:10.1186/1476-511X-10-222>
- 18 Leonardi, R., Frank, M. W., Jackson, P. D., Rock, C. O. & Jackowski, S. Elimination of the CDP-ethanolamine pathway disrupts hepatic lipid homeostasis. *J Biol Chem* **284**, 27077-27089 (2009). <https://doi.org:10.1074/jbc.M109.031336>
- 19 Ravandi, A., Kuksis, A. & Shaikh, N. A. Glucosylated glycerophosphoethanolamines are the major LDL glycation products and increase LDL susceptibility to oxidation: evidence of their presence in atherosclerotic lesions. *Arterioscler Thromb Vasc Biol* **20**, 467-477 (2000). <https://doi.org:10.1161/01.atv.20.2.467>
- 20 Tabrizi, R. *et al.* The effects of inositol supplementation on lipid profiles among patients with metabolic diseases: a systematic review and meta-analysis of randomized controlled trials. *Lipids Health Dis* **17**, 123 (2018). <https://doi.org:10.1186/s12944-018-0779-4>
- 21 Faria-Pereira, A. & Morais, V. A. Synapses: The Brain's Energy-Demanding Sites. *Int J Mol Sci* **23** (2022). <https://doi.org:10.3390/ijms23073627>
- 22 Pulido, C. & Ryan, T. A. Synaptic vesicle pools are a major hidden resting metabolic burden of nerve terminals. *Sci Adv* **7**, eabi9027 (2021). <https://doi.org:10.1126/sciadv.abi9027>
- 23 Pathak, D. *et al.* The role of mitochondrially derived ATP in synaptic vesicle recycling. *J Biol Chem* **290**, 22325-22336 (2015). <https://doi.org:10.1074/jbc.M115.656405>
- 24 Aiello, G. L. & Bach-y-Rita, P. The cost of an action potential. *J Neurosci Methods* **103**, 145-149 (2000). [https://doi.org:10.1016/s0165-0270\(00\)00308-3](https://doi.org:10.1016/s0165-0270(00)00308-3)
- 25 Berndt, N. & Holzhutter, H. G. The high energy demand of neuronal cells caused by passive leak currents is not a waste of energy. *Cell Biochem Biophys* **67**, 527-535 (2013). <https://doi.org:10.1007/s12013-013-9538-3>
- 26 Chow, H. M. *et al.* ATM is activated by ATP depletion and modulates mitochondrial function through NRF1. *J Cell Biol* **218**, 909-928 (2019). <https://doi.org:10.1083/jcb.201806197>
- 27 Batra, R. *et al.* The landscape of metabolic brain alterations in Alzheimer's disease. *Alzheimers Dement* **19**, 980-998 (2023). <https://doi.org:10.1002/alz.12714>
- 28 Ludbrook, J. Advantages of permutation (randomization) tests in clinical and experimental pharmacology and physiology. *Clin Exp Pharmacol Physiol* **21**, 673-686 (1994). <https://doi.org:10.1111/j.1440-1681.1994.tb02570.x>

- 29 Azriel, D. & Schwartzman, A. The Empirical Distribution of a Large Number of Correlated Normal Variables. *J Am Stat Assoc* **110**, 1217-1228 (2015). <https://doi.org/10.1080/01621459.2014.958156>

Point-to-point response to reviewer's comments

Reviewer #1 (Remarks to the Author):

Comment 1—There is a single, still needed correction (that may have arisen because of a misuse of English): Line 799--change the word "enhanced" before night sweats to "decreased".

Response 1—We thank the reviewer for catching the detail. The wording is now corrected (Line 709).

Reviewer #2 (Remarks to the Author):

Comment 1—A legend identifying experimental groups is missing from figure 6D.

Response 1—We thank the reviewer for catching the details. The corresponding group labelling information is now added back to the figure panel.

Comment 2—I could not find a clear definition of how they decided whether a brain region was displaying 'neurite loss' for figures S3 and S11. This should be expanded upon in the methods.

Response 2— In response to the reviewer's comment, detailed descriptions of the methods used to generate Supplementary Figures 3i-j and 11c-d have been added back to the Methods section as follows (Lines 1263-1284):

“To estimate neurite integrity, MAP2 staining or ectopic expression of cytosolic GFP signals was performed on brain sections for subsequent analysis. For each condition, 10 images were captured using an epifluorescence microscope (Nikon) from five biological replicates, focusing on the prefrontal cortex and hippocampal regions (i.e., CA1-3). The images were binarized, and objects were analyzed in FIJI (Version 1.57q) to determine the average neurite length in the aforementioned regions. Total neurite length was normalized to the values obtained from the corresponding control treatment groups.

For the VCD study (Fig. 2j, Supplementary Fig.3i-j), five cohorts of mice were tested. The number of mice in each cohort showing signs of neurite loss—characterized by fragmentation and/or reduced MAP2 signal density within the same unit area compared to the vehicle control group—was counted (Supplementary Fig.3i) to determine which CA region was most severely affected. Additionally, the normalized ratio of total neurite length in VCD-treated groups relative to the control treatment group was calculated and presented (Supplementary Fig.3j).

Similarly, for the *Esrra* knockdown study (Fig. 5b, Supplementary Fig. 11c-d), four cohorts of mice were tested. The number of mice in each cohort exhibiting neurite loss—characterized by fragmentation and/or reduced densities of cytosolic GFP signals within the same unit area compared to the scramble shRNA control group—was counted (Supplementary Fig.11c) in the prefrontal cortex and CA1-2 hippocampal regions. Additionally, the normalized ratio of total neurite length in the *Esrra* knockdown group relative to the scramble shRNA control group was calculated and presented (Supplementary Fig.11d).”

Reviewer #3 (Remarks to the Author):

Comment 1—I still think that the abstract can be further improved. It should be more self-explanatory, for instance describing, when necessary, the human samples and specific mouse models (perimenopause/VCD, 3xAD-Tg...), including ages and sex.

Response 1—In response to the reviewer's and editor's request, we now rewrote the abstract to include more description of samples, age and sex in various key findings.